# Hessian Free Efficient Single Loop Iterative Differentiation Methods for Bi-Level Optimization Problems

**Peiran Yu**                                                              *peiran.yu@uta.edu*
*Department of Computer Science and Engineering*
*University of Texas at Arlington*

**Junyi Li**                                                              *junyili.ai@gmail.com*
*Department of Computer Science*
*University of Maryland, College Park*

**Heng Huang**                                                              *heng@umd.edu*
*Department of Computer Science*
*University of Maryland, College Park*

**Reviewed on OpenReview:** *https://openreview.net/forum?id=X59U5CHnfr*

## Abstract

Bilevel optimization problems have been actively studied in recent machine learning research due to their broad applications. In this work, we investigate single-loop methods with iterative differentiation (ITD) for nonconvex bilevel optimization problems. For deterministic bilevel problems, we propose an efficient single-loop ITD-type method (ES-ITDM). Our method employs historical updates to approximate the hypergradient. More importantly, based on ES-ITDM, we propose a new method that avoids computing Hessians. This Hessian-free method requires fewer backpropagations and thus has a lower computational cost. We analyze the convergence properties of the proposed methods in two aspects. We provide the convergence rates of the sequences generated by ES-ITD based on the Kurdyka-Łojasiewicz (KL) property. We also show that the Hessian-free stochastic ES-ITDM has the best-known complexity while has cheaper computation. The empirical studies show that our Hessian-free stochastic variant is more efficient than existing Hessian-free methods and other state-of-the-art bilevel optimization approaches.

## 1 Introduction

Bilevel optimization problems arise in various machine learning scenarios, including game theory Stackelberg (1952), meta-learning (Franceschi et al., 2018; Zügner & Günnemann, 2019; Finn et al., 2017a; Snell et al., 2017), hyperparameter optimization (Franceschi et al., 2017; Pedregosa, 2016; Grazzi et al., 2020; Mehra & Hamm, 2021; Maclaurin et al., 2015), and reinforcement learning (Hong et al., 2020). Please refer to (Liu et al., 2022b). A bilevel optimization problem involves two optimization problems, wherein one problem (the upper-level problem) includes the solution of another optimization problem (the lower-level problem). A typical formulation of this problem takes the following form:

$$
\begin{aligned}
\min_{x \in \mathbb{R}^n} \ & f(x) := F(x, y(x)), \\
\text{s.t.} \ \ & y(x) \in \operatorname*{arg\,min}_{y \in \mathbb{R}^m} G(x, y),
\end{aligned}
\tag{1}
$$

where $F : \mathbb{R}^n \times \mathbb{R}^m \to \mathbb{R}$ and $G : \mathbb{R}^n \times \mathbb{R}^m \to \mathbb{R}$ are continuously differentiable functions.

Gradient-based methods are popular for solving (1) due to their ease of implementation and efficiency. A key issue in gradient-based methods is how to approximate the hypergradient $\nabla f$. There are two prevailing

approaches in the current literature: iterative differentiation (ITD) that aims at approximating the Jacobian of $y(x)$ when calculating the hypergradient (Franceschi et al., 2017; 2018; Finn et al., 2017a; Liu et al., 2020; Ghadimi & Wang, 2018; Ji et al., 2021; Rajeswaran et al., 2019) and approximate implicit differentiation (AID) (Chen et al., 2022b; Ji et al., 2021; Li et al., 2022; Gould et al., 2016; Lorraine et al., 2020). The framework of ITD-based methods are easier to understand. Also, compared with AID-based methods, ITD-based methods dose not require solving an additional linear equation, which may have better accuracy in computation. In this paper, we focus on the ITD approach.

In general, if we view $y(x)$ as a function of $x$, ITD methods are designed based on the chain rule of the gradient of $F(x, y(x))$, which has the following form:

$$\nabla f(x) = \nabla_x F(x, y(x)) - J(y(x))\nabla_y F(x, y(x)). \tag{2}$$

How to approximate the $J(y(x))$ is the key concern of ITD methods. In Ghadimi & Wang (2018), the closed form of $J(y(x))$ was considered. Ghadimi & Wang (2018) showed that

$$J(y(x)) := -\nabla_{xy}G(x, y(x))^T \nabla_{yy}G(x, y(x))^{-1}.$$

The method proposed in Ghadimi & Wang (2018) replaced $(x, y(x))$ in the above formula with $(x^l, y^{l+1})$, where $l$ is the iteration. Franceschi et al. (2017) provided forward and backward ways to approximate $J(y(x^l))$ directly. Ji et al. (2021) also used the backward way to approximate $J(y(x^l))$. To reduce the time and space complexity of estimating $\nabla f(x^l)$, Shaban et al. (2019) proposed a truncated back-propagation to approximate $J(y(x^l))$. However, all above mentioned methods are double loop methods, which can be sophisticated and computationally expensive.

To enhance the efficiency of ITD-based methods, single-loop techniques have been proposed in prior work (Yang et al., 2021; Guo et al., 2021; Khanduri et al., 2021; Li et al., 2022; Chen et al., 2022a; Hong et al., 2020). Guo et al. (2021) introduced SVRB, which updates $y^{l+1}$ using a single gradient descent step. Chen et al. (2022a) proposed STABLE, which updates $x$ and $y$ based on continuous-time dynamics. One challenge faced by SVRB and STABLE is the need to compute a matrix inverse in each iteration, which is computational expensive. An alternative approach is to use the Neumann series, as seen in Khanduri et al. (2021); Yang et al. (2021); Hong et al. (2020). In particular, the inverse $\nabla_{yy}G(x, y)^{-1}$ is approximated by $\sum_{i=1}^{b} (I - \nabla_{yy}G(x, y))^i$ with some $b \in \mathbb{N}_+$. The greater $b$ is, the less error this approximation has. However, if $y$ is not a good approximation of $y(x)$, which is likely the case for most existing single-loop methods that only possess one inner loop to update $y$, the Neumann series approach to approximate $J(y(x))$ can still result in significant errors. In this study, we introduce a novel approach for approximating the Jacobian $J(y(x))$.

Another limitation in many existing ITD-type methods relies heavily on the calculation of the Hessian, its inverse, or the multiplication of the Hessian with a vector. These computations incur high computational costs. Additionally, scenarios may arise where computing the Hessian of $G$ is challenging or where $G$ lacks second-order differentiability. For instance, in cases where the lower-level problem involves robust regression, $G$ can take the form of the Huber loss, which is not twice differentiable (Huber, 1964; Hastie et al., 2009). Consequently, there has been a pursuit of methods that avoid Hessian computation. Gu et al. (2021) applied Gaussian smoothing (GS) techniques (Nesterov & Spokoiny, 2017), enabling their method to bypass the computation of gradients of $G$. However, Gu et al. (2021) does not leverage any first-order information from $F$ and $G$, potentially leading to a larger discrepancy between the approximation and the true hypergradient. Similarly, Sow et al. (2022) employed GS techniques to estimate the Jacobian of $y(x)$. Nevertheless, their resulting method needs to compute the full gradient of $G$, which is often impractical in real-world applications. Additionally, the methods introduced in Gu et al. (2021); Sow et al. (2022) involve a double-loop structure, which can be inefficient. Thus, in this work, we propose a stochastic single loop method that avoids calculating the Hessian of $G$.

Another aspect that is overlooked in single-loop bilevel methods is the convergence analysis of the generated sequences, specifically the convergence of $(x^l, y^l)$. Sequential convergence holds significance as it illustrates the method's behavior in the long run. For example, it helps determine whether the accumulated point of the generated sequence is a stationary point of (1), whether the generated sequence achieves global convergence, and the its convergence rate. The third goal of this work is to provide sequential convergence guarantees for the single-loop bilevel method.

## 1.1 Main Contributions of Our Paper

In this paper, we propose efficient single loop ITD-type (ES-ITD) methods for the deterministic and stochastic bilevel optimization problem (1) and (7). Our contributions are three fold:

- The proposed ES-ITD method makes use of much information about lower level updates as the double loop method has but retains the single loop computation cost.

- We propose a computationally efficient Hessian free method for the stochastic bilevel optimization problem (7). Based on a natural extension of ES-ITDM using the stochastic gradients or Hessians of $F$ and $G$, we use the Gaussian smoothing techniques to approximate the stochastic Hessian of $G$. The resulting method is called Hessian free stochastic ES-ITD method (HF-SES-ITD). Compared to current single loop method, HF-SES-ITD method only uses the first order information to approximate the Jacobian of $y(x)$. **Compared with Hessian vector multiplication, our method is much cheaper because it only needs computing gradient vector inner product**.

- We provide convergence analysis of the proposed methods. For the deterministic method, we analyze the convergence of the sequences $\{x^l\}$, $\{y^l\}$ and $\{J^l\}$ generated by ES-ITDM, where $J^l$ is the approximation of the Jacobian of $y(x^l)$. To this end, we propose a new potential function. We show that the iterative value of the potential function is nonincreasing. After that, we show that the successive changes of the generated sequence converge to zero. We show that any accumulation point of $\{x^l\}$ is a stationary point of (1). Furthermore, under the Kurdyka-Łojasiewicz (KL) property, we derive the convergence rates of $\{(x^l, y^l, J^l)\}$. Especially, when the potential function is a KL function with exponent $\frac{1}{2}$, we show that $\{(x^l, y^l, J^l)\}$ converges linearly. **As far as we know, this is the first work that provides the convergence rate of the sequences generated by single loop methods.** Compared with sequential convergence analysis for methods solving general nonconvex optimization problems, the nested formula for updating $J^{l+1}$ in our method makes a technical challenge in our sequential convergence analysis nontrivial. We also give convergence guarantees of the stochastic variant of ES-ITDM. We show the resulted method has a complexity of $O(\epsilon^{-2})$ to reach an $\epsilon$-stationary point under mild assumptions.

- We evaluate our methods via the hyper-parameters learning task. We first compare our methods with the current Hessian-free methods (Gu et al., 2021; Sow et al., 2022). Then we compare our methods with other popular single loop bilevel optimization methods in Franceschi et al. (2017); Grazzi et al. (2020); Ji et al. (2021); Grazzi et al. (2020); Sow et al. (2022); Guo et al. (2021); Chen et al. (2022a). In both comparisons, our stochastic Hessian free fully single loop method has better performance in both time and accuracy.

Table 1: Comparisons between our method and ESJ , BA, ITD-Bio, FMM = Forward Mode Method, TTSA, SUSTAIN, MRBO, SVRB, STABLE. HVM = Hessian Vector Multiplication; ds= diminishing stepsize; cs= constant stepszise; $\epsilon$ denotes an ($\epsilon$ + variance errors)-accuracy for $\|\nabla f(x^{\max_{\text{iter}}})\|$.

| | Methods | Loops | Use Matrix inverse | Use HVM | Sequential Convergence | Complexity |
|---|---|---|---|---|---|---|
| Deterministic Methods | ESJ Sow et al. (2022) | Double | No | No | | $O(\epsilon^{-4})$ with ds |
| | BA Ghadimi & Wang (2018) | | Yes | | | $O(\epsilon^{-2})$ with cs |
| | ITD-Bio Ji et al. (2021) | | | Yes | | $O(\epsilon^{-2})$ with cs |
| | FMM Franceschi et al. (2017) | | | | | |
| | ES-ITDM (Ours) | Single | No | | **Yes** | $O(\epsilon^{-2})$ with cs |
| Stochastic Methods | TTSA Hong et al. (2020) | Single | No | Yes | | $O(\epsilon^{-2.5})$ with ds |
| | SUSTAIN Khanduri et al. (2021) | | | | | $O(\epsilon^{-3})$ with ds |
| | MRBO Yang et al. (2021) | | | | | $O(\epsilon^{-3})$ with ds |
| | SVRB Guo et al. (2021) | | Yes | | | $O(\epsilon^{-3})$ with ds |
| | STABLEChen et al. (2022a) | | | | | $O(\epsilon^{-4})$ with ds |
| | SES-ITDM with (S) | | No | | | $O(\epsilon^{-2})$ with cs |
| | SES-ITDM with (HF) | | | **No** | | $O(\epsilon^{-2})$ with cs |

## 2 Related Work

Comparisons between our methods and current ITD methods are summarized in Table 1.

**Bilevel Optimization Methods.** When $F$ and $G$ in (1) are deterministic functions, Franceschi et al. (2017); Ji et al. (2021) proposed double loop ITD methods for (1). Ji et al. (2021) used the backward way to approximate $J(y(x^l))$. For stochastic problem (7), Ghadimi & Wang (2018) introduced a double loop method that solves the lower level subproblem with multiple stochastic gradient steps and replaces each element in (2) with the stochastic gradients and Hessian. Besides double loop methods, single loop methods were proposed in Yang et al. (2021); Guo et al. (2021); Khanduri et al. (2021); Li et al. (2022); Chen et al. (2022a); Hong et al. (2020); Li & Huang (2024). Guo et al. (2021) presented SVRB which updates $y^{l+1}$ using one stochastic gradient descent step. SVRB employs the variance reduction on each element in the right-hand side of (2). Chen et al. (2022a) proposed STABLE that updates $x$ and $y$ based on the continuous-time dynamics. Also using Neumann series, Yang et al. (2021) designed a momentum-based stochastic single loop method. Yu et al. (2024) considers adding dropout to address the overfitting problems in bilevel training tasks.

**Sequential Convergence Analysis.** The sequential convergence analysis is a fundamental problem of first-order methods. It has been investigated for nonconvex minimization problems (Bolte et al., 2014; Attouch et al., 2010; Yu et al., 2021; Li & Pong, 2016). However, the sequential analysis of bilevel optimization methods is still in its early stages. Chen et al. (2023) used the KL assumption to analyze the sequential convergence of an AID-type method. However, they assume that the linear equation in their AID method is solved exactly, which creates a gap between theory and practice. Additionally, the method they analyze is a double-loop method, whereas ours is single-looped. Liu et al. (2022a) demonstrated that the sequence generated by their proposed method accumulates at a stationary point. In our analysis, we further provide the convergence **rate** of the generated sequence. Under the KL assumption, we show that it can converge linearly, sublinearly, or finitely.

## 3 Notation and Preliminaries

In this paper, we denote $\mathbb{R}^n$ the $n$-dimensional Euclidean space with inner product $\langle \cdot, \cdot \rangle$ and Euclidean norm $\|\cdot\|$. We denote the spectrum norm of a matrix $A \in \mathbb{R}^{n \times m}$ as $\|A\|$ and the Frobenius norm of $A$ as $\|A\|_F$. For a ramdom variable $\xi$ defined on a probability space $(\Xi, \Sigma, P)$, we denote its expectation as $\mathbb{E}\xi$. Given an event $A$, the conditional expectation of $\xi$ is denoted as $\mathbb{E}_A(\xi)$.

We say an extended-real-valued function $f : \mathbb{R}^n \to [-\infty, \infty]$ is proper if $\mathrm{dom} f = \{x \in \mathbb{R}^n : f(x) < \infty\}$ is not empty and $f$ never equals $-\infty$. We say a proper function $f$ is closed if it is lower semicontinuous. A proper closed function is said to be level-bounded if for any $a \in \mathbb{R}$, the set $\{x : f(x) \le a\}$ is bounded. For a function $F : \mathbb{R}^{n+m} \to \mathbb{R}$, we denote the function $F(x, y)$ with respect to $y$ for a fixed $x$ as $F(x, \cdot)$ and denote the function $F(x, y)$ with respect to $x$ for a fixed $y$ as $F(\cdot, y)$. Following Rockafellar & Wets (1998), the regular subdifferential of a proper function $f$ at $x \in \mathrm{dom} f$ is defined as $\hat{\partial} f(x) := \{\xi \in \mathbb{R}^n : \liminf_{z \to x, \, z \ne x} \frac{f(z) - f(x) - \langle \xi, z - x \rangle}{\|z - x\|} \ge 0\}$. The subdifferential of $f$ at $x \in \mathrm{dom} f$ is defined by $\partial f(x) := \{\xi \in \mathbb{R}^n : \exists x^k$ with $x^k \to x$ and $f(x^k) \to f(x), \, \xi^k \to \xi$ with $\xi^k \in \hat{\partial} f(x^k), \, \forall k\}$. For $x \notin \mathrm{dom} f$, we define $\hat{\partial} f(x) = \partial f(x) = \emptyset$. We denote $\mathrm{dom} \partial f := \{x : \partial f(x) \ne \emptyset\}$. We say $x$ is a sationary point of $f$ if $0 \in \partial f(x)$. For a twice differential function $F : \mathbb{R}^m \times \mathbb{R}^n \to \mathbb{R}$, we denote $\nabla_x F(x, y)$ and $\nabla_y F(x, y)$ as the partial gradients $\frac{\partial F(x,y)}{\partial x}$ and $\frac{\partial F(x,y)}{\partial y}$ correspondingly. We denote $\nabla_{xy} F(x, y) := \frac{\partial^2 F(x,y)}{\partial x \partial y}$ and $\nabla_{yy} F(x, y) := \frac{\partial^2 F(x,y)}{\partial y \partial y}$. For a function $g(x) := \mathbb{E}_{\xi \sim P} g(x; \xi)$ with distribution $P$, let $\mathcal{S} = \{\xi_j\}_{j=1}^{|s|}$ be a mini-batch of samples drawn from $P$. We denote $g(x; \mathcal{S}) := \frac{1}{|\mathcal{S}|} \sum_{j=1}^{|\mathcal{S}|} g(x; \xi_j)$.

Now, we first present the basic assumptions for (1).

**Assumption 1.** *Consider* (1). *Suppose the following assumptions hold:*

(i) *Suppose $F$ is Lipschitz continuous with modulus $L_F$.*

(ii) *For any fixed $\bar{x}$, $\nabla_x F(\bar{x}, \cdot)$ and $\nabla_y F(\bar{x}, \cdot)$ are Lipschitz continuous with modulus $L_{12}^F > 0$ and $L_{22}^F > 0$ respectively, i.e., for any $y_1$ and $y_2$, it holds that $\|\nabla_x F(\bar{x}, y_1) - \nabla_x F(\bar{x}, y_2)\| \le L_{12}^F$ and $\|\nabla_y F(\bar{x}, y_1) - \nabla_y F(\bar{x}, y_2)\| \le L_{22}^F$.*

*(iii) There exists $C_y^F > 0$ such that $\|\nabla_y F(x, y)\| \leq C_y^F$ for any $x$ and $y$.*

*(iv) For any fixed $\bar{y}$, $\nabla_y F(\cdot, \bar{y})$ is Lipschitz continuous with modulus $L_{21}^F > 0$.*

**Assumption 2.** *Consider* (1). *Suppose the following assumptions hold:*

*(i) Denote $z = (x, y)$. $G$ is twice continuously differentiable in $z$. In addition, suppose $\nabla_y G(x, \cdot)$ is Lipschitz continuous with modulus $L_G > 0$ for any $x$.*

*(ii) For any $x$, $G(x, \cdot)$ is strongly convex with modulus $\mu_G$.*

*(iii) For any $x$, $\nabla_{xy}^2 G(x, \cdot)$ and $\nabla_{yy}^2 G(x, \cdot)$ are Lipschitz continuous with modulus $L_{G_x^y}$ and $L_{G_y^y}$.*

*(iv) For any $y$, $\nabla_{xy}^2 G(\cdot, y)$ and $\nabla_{yy}^2 G(\cdot, y)$ are Lipschitz continuous with modulus $L_{G_x^x}$ and $L_{G_y^x}$.*

*(v) There exist $C_{Gxy} > 0$ and $C_{Gyy} > 0$ such that $\|\nabla_{xy}^2 G(x, y)\| \leq C_{Gxy}$ and $\|\nabla_{yy}^2 G(x, y)\| \leq C_{Gyy}$ for any $x$ and $y$.*

## 4 Efficient Single Loop ITD Method for Deterministic Problem (1)

To develop the fully single loop method, we investigate a basic double loop ITD algorithm. At each iteration $l$, let $y^{l+1} = y^{l,K}(x^l)$, where $y^{l,K}(x^l)$ is the output of Algorithm 1. Note that $y^{l,K}(x^l)$ is a function of $x^l$.

---

**Algorithm 1** A gradient descent method for $\min_y G(x^l, y)$

---

Input $y^{l,0}$, $\gamma > 0$.
For $k = 0, \ldots, K-1$
Let $y^{l,k+1}(x^l) = y^{l,k}(x^l) - \gamma \nabla_y G(x^l, y^{l,k})$.
Output $y^{l+1} = y^{l,K}(x^l)$.

---

The double loop ITD method approximates the Jacobian of $y(x)$ with the Jacobian of $y^{l,K}(x^l)$. Using the chain rule, we have the following formula for the Jacobian of $y^{l,K}(x^l)$. [1]

$$J(y^{l,K}(x^l)) = -\sum_{k=0}^{K-1} \gamma \nabla_{xy} G(x^l, y^{l,k}(x^l)) \times \prod_{s=k+1}^{K-1} \left( I - \gamma \nabla_{yy} G(x^l, y^{l,s}(x^l)) \right). \tag{3}$$

Then, at the outer loop, we let $J^{l+1} = J(y^{l,K}(x^l))$ and update $x^{l+1}$ with

$$x^{l+1} = x^l - \beta \nabla_x F(x^l, y^{l+1}) + (J^{l+1})^T \nabla_y F(x^l, y^{l+1}).$$

If $K = 1$, we get a single loop bilevel method. However, the resulting Jacobian approximation becomes $J(y^{l,1}(x^l)) = -\gamma \nabla_{xy} G(x^l, y^{l,0})$. $\nabla_{xy} G(x^l, y^{l,0})$ does not make use of previous information about the $y$ coordinate. Even if we choose $y^{l,0}$ in Algorithm 1 to be the last update $y^l$, $J(y^{l,1}(x^l))$ still does not make use of the previous information such as $\{\nabla_{yy} G(x^l, y^l)\}_l$ or $\{\nabla_{xy} G(x^l, y^l)\}_l$. Thus, simply changing the double loop method to a single loop method by letting $K = 1$ will lose information in the previous iterates. However, we notice that (3) implies

$$J(y^{l,K+1}(x^l)) = J(y^{l,K}(x^l))\left( I - \gamma \nabla_{yy} G(x^l, y^{l,K}(x^l)) \right) - \gamma \nabla_{xy} G(x^l, y^{l,K}(x^l)). \tag{4}$$

Inspired by this, we update

$$J^{l+1} = J^l \left( I - \gamma \nabla_{yy} G(x^l, y^l) \right) - \gamma \nabla_{xy} G(x^l, y^l) \tag{5}$$

---

[1] We denote $\Pi_{s=k+1}^{K-1} \left( I - \gamma \nabla_{yy} G(x^l, y^{l,s}) \right)$ as $I$ when $K = 1$ by convention

and propose Algorithm 2. In this way, we make full use of the previous information to update the approximation of the Jacobian. (5) implies that

$$J^{l+1} = -\sum_{k=0}^{l} \gamma \nabla_{xy} G(x^k, y^k) \times \prod_{s=k+1}^{l} (I - \gamma \nabla_{yy} G(x^s, y^s)) \tag{6}$$

Comparing (6) with (3), there are the following differences:

- At iteration $l$, computing $J(y^{l,K}(x^l))\nabla_y F(x^l, y^{l+1})$ based on (3) needs $O(K^2)$ Hessian-vector multiplications (HVM), where $K$ is the iterations of the inner loop. The cost of each HVM is $O(\max\{n, m\})$ using the technic in Pearlmutter (1994). Here, We have two equivalent ways to update $J^{l+1}\nabla_y F(x^l, y^{l+1})$: either by using (5) and making use of $J^l$, or by using (6) without involving $J^l$. When applying deduction (5) in the algorithm, we need $n+1$ HVM computations. However, when using (5) to update $J^{l+1}$, we must compute $J^{l+1} = J^l \left( I - \gamma \nabla_{yy} G(x^l, y^l) \right) - \gamma \nabla_{xy} G(x^l, y^l)$, which involves an $\mathbb{R}^{n \times m} \times \mathbb{R}^{m \times m}$ matrix operation. When calculating $J^l \times \nabla_{yy} G(x^l, y^l)$, it requires $n$ HVM computations. Therefore, in each iteration, we actually need $n+1$ HVM computations, which is less than $O(k^2)$. Thus, our method is advantageous when $n+1 \leq k^2$. On the other hand, when using (6), we require $l^2$ HVM computations. In summary, our method outperforms the classical ITD method when either $n+1 \leq k^2$ or $l^2 < k^2$.

- The formula in (6) makes use of historical updates $\{x^0, \ldots, x^l\}$ and $\{y^0, \ldots, y^l\}$, while (3) only depends on $\{x^l\}$. Although making use of historical updates requires more storage, (6) makes use of more information to approximate the Jacobian of $y(x^l)$.

---

**Algorithm 2** Efficient Single ITD Method (ES-ITDM) for (1)

1: Input: $\alpha, \beta, \gamma > 0$, $x^0 \in \mathbb{R}^n$, $y^0 \in \mathbb{R}^m$, $J^0 \in \mathbb{R}^{n \times m}$ and $N \in \mathbb{N}_+$.
2: **for** $l = 0, \ldots, N-1$ **do**
3:    Let $y^{l+1} = y^l - \alpha \nabla_y G(x^l, y^l)$.
4:    Compute
$$\hat{\nabla} f(x^l) = \nabla_x F(x^l, y^{l+1}) + (J^{l+1})^T \nabla_y F(x^l, y^{l+1}).$$
   with $J^{l+1}$ be defined as in (6).
5:    Let $x^{l+1} = x^l - \beta \hat{\nabla} f(x^l)$.
6: **end for**
7: Output $(y^N, x^N)$.

---

## 5 Efficient Single Loop ITD Method for Stochastic Problem (7)

In many applications that involve bilevel optimization problems, we need to consider a stochastic bilevel problem:

$$\begin{aligned}
&\min_{x \in \mathbb{R}^n} f(x) := \mathbb{E}_{\xi \sim P} F(x, y(x); \xi), \\
&\text{s.t. } y(x) \in \arg\min_{y \in \mathbb{R}^m} \mathbb{E}_{\eta \sim P'} G(x, y; \eta),
\end{aligned} \tag{7}$$

where $P$ and $P'$ are distributions.

One natural idea is to replace the gradient/Hessian in Algorithm 2 with the mini-batch gradients/Hessian of $F$ and $G$. At iteration $l$, we draw a sample batch $\mathcal{S}^l$. Then using

$$\tilde{J}^{l+1} = -\sum_{k=0}^{l} \gamma \nabla_{xy} G(x^k, y^k; \mathcal{S}^l) \times \prod_{s=k+1}^{l} (I - \gamma \nabla_{yy} G(x^k, y^k; \mathcal{S}^l)) \tag{8}$$

to approximate $J^{l+1}$ in (6).

Now, based on the mini-batch Hessian of $G$, we propose a Hessian-free approach to estimate the Hessian of $G$. We present this approach as (HF) in Algorithm 3.

---

**Algorithm 3** Hessian Free Stochastic efficient single loop stochastic ITD method (HF-SES-ITDM) for (7)

1: Input: $\alpha, \gamma, \beta, \mu, \nu > 0$, $x^0 \in \mathbb{R}^n$, $y^0 \in \mathbb{R}^m$, $N \in \mathbb{N}_+$.
2: **for** $l = 0, 1, 2, \cdots, N$ **do**
3:    Draw a sample batch $\mathcal{I}^l$ change the notation in the proofs. Let $y^{l+1} = y^l - \alpha \nabla_y G(x^l, y^l; \mathcal{I}^l)$.
4:    Draw a sample batch $\mathcal{B}^l$ and $\mathcal{S}^l$. Compute

$$\hat{\nabla} f(x^l) = \nabla_x F(x^l, y^{l+1}; \mathcal{B}^l) + (\hat{J}^{l+1})^T \nabla_y F(x^l, y^{l+1}; \mathcal{B}^l),$$

   with $\hat{J}^{l+1}$ defined in (11).
5:    Let $x^{l+1} = x^l - \beta \hat{\nabla} f(x^l)$.
6: **end for**

---

The idea of Algorithm 3 is as follows. We aim to approximate $\nabla_{xy} G(x, y; \mathcal{S}^l)$ and $\nabla_{yy} G(x, y; \mathcal{S}^l)$ with the first-order information of $G$. $\nabla_{xy} G(x, y; \mathcal{S}^l)$ and $\nabla_{yy} G(x, y; \mathcal{S}^l)$ consist of the gradients of $\{\nabla_{y_j} G(x^l, y^l; \mathcal{S}^l)\}_{j=1}^m$ with respect to $x$ and $y$. Inspired by Nesterov & Spokoiny (2017), we use Gaussian smoothing technique as follows. Pick $Q \in \mathbb{N}$, $\mu > 0$ and $\nu > 0$. For $j = 1, \ldots, Q$, generate $u_j^l \sim \mathcal{N}(0, I) \in \mathbb{R}^n$ and $v_j^l \sim \mathcal{N}(0, I) \in \mathbb{R}^m$. For $q = 1, \ldots, m$, let

$$\nabla_x \nabla_{y_q} G(x^l, y^l; \mathcal{S}^l) \approx \frac{1}{Q} \sum_{j=1}^Q \frac{\nabla_{y_q} G(x^l + \mu u_j^l, y^l; \mathcal{S}^l) - \nabla_{y_q} G(x^l, y^l; \mathcal{S}^l)}{\mu} u_j^l$$

and

$$\nabla_y \nabla_{y_q} G(x^l, y^l; \mathcal{S}^l) \approx \frac{1}{Q} \sum_{j=1}^Q \frac{\nabla_{y_q} G(x^l, y^l + \nu v_j^l; \mathcal{S}^l) - \nabla_{y_q} G(x^l, y^l; \mathcal{S}^l)}{\nu} v_j^l.$$

Notice that the fractions in the above equalities are scalers of dimension 1. Now, we approximate $\nabla_{xy} G(x, y; \mathcal{S}^l)$ and $\nabla_{yy} G(x, y; \mathcal{S}^l)$ as follows:

$$\nabla_{xy} G(x^l, y^l; \mathcal{S}^l) = \begin{bmatrix} (\nabla_x \nabla_{y_1} G(x^l, y^l; \mathcal{S}^l)^T \\ \vdots \\ (\nabla_x \nabla_{y_m} G(x^l, y^l; \mathcal{S}^l)^T \end{bmatrix} \approx \frac{1}{\mu Q} \sum_{j=1}^Q \left( \nabla G(x^l + \mu u_j^l, y^l; \mathcal{S}^l) - \nabla G(x^l, y^l; \mathcal{S}^l) \right) (u_j^l)^T \tag{9}$$

$$=: \hat{\nabla}_{xy} G(x^l, y^l)$$

and

$$\nabla_{yy} G(x^l, y^l; \mathcal{S}^l) = \begin{bmatrix} \nabla_y \nabla_{y_1} G(x^l, y^l; \mathcal{S}^l)^T \\ \vdots \\ \nabla_y \nabla_{y_m} G(x^l, y^l; \mathcal{S}^l)^T \end{bmatrix} \approx \frac{1}{\nu Q} \sum_{j=1}^Q \left( \nabla G(x^l, y^l + \nu v_j^l; \mathcal{S}^l) - \nabla G(x^l, y^l; \mathcal{S}^l) \right) (v_j^l)^T \tag{10}$$

$$=: \hat{\nabla}_{yy} G(x^l, y^l).$$

Now, we use

$$\hat{J}^{l+1} = -\sum_{k=0}^l \gamma \hat{\nabla}_{xy} G(x^k, y^k) \times \prod_{s=k+1}^l \left( I - \gamma \hat{\nabla}_{yy} G(x^k, y^k) \right) \tag{11}$$

to approximate $J^{l+1}$ in (6).

**Remark 1.** *Now we compare the computation cost of SES-ITDM and HF-SES-ITDM. The main cost is computing*

$$\mathfrak{D}^l := \hat{J}^{l+1} \nabla_y F(x^l, y^{l+1}; \mathcal{B}^l).$$

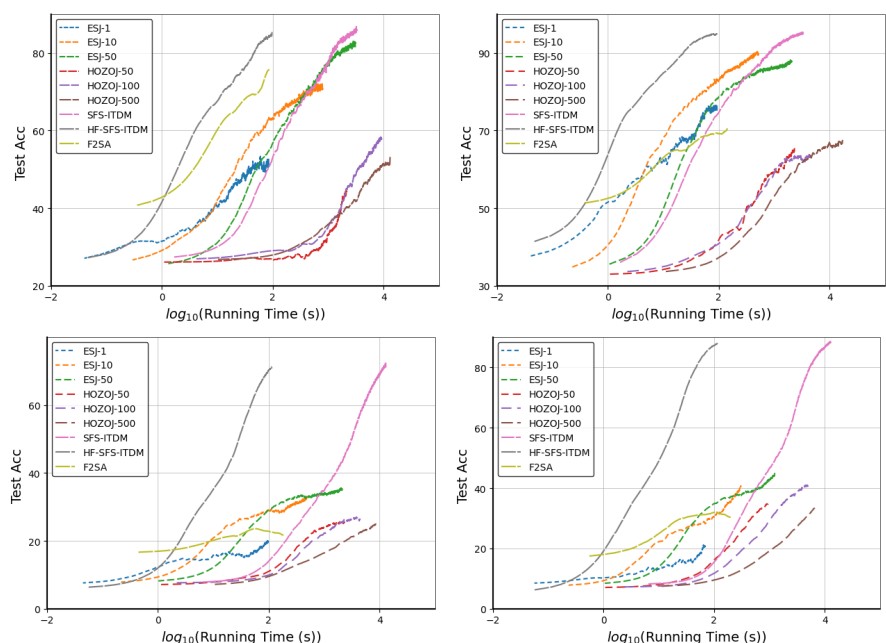

Figure 1: Test Accuracy *w.r.t.* $log_{10}$(Training Time) for the hyper-representation task. The top two plots are 5-way-1-shot (left) and 5-way-5-shot (right), and the bottom two plots are 20-way-1-shot (left) and 20-way-5-shot (right). Postfix of ESJ and HOZOJ methods is the number of queried noise vectors ($Q$).

When $\hat{J}^{l+1}$ is chosen as (8), computing $\mathfrak{D}^l$ needs to compute the HVM. To this end, we need to compute the gradient $\nabla_y G$ first and then compute the HVM, which requires 2 backpropagation in implementation. However, when $\hat{J}^{l+1}$ is chosen as (11), computing $\mathfrak{D}^l$ only needs to compute the gradient $\nabla_y G$ and one vector inner product using the formulas (9) and (10). Thus, computing $\mathfrak{D}^l$ only need 1 backpropagation in implementation. When the dimension of parameters is large, this can save time compared to methods that need to compute the gradients and HVM. However, computing $\mathfrak{D}^l$ with $\hat{J}^{l+1}$ in (11) needs to calculate $Q$ times of gradients and inner products. Thus, there is a tradeoff between using (8) and (11).

## 6 Convergence Analysis

### 6.1 Convergence Analysis for Algorithm 2

Now we analyze Algorithm 2. We first give the following theorem that will be used in proving our main convergence properties.

**Theorem 1.** *Consider* (1) *and let Assumptions 1 and 2 hold. Let* $\{(x^l, y^l, J^l)\}$ *be generated by Algorithm 2. Suppose $F$ is bounded from below. Suppose $\alpha \in (0, \frac{1}{\mu_G})$. Then there exist $\beta > 0$ and $\gamma \in (0, \frac{1}{\mu_g})$, $\Delta > 0$ and $\delta \in (0, 1)$ such that*

$$H^{l+1} \leq H^l - \Delta\|x^l - x^{l-1}\|^2 - \delta\|y(x^l) - y^{l+1}\|^2 - \delta\|J(y(x^l)) - J^{l+1}\|_F^2,$$

*where $H^l := H(x^l, x^{l-1}, y^{l+1}, J^{l+1})$ with*

$$H(x, x', y, J) := f(x) + \Delta\|x - x'\|^2 + \|y(x) - y\|^2 + \|J(y(x)) - J\|_F^2.$$

*Furthermore, $\{H^l\}$ is nonincreasing and there exists $H^*$ such that $\{H^{l+1}\}$ is convergent to $H^*$.*

Thanks to Theorem 1, we now have the following observation indicating the limiting point of the generated sequence satisfies the first order optimality condition of (1).

**Corollary 1.** *Let assumptions in Theorem 1 hold and suppose $F$ is level-bounded. Then any accumulation point $x^*$ of $\{x^l\}$ satisfies $\nabla f(x^*) = 0$. In addition, any accumulation point $y^*$ of $\{y^l\}$ is an optimal solution of the lower level problem in (1) defined with $x^*$.*

Now, we show the global convergence properties of the sequences generated by Algorithm 2. To this end, we introduce the Kurdyka-Łojasiewicz (KL) property.

**Definition 1** (**Kurdyka-Łojasiewicz Function**). *We say a proper closed function $f : \mathbb{R}^n \to (-\infty, \infty]$ satisfies the Kurdyka-Łojasiewicz (KL) property at $\hat{x} \in \mathrm{dom}\partial f$ with exponent $\vartheta \in [0, 1)$ if there are $a \in (0, \infty]$, a neighborhood $V$ of $\hat{x}$ and $a_0 > 0$ such that $\mathrm{dist}(0, \partial f(x)) \geq a_0 (f(x) - f(\hat{x}))^\vartheta$ for any $x \in V$ with $f(\hat{x}) < f(x) < f(\hat{x}) + a$. A proper closed function $f$ satisfying the KL property with exponent $\vartheta \in [0, 1)$ at every point in $\mathrm{dom}\,\partial f$ is called a KL function with exponent $\vartheta$.*

Many functions are KL functions. It is known that proper closed semi-algebraic functions (i.e., functions whose graphs are unions and intersections of polynomial functions) satisfy the KL property, Attouch et al. (2010); Li & Pong (2018); Attouch et al. (2013); Bolte et al. (2017). Semi-algebraic functions include widely used losses such as quadratic loss, L2 loss, Huber loss, hinge loss, and 0-1 loss. KL property is a general property in convergence analysis when the considered function is not smoothness.

Under the KL assumption, we have the following convergence rate of $\{(x^l, y^l, J^{l+1})\}$ generated by 2.

**Theorem 2.** *Consider (1). Let $H$ be defined as in Theorem 1. Suppose assumptions in Theorem 1 hold and $F$ is level-bounded. Suppose $H$ is a KL function with exponent $\vartheta$, then*

*(i) when $\vartheta = 0$, $\{(x^l, y^l, J^{l+1})\}$ converges finitely;*

*(ii) when $\vartheta \in (0, \frac{1}{2}]$, $\{(x^l, y^l, J^{l+1})\}$ converges linearly;*

*(iii) when $\vartheta \in (\frac{1}{2}, 1)$, $\{(x^l, y^l, J^{l+1})\}$ converges sublinearly.*

**Remark 2.** *Note that, together with Corollary 1, Theorem 2 shows that the limiting point $x^*$, to which the sequence $x^l$ converges, is a stationary point of (1). In addition, the limiting point $y^*$, to which the sequence $y^l$ converges, is the optimal solution of the lower-level problem in (1) defined with $x^*$.*

**Remark 3.** *Since the lower-level problem is strongly convex, the lower-level minimizer is unique. When the lower-level problem is semi-algebraic, and since $y(x)$ is the infimum projection of a semi-algebraic function, its graph is also the intersection of polynomial functions, making $y(x)$ semi-algebraic, as well as its Jacobian $J(y(x))$. If the upper-level objective is also semi-algebraic, then the potential function $H$ is a KL function, satisfying the assumption in Theorem 2.*

## 6.2 Convergence Analysis for Algorithm 3

In this section, we make the following assumption.

**Assumption 3.** *Consider (7). Suppose for any fixed $y$,*

$$\mathbb{E}\nabla_x F(x, y; \xi) = \nabla_x F(x, y)$$
$$\mathbb{E}\|\nabla_x F(x, y; \xi) - \nabla_x F(x, y)\|^2 \leq \sigma_B^2.$$

*Suppose*

$$\mathbb{E}\|\nabla_y G(x, y; \xi) - \nabla_y G(x, y)\|^2 \leq \sigma_G^2,$$
$$\mathbb{E}\|\nabla_{xy} G(x, y; \xi) - \nabla_{xy} G(x, y)\|^2 \leq \sigma_{xy}^2,$$
$$\mathbb{E}\|\nabla_{yy} G(x, y; \xi) - \nabla_{yy} G(x, y)\|^2 \leq \sigma_{yy}^2.$$

Now, we present the short version of the convergence result of Algorithm 3.

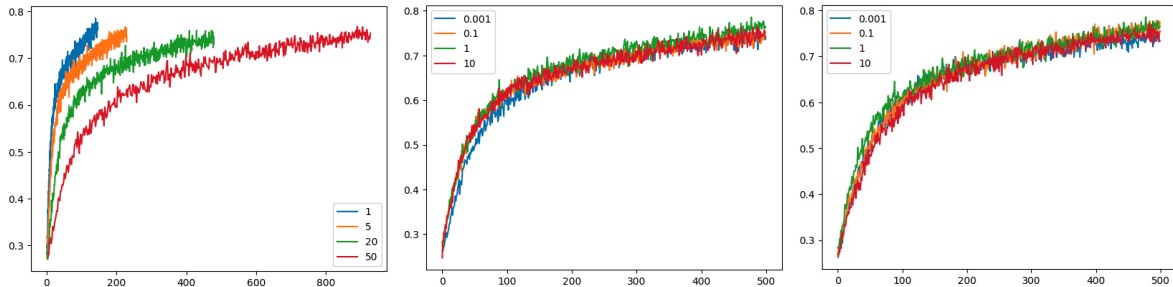

Figure 2: Test Accuracy *w.r.t.* $log_{10}$(Training Time) for the hyper-representation task. Ablation study HF-SES-ITDM behave when choosing different $Q$, $\nu$ and $\mu$. As we can see, the greater $Q$ is, the slower the algorithm is. But when $Q = 1$, the accuracy is still competitive with the accuracy with greater $Q$'s

**Theorem 3.** *Suppose Assumptions 1, 2 and 3 hold. Suppose $\alpha < \frac{1}{L_G}$. Let $\{x^l\}$ be generated by Algorithm 3 with $\hat{J}^{l+1}$ in Step 4 being generated by (11) based on Gaussian sampling. Suppose the optimal value for (7) is $f^* > -\infty$. Then there exists $\beta$ small enough such that*

$$\frac{1}{N+1} \sum_{l=0}^{N} \mathbb{E}\|\nabla f(x^l)\|^2 \leq \frac{1}{N+1} E_1 + E_2,$$

*where*

$$E_1 = A_1(C_y^F)^2 + A_1(f(x^0) - f^*) + A_1\|y(x^0) - y^0\|^2 + A_1\|J(y(x^0)) - \nabla_{xy}G(x^0, y^0)\|^2$$

*for some $A_1 > 0$, $E_2 = A_2 \max\{\bar{\Delta}_{xy} + \bar{\Delta}_{yy}, \sigma_B^2, \sigma_G^2\}$ for some $A_2 > 0$ with $\bar{\Delta}_{xy}$ and $\bar{\Delta}_{yy}$ being the upper bound of $\mathbb{E}_{v^l, S^l|R^l}\|\nabla_{xy}G(x^l, y^l) - \hat{\nabla}_{xy}G(x^l, y^l)\|^2$ and $\mathbb{E}_{v^l, S^l|R^l}\|\nabla_{yy}G(x^l, y^l) - \hat{\nabla}_{yy}G(x^l, y^l)\|^2$ respectively.*

## 7  Experiments

In this section, we test the efficacy of the proposed algorithms: Algorithm 2 and Algorithm 3 on the hyper-representation learning task Franceschi et al. (2018).[2] Hyper-representation refers to a shared representation (or shared deep neural network) across multiple tasks in a meta-learning framework. The parameters in the shared representation are referred to as hyperparameters. Let $\phi(\cdot; \lambda)$ denote the hyper-representation mapping, parameterized by $\lambda$. When applying this to solve a specific classification task, a linear layer $w$ is added on top of the hyper-representation, and only the parameters $w$ are trained, while the parameters $\lambda$ in the shared "hyper-representation" remain fixed. To train the parameters in the hyper-representation, [Franceschi et al. (2018)] formulated it as the bilevel programming problem (1), where the upper-level objective minimizes a validation loss, and the lower-level objective minimizes a task-specific training loss. Specifically, we have the following problem:

$$\min_{\lambda} \frac{1}{|D_{\mathcal{V},\xi}|} \sum_{(x_i,y_i) \in D_{\mathcal{V},\xi}} l((\omega^*(\lambda))^T \phi(x_i; \lambda), y_i)$$
$$\text{s.t.} \quad w^*(\lambda) = \arg\min_{w} l_{tr}(\lambda, w),$$

where

$$l_{tr}(\lambda, w) := \frac{1}{|D_{\mathcal{T},\xi}|} \sum_{(x_i,y_i) \in D_{\mathcal{T},\xi}} l(\omega^T \phi(x_i; \lambda), y_i) + C\|w\|^2$$

---

[2]Our code is available at `https://github.com/Peiran225/Bilevel_Hessian_free`.

with $l(\cdot)$ denoting the cross entropy loss, $D_{\mathcal{T},\xi}$ and $D_{\mathcal{V},\xi}$ being training and validation dataset for a randomly sampled meta task. Here $\lambda = \{\lambda_i\}_{i \in \mathcal{D}_{\mathcal{T}}}$ are hyper-representations and $C \geq 0$ is a tuning parameter to gaurantee the inner problem to be strongly convex. In experiment, we set $C = 0.01$. All experiments are run over a machine with Intel Xeon Gold 6248 CPU and 4 Nvidia Tesla V100 GPUs. The code is written with Pytorch.

In the task, we perform the hyper-representation learning task over the Omniglot dataset Lake et al. (2015), the details of the ominiglot dataset and the formulation of our hyper-representation task are included in the supplementary. In general the target of our task is to learn a useful hyper-representation such that we can learn a linear classifier on top of it with a small number of samples and training cost.

We first compare our algorithms with two existing Hessian free methods, *i.e.* the HOZOJ Gu et al. (2021), ESJ Sow et al. (2022) and F2SA in Kwon et al. (2023). HOZOJ is a hyper-parameter optimization method which applies the evolution strategy over the hyper-gradient directly. ESJ applies the evolution strategy over the Jacobian matrix. Kwon et al. (2023) view the bilevel problem as a constrained optimization problem and use a penalty-type method that only requires the first-order information. As a comparison, our **HF-SES-ITDM** instead approximates the second order derivatives $\nabla_{xy}G$ and $\nabla_{yy}G$. The experimental results are summarized in Fig. 1. We search hyper-parameters for all methods, and we find setting the outer learning rate $\beta = 0.1$ and the inner learning rate $\alpha = 0.4$ can get good performance for all methods. For F2SA, we set the additional Lagrange multiplier as 2. We find the scale of the noise ($\mu$ and $\nu$ in **HF-SES-ITDM** and $\mu$ in ESJ and HOZOJ) and the number of Gaussian vectors ($Q$) are very important for the model performance.

We also test HF-SES-ITDM with (11) being replaced by (8), which is named as SES-ITDM. As shown by Fig. 1, **HF-SES-ITDM** greatly accelerates the **SES-ITDM** by avoiding computing the second derivatives explicitly. Furthermore, the ESJ and HOZOJ methods need to query a relatively large amount of noise vectors to get good performance. For example, in the 5 way 1 shot case, ESJ needs to query 50 noise vectors to reach the same test accuracy as **HF-SES-ITDM** with $Q = 1$, and has longer running time cost. For HOZOJ, we need to query at least 50 noise vector to get converged training, while for 100 and 500 noise vectors, we still observe a great performance margin compared to our method. The need of many noise vectors for ESJ and HOZOJ might come from the evolution strategy used in both methods to approximate higher level properties, and thus lead to more noise.

Figure 2 shows how HF-SES-ITDM behave when choosing different $Q$, $\nu$ and $\mu$. As we can see, the greater $Q$ is, the slower the algorithm is. But when $Q = 1$, the accuracy is still competitive with the accuracy with greater $Q$'s.

## 8 Conclusions

In this paper, we focused on studying the bilevel optimization problem and proposed a new single-loop ITD method that is more efficient in approximating the Jacobian of the lower-level solution with respect to the upper-level variable. We proposed a Hessian-free stochastic bilevel optimization. Based on a natural stochastic extension of ES-ITDM, we first proposed a Hessian-free stochastic ES-ITDM. This Hessian-free variant eliminates the need to compute the Hessian vector multiplication, thus potentially leading to faster implementation. We theoretically analyze the proposed methods. For the deterministic method, we investigate the global convergence rates of the generated sequences. As for the stochastic method, we conducted an analysis of its complexity. Our methods were validated using the hyper-representation learning task. In experiments, our Hessian-free stochastic ES-ITDM demonstrated greater efficiency compared to existing Hessian-free methods.

## Acknowledgment

This work was partially supported by NSF IIS 2347592, 2347604, 2348159, 2348169, DBI 2405416, CCF 2348306, and CNS 2347617.

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

## A    Additional Preliminaries

For any matrices $A$ and $B$, we denote $\text{trace}(A^T B) := \langle A, B \rangle$. Given independent random variables $\xi_1, \ldots, \xi_p$ and a function $f(\xi_1, \ldots, \xi_p)$, we denote the conditional expectation of $g$ with respect to $\xi_i$ as $\mathbb{E}_{\xi_i|A} g(\xi_1, \ldots, \xi_p)$.

Under Assumptions 1 and 2, we have the following properties (Lemma 2.1 and Lemma 2.2 in Ghadimi & Wang (2018)) about $f$ in (1).

**Lemma 1.** *Consider* (1) *and suppose Assumptions 1 and 2 hold. Then $y(x)$ is Lipschitz continuous with $L_y := \frac{C_{Gxy}}{\mu_G}$. In addition, $\nabla f$ is Lipschitz continuous with modulus $L_f := \frac{(L_{21}^F + C)C_{Gxy}}{\mu_G} + L_{12}^F + C_y^F \left( \frac{L_{G_x^x} C_y^F}{\mu_G} + \frac{L_{G_x^x} C_{Gxy}}{\mu_G^2} \right)$, where $C := L_{12}^F + \frac{L_{22}^F}{\mu_G} + C_y^F \left( \frac{L_{G_y^y} C_y^F}{\mu_G} + \frac{L_{G_y^y} C_{Gxy}}{\mu_G^2} \right)$.*

**Lemma 2.** *Consider* (1) *and suppose Assumption 2 holds. Then for any $x$, it holds that $J(y(x)) = - \left( \nabla_{yy}^2 G(x, y(x)) \right)^{-1} \nabla_{xy}^2 G(x, y(x))$ is Lipschitz continuous with modulus $L_J := \frac{L_{G_x^x} + L_{G_x^y} L_y}{\mu_G} + \frac{(L_{G_y^x} + L_{G_y^y} L_y) C_{Gxy}}{\mu_G^2}$. Also, for any $x$, it holds that $\|J(y(x))\|_F \le M_J := \frac{C_{Gxy}}{\mu_G}$.*

*Proof.* By Lemma 2.1 of Ghadimi & Wang (2018), we have that

$$J(y(x)) = - \left( \nabla_{yy}^2 G(x, y(x)) \right)^{-1} \nabla_{xy}^2 G(x, y(x)).$$

Thank to Lemma 1 and Assumption 2, it is easy to see that $\nabla_{xy}^2 G(x, y(x))$ is Lipschitz continuous with modulus $L_{G_x^x} + L_{G_x^y} L_y$. Also, thanks to Lemma 1 and Assumption 2, we know that $\nabla_{yy}^2 G(x, y(x))$ is Lipschitz continuous with $L_{G_y^x} + L_{G_y^y} L_y$. Thus, for any $x_1$ and $x_2$, it holds that

$$
\begin{aligned}
&\|J(y(x_1)) - J(y(x_2))\| \\
&\le \| \left( \nabla_{yy}^2 G(x_1, y(x_1)) \right)^{-1} \nabla_{xy}^2 G(x_1, y(x_1)) - \left( \nabla_{yy}^2 G(x_1, y(x_1)) \right)^{-1} \nabla_{xy}^2 G(x_2, y(x_2)) \| \\
&+ \| \left( \nabla_{yy}^2 G(x_1, y(x_1)) \right)^{-1} \nabla_{xy}^2 G(x_2, y(x_2)) - \left( \nabla_{yy}^2 G(x_2, y(x_2)) \right)^{-1} \nabla_{xy}^2 G(x_2, y(x_2)) \| \\
&= \| \left( \nabla_{yy}^2 G(x_1, y(x_1)) \right)^{-1} \left( \nabla_{xy}^2 G(x_1, y(x_1)) - \nabla_{xy}^2 G(x_2, y(x_2)) \right) \| \\
&+ \| \left( \left( \nabla_{yy}^2 G(x_1, y(x_1)) \right)^{-1} - \left( \nabla_{yy}^2 G(x_2, y(x_2)) \right)^{-1} \right) \nabla_{xy}^2 G(x_2, y(x_2)) \| \\
&= \| \left( \nabla_{yy}^2 G(x_1, y(x_1)) \right)^{-1} \left( \nabla_{xy}^2 G(x_1, y(x_1)) - \nabla_{xy}^2 G(x_2, y(x_2)) \right) \| + \| \Gamma \nabla_{xy}^2 G(x_2, y(x_2)) \| \\
&\le \mu_G^{-1} \left( L_{G_x^x} + L_{G_x^y} L_y \right) \|x_1 - x_2\| + \mu_G^{-2} \left( L_{G_y^x} + L_{G_y^y} L_y \right) C_{Gxy} \|x_1 - x_2\|.
\end{aligned}
$$

where $\Gamma := \left( \nabla_{yy}^2 G(x_1, y(x_1)) \right)^{-1} \left( \nabla_{yy}^2 G(x_2, y(x_2)) - \nabla_{yy}^2 G(x_1, y(x_1)) \right) \left( \nabla_{yy}^2 G(x_2, y(x_2)) \right)^{-1}$. These together with Assumption 2 (ii) and (iii) gives the Lipschitz continuity of $J(y(x))$.

On the other hand, using Assumption 2 (i) and (ii), we have

$$\|J(y(x))\|_F \le \left\| \left( \nabla_{yy}^2 G(x, y(x)) \right)^{-1} \right\| \|\nabla_{xy}^2 G(x, y(x))\| \le \frac{C_{Gxy}}{\mu_G}.$$

$\square$

## B    Proofs for results in Section Convergence Analysis for Algorithm 2

### B.1    Proofs for Theorem 1

To prove Theorem 1, we give the following lemmas first.

**Lemma 3.** *Consider* (1) *and let Assumptions 1 and 2 hold. Let $\{(x^l, y^l, J^l)\}$ be generated by Algorithm 2. Then, it holds that*

$$
\begin{aligned}
&\|\nabla f(x^l) - \hat{\nabla} f(x^l)\|^2 \\
&\le \left( 2(L_{12}^F)^2 + 4 M_J^2 (L_{22}^F)^2 \right) \|y(x^l) - y^{l+1}\|^2 + 4(C_y^F)^2 \|J(y(x^l)) - J^{l+1}\|_F^2.
\end{aligned}
\tag{12}
$$

*Proof.* First, using the chain rule, we have that

$$\nabla f(x^l) = \nabla_x F(x^l, y(x^l)) + (J(y(x^l)))^T \nabla_y F(x^l, y(x^l)).$$

This together with the definition of $\hat{\nabla} f(x^l)$ in Algorithm 2, we have that

$$
\begin{aligned}
&\|\nabla f(x^l) - \hat{\nabla} f(x^l)\|^2 \\
&= \|\nabla_x F(x^l, y(x^l)) + (J(y(x^l)))^T \nabla_y F(x^l, y(x^l)) - \nabla_x F(x^l, y^{l+1}) - (J^{l+1})^T \nabla_y F(x^l, y^{l+1})\|^2 \\
&\leq 2\|\nabla_x F(x^l, y(x^l)) - \nabla_x F(x^l, y^{l+1})\|^2 \\
&\quad + 2\|(J(y(x^l)))^T \nabla_y F(x^l, y(x^l)) - (J^{l+1})^T \nabla_y F(x^l, y^{l+1})\|^2.
\end{aligned}
\tag{13}
$$

For the first term in the above relation, we have that

$$\|\nabla_x F(x^l, y(x^l)) - \nabla_x F(x^l, y^{l+1})\|^2 \leq (L_{12}^F)^2 \|y^{l+1} - y(x^l)\|^2.$$

For the second term in (13), it holds that

$$
\begin{aligned}
&\|J(y(x^l))^T \nabla_y F(x^l, y(x^l)) - (J^{l+1})^T \nabla_y F(x^l, y^{l+1})\|^2 \\
&\leq 2\|J(y(x^l))^T \nabla_y F(x^l, y(x^l)) - J(y(x^l))^T \nabla_y F(x^l, y^{l+1})\|^2 \\
&\quad + 2\|J(y(x^l))^T \nabla_y F(x^l, y^{l+1}) - (J^{l+1})^T \nabla_y F(x^l, y^{l+1})\|^2 \\
&= 2\|J(y(x^l))^T \left( \nabla_y F(x^l, y(x^l)) - \nabla_y F(x^l, y^{l+1}) \right)\|^2 \\
&\quad + 2\|(J(y(x^l)) - J^{l+1})^T \nabla_y F(x^l, y^{l+1})\|^2 \\
&\leq 2M_J^2 (L_{22}^F)^2 \|y(x^l) - y^{l+1}\|^2 + 2(C_y^F)^2 \|(J(y(x^l)) - J^{l+1})^T\|^2,
\end{aligned}
\tag{14}
$$

where the last inequality is thanks to Lemma 2 and Assumptions 1 and 2.

Combining the above inequality with (14) and (13), we obtain that

$$
\begin{aligned}
&\|\nabla f(x^l) - \hat{\nabla} f(x^l)\|^2 \\
&\leq 2(L_{12}^F)^2 \|y^{l+1} - y(x^l)\|^2 + 4M_J^2 (L_{22}^F)^2 \|y(x^l) - y^{l+1}\|^2 + 4(C_y^F)^2 \|(J(y(x^l)) - J^{l+1})^T\|^2 \\
&\leq \left( 2(L_{12}^F)^2 + 4M_J^2 (L_{22}^F)^2 \right) \|y(x^l) - y^{l+1}\|^2 + 4(C_y^F)^2 \|J(y(x^l)) - J^{l+1}\|^2.
\end{aligned}
$$

Thus, the conclusion follows from the fact that $\|A\| \leq \|A\|_F$ for any matrix $A$. $\qquad \square$

**Lemma 4.** *Consider* (1) *and let Assumptions 1 and 2 hold. Let $s > 0$. Let $\{(x^l, y^l, J^l)\}$ be generated by Algorithm 2 Then, it holds that*

$$
\begin{aligned}
\|(J(y(x^l)) - J^{l+1})^T\|_F^2 &\leq 6(1 + \frac{1}{s^2})\sqrt{n}\gamma^2 \left( L_{G_x^y}^2 + L_{G_y^y}^2 M_J^2 \right) \|y(x^{l-1}) - y^l\|^2 \\
&\quad + 3(1 + \frac{1}{s^2})\sqrt{n} \left( \left( 2\gamma^2 L_{G_x^y}^2 + 2\gamma^2 L_{G_y^y}^2 M_J^2 \right) L_y^2 + (1 - \gamma\mu_G)^2 L_J^2 \right) \|x^l - x^{l-1}\|^2 \\
&\quad + (1 + s^2)\sqrt{n}(1 - \gamma\mu_G)^2 \|J(y(x^{l-1})) - J^l\|_F^2.
\end{aligned}
\tag{15}
$$

*Proof.* Using Step 4 of Algorithm 2 and the fact that $J(y(x^l)) = J(y(x^l))(I - \gamma\nabla_{yy}G(x^l, y(x^l))) - \gamma\nabla_{xy}G(x^l, y(x^l))$, we have that

$$\|(J(y(x^l)) - J^{l+1})^T\|^2$$

$$\leq \|J(y(x^l))(I - \gamma\nabla_{yy}G(x^l, y(x^l))) - \gamma\nabla_{xy}G(x, y(x^l)) - J^l(I - \gamma\nabla_{yy}G(x^l, y^l)) + \gamma\nabla_{xy}G(x^l, y^l)\|^2$$

$$\overset{(a)}{\leq} 3(1 + \frac{1}{s^2})\gamma^2\|\nabla_{xy}G(x^l, y(x^l)) - \nabla_{xy}G(x^l, y^l)\|^2$$

$$+ 3(1 + \frac{1}{s^2})\|J(y(x^l))(\gamma\nabla_{yy}G(x^l, y^l) - \gamma\nabla_{yy}G(x^l, y(x^l)))\|^2$$

$$+ 3(1 + \frac{1}{s^2})\|\left(J(y(x^l)) - J(y(x^{l-1}))\right)(I - \gamma\nabla_{yy}G(x^l, y^l))\|^2$$

$$+ (1 + s^2)\|\left(J(y(x^{l-1})) - J^l\right)(I - \gamma\nabla_{yy}G(x^l, y^l))\|^2$$

$$\overset{(b)}{\leq} 3(1 + \frac{1}{s^2})\gamma^2 L_{G_x^y}^2\|y(x^l) - y^l\|^2 + 3(1 + \frac{1}{s^2})\gamma^2 L_{G_y^y}^2 M_J^2\|y^l - y(x^l)\|^2$$

$$+ 3(1 + \frac{1}{s^2})L_J^2\|x^l - x^{l-1}\|^2\|(I - \gamma\nabla_{yy}G(x^l, y^l))\|^2$$

$$+ (1 + s^2)\|\left(J(y(x^{l-1})) - J^l\right)(I - \gamma\nabla_{yy}G(x^l, y^l))\|^2,$$

where (a) uses the fact that $\|a_1 + a_2 + a_3 + a_4\|^2 \leq (1 + \frac{1}{s^2})\|a_1 + a_2 + a_3\|^2 + (1 + s^2)\|a_4\|^2 \leq 3(1 + \frac{1}{s^2})(\|a_1\|^2 + \|a_2\|^2 + \|a_3\|^2) + (1 + s^2)\|a_4\|^2$ for any matrices $\{a_1, a_2, a_3, a_4\}$, (b) uses Assumption 2 and Lemma 2. Using Assumption 2, the above inequality can be further passed to

$$\|(J(y(x^l)) - J^{l+1})^T\|^2 \leq 3(1 + \frac{1}{s^2})\gamma^2 L_{G_x^y}^2\|y(x^l) - y^l\|^2 + 3(1 + \frac{1}{s^2})\gamma^2 L_{G_y^y}^2 M_J^2\|y^l - y(x^l)\|^2$$

$$+ 3(1 + \frac{1}{s^2})(1 - \gamma\mu_G)^2 L_J^2\|x^l - x^{l-1}\|^2 + (1 + s^2)(1 - \gamma\mu_G)^2\|J(y(x^{l-1})) - J^l\|^2$$

$$\leq 3(1 + \frac{1}{s^2})\left(2\gamma^2 L_{G_x^y}^2 + 2\gamma^2 L_{G_y^y}^2 M_J^2\right)\|y(x^{l-1}) - y^l\|^2$$

$$+ 3(1 + \frac{1}{s^2})\left(2\gamma^2 L_{G_x^y}^2 + 2\gamma^2 L_{G_y^y}^2 M_J^2\right)L_y^2\|x^l - x^{l-1}\|^2$$

$$+ 3(1 + \frac{1}{s^2})(1 - \gamma\mu_G)^2 L_J^2\|x^l - x^{l-1}\|^2 + (1 + s^2)(1 - \gamma\mu_G)^2\|J(y(x^{l-1})) - J^l\|^2$$

$$= 6(1 + \frac{1}{s^2})\gamma^2\left(L_{G_x^y}^2 + L_{G_y^y}^2 M_J^2\right)\|y(x^{l-1}) - y^l\|^2$$

$$+ 3(1 + \frac{1}{s^2})\left(\left(2\gamma^2 L_{G_x^y}^2 + 2\gamma^2 L_{G_y^y}^2 M_J^2\right)L_y^2 + (1 - \gamma\mu_G)^2 L_J^2\right)\|x^l - x^{l-1}\|^2$$

$$+ (1 + s^2)(1 - \gamma\mu_G)^2\|J(y(x^{l-1})) - J^l\|^2,$$

where the last inequality uses Lemma 1. Then the conclusion follows from the fact that $\|A\|_F \leq \|A\| \leq \|A\|_F$ for any matrix $A$. $\square$

Now we are ready to present the detailed version of Theorem 1.

**Theorem 4.** *Consider* (1) *and let Assumptions 1 and 2 hold. Suppose that $F$ is bounded from below. Let $\{(x^l, y^l, J^l)\}$ be generated by Algorithm 2. Let $\alpha \in (0, \frac{1}{\mu_G})$. Denote $\tilde{\zeta} := \frac{2\mu_G}{\alpha^{-1} + \mu_G}$. Let $d_y > 0$ and $d_y^2 \in (0, \frac{1}{1-\tilde{\zeta}} - 1)$. Denote $\zeta := (1 + d_y^2)(1 - \tilde{\zeta})$. Let $s > 0$ and satisfies $s^2 < \frac{1}{\sqrt{n}(1 - \gamma\mu_g)} - 1$. Let $r > 0$ and satisfies $r > \max\left\{\frac{8(C_y^F)^2}{1 - (1+s^2)\sqrt{n}(1-\gamma\mu_G)^2}, \frac{4(L_{12}^F)^2 + 8M_J^2(L_{22}^F)^2}{\zeta - 6(1+s^2)\sqrt{n}\left(L_{G_x^y}^2 + L_{G_y^y}^2\frac{C_{Gxy}^2}{\mu_g^2}\right)}\right\}$. Let $(1 - \frac{1}{n})\frac{1}{\mu_G} < \gamma <$*

$$\min\left\{\sqrt{\frac{\zeta}{6(1+s^2)\sqrt{n}\left(L_{G_x^y}^2 + L_{G_y^y}^2\frac{C_{Gxy}^2}{\mu_g^2}\right)}}, \frac{1}{\mu_G}\right\}. \text{ Let } \delta > 0 \text{ and satisfies}$$

$$\delta < 1 - \frac{4(L_{12}^F)^2 + 8M_J^2(L_{22}^F)^2}{r} - (1 - \zeta) - 6(1 + \frac{1}{s^2})\sqrt{n}\gamma^2\left(L_{G_x^y}^2 + L_{G_y^y}^2 M_J^2\right) \tag{16}$$

*and*

$$\delta < 1 - \frac{8(C_y^F)^2}{r} - (1 + s^2)\sqrt{n}(1 - \gamma\mu_G)^2. \tag{17}$$

*Denote*

$$C_2 := (1 + d_y^{-2})(1 - \tilde{\zeta})L_y^2 + 3(1 + \frac{1}{s^2})\sqrt{n}\left(\left(2\gamma^2 L_{G_x^y}^2 + 2\gamma^2 L_{G_y^y}^2 M_J^2\right)L_y^2 + (1 - \gamma\mu_G)^2 L_J^2\right).$$

*Let $\Delta > 0$ and suppose $\beta$ is small enough such that*

$$\frac{L_f + 4r - 2\beta^{-1}}{2} + C_2 < -\Delta. \tag{18}$$

*Denote*

$$H(x, x', y, J) := f(x) + \Delta\|x - x'\|^2 + \|y(x) - y\|^2 + \|(J(y(x)) - J\|_F^2.$$

*Then $H^l := H(x^l, x^{l-1}, y^{l+1}, J^{l+1})$ is nonincreasing and*

$$H^{l+1} \le H^l - \Delta\|x^l - x^{l-1}\|^2 - \delta\|y(x^l) - y^{l+1}\|^2 - \delta\|J(y(x^l)) - J^{l+1}\|_F^2. \tag{19}$$

*Furthermore, there exists $H^*$ such that $\{H^{l+1}\}$ is convergent to $H^*$.*

*Proof.* Using Lemma 1, we have that

$$
\begin{aligned}
f(x^{l+1}) &\le f(x^l) + \left\langle \nabla f(x^l), x^{l+1} - x^l \right\rangle + \frac{L_f}{2}\|x^{l+1} - x^l\|^2 \\
&= f(x^l) + \left\langle \hat{\nabla} f(x^l), x^{l+1} - x^l \right\rangle + \frac{1}{2\beta}\|x^{l+1} - x^l\|^2 \\
&\quad + \left\langle \nabla f(x^l) - \hat{\nabla} f(x^l), x^{l+1} - x^l \right\rangle + \frac{L_f - \beta^{-1}}{2}\|x^{l+1} - x^l\|^2 \\
&\overset{(a)}{\le} f(x^l) - \frac{1}{2\beta}\|x^{l+1} - x^l\|^2 + \left\langle \nabla f(x^l) - \hat{\nabla} f(x^l), x^{l+1} - x^l \right\rangle + \frac{L_f - \beta^{-1}}{2}\|x^{l+1} - x^l\|^2 \\
&= f(x^l) + \left\langle \nabla f(x^l) - \hat{\nabla} f(x^l), x^{l+1} - x^l \right\rangle + \frac{L_f - 2\beta^{-1}}{2}\|x^{l+1} - x^l\|^2 \\
&\le f(x^l) + \frac{2}{r}\|\nabla f(x^l) - \hat{\nabla} f(x^l)\|^2 + \frac{L_f + 4r - 2\beta^{-1}}{2}\|x^{l+1} - x^l\|^2.
\end{aligned}
$$

where (a) is because $x^{l+1}$ is minimizer of $\min_x \left\langle \hat{\nabla} f(x^l), x - x^l \right\rangle + \frac{1}{2\beta}\|x - x^l\|^2$ whose objective is strongly convex. Using (12), the above inequality can be further passed to

$$
\begin{aligned}
f(x^{l+1}) &\le f(x^l) + \frac{L_f + 4r - 2\beta^{-1}}{2}\|x^{l+1} - x^l\|^2 \\
&\quad + \frac{4(L_{12}^F)^2 + 8M_J^2(L_{22}^F)^2}{r}\|y(x^l) - y^{l+1}\|^2 + \frac{8(C_y^F)^2}{r}\|J(y(x^l)) - J^{l+1}\|_F^2.
\end{aligned} \tag{20}
$$

On the other hand, recall that $\tilde{\zeta} = \frac{2\mu_G}{\alpha^{-1} + \mu_G}$. Thanks to the assumption that $\alpha < \frac{1}{\mu_G}$, we have that $\tilde{\zeta} \in (0, 1)$. Thanks to Assumption 2 (i) and (ii), it is easy to show (see Theorem 29 of Beck (2017)) that for $l \ge 0$,

$$
\begin{aligned}
\|y(x^l) - y^{l+1}\|^2 &\le (1 - \tilde{\zeta})\|y(x^l) - y^l\|^2 \\
&\le (1 + d_y^2)(1 - \tilde{\zeta})\|y(x^{l-1}) - y^l\|^2 + (1 + d_y^{-2})(1 - \tilde{\zeta})\|y(x^l) - y(x^{l-1})\|^2 \\
&\le (1 + d_y^2)(1 - \tilde{\zeta})\|y(x^{l-1}) - y^l\|^2 + (1 + d_y^{-2})(1 - \tilde{\zeta})L_y^2\|x^l - x^{l-1}\|^2 \\
&\le (1 - \zeta)\|y(x^{l-1}) - y^l\|^2 + (1 + d_y^{-2})(1 - \tilde{\zeta})L_y^2\|x^l - x^{l-1}\|^2,
\end{aligned} \tag{21}
$$

where the third inequality follows from Lemma 1 and the last inequality is because the definition of $\zeta$. Thanks to the assumption that $d_y^2 \in (0, \frac{1}{1-\tilde{\zeta}} - 1)$, $\zeta \in (0, 1)$.

Summing (20), (15) with $l = l+1$ and (21) with $l = l+1$, we have that

$$
\begin{aligned}
& f(x^{l+1}) + \|y(x^{l+1}) - y^{l+2}\|^2 + \|J(y(x^{l+1})) - J^{l+2}\|_F^2 \\
& \leq f(x^l) + \frac{L_f + 4r - 2\beta^{-1}}{2}\|x^{l+1} - x^l\|^2 + \frac{4(L_{12}^F)^2 + 8M_J^2(L_{22}^F)^2}{r}\|y(x^l) - y^{l+1}\|^2 \\
& + \frac{8(C_y^F)^2}{r}\|J(y(x^l)) - J^{l+1}\|_F^2 + (1-\zeta)\|y(x^l) - y^{l+1}\|^2 + (1+d_y^{-2})(1-\tilde{\zeta})L_y^2\|x^{l+1} - x^l\|^2 \\
& + 6(1 + \frac{1}{s^2})\sqrt{n}\gamma^2\left(L_{G_x^y}^2 + L_{G_y^y}^2 M_J^2\right)\|y(x^l) - y^{l+1}\|^2 \\
& + 3(1 + \frac{1}{s^2})\sqrt{n}\left(\left(2\gamma^2 L_{G_x^y}^2 + 2\gamma^2 L_{G_y^y}^2 M_J^2\right)L_y^2 + (1-\gamma\mu_G)^2 L_J^2\right)\|x^{l+1} - x^l\|^2 \\
& + (1 + s^2)\sqrt{n}(1-\gamma\mu_G)^2\|J(y(x^l)) - J^{l+1}\|_F^2 \\
& = f(x^l) + \left(\frac{L_f + 4r - 2\beta^{-1}}{2} + C_2\right)\|x^{l+1} - x^l\|^2 + \left(\frac{8(C_y^F)^2}{r} + (1+s^2)\sqrt{n}(1-\gamma\mu_G)^2\right)\|J(y(x^l)) - J^{l+1}\|_F^2 \\
& + \left(\frac{4(L_{12}^F)^2 + 8M_J^2(L_{22}^F)^2}{r} + (1-\zeta) + 6(1 + \frac{1}{s^2})\sqrt{n}\gamma^2\left(L_{G_x^y}^2 + L_{G_y^y}^2 M_J^2\right)\right)\|y(x^l) - y^{l+1}\|^2,
\end{aligned}
\tag{22}
$$

where $C_2 = (1+d_y^{-2})(1-\tilde{\zeta})L_y^2 + 3(1 + \frac{1}{s^2})\sqrt{n}\left(\left(2\gamma^2 L_{G_x^y}^2 + 2\gamma^2 L_{G_y^y}^2 M_J^2\right)L_y^2 + (1-\gamma\mu_G)^2 L_J^2\right)$.

On the other hand, thanks to the assumption on $\gamma$, we have that $\zeta - 6(1+s^2)\sqrt{n}\gamma^2\left(L_{G_x^y}^2 + L_{G_y^y}^2\frac{C_{G_{xy}}^2}{\mu_g^2}\right) > 0$. Using the assumption on $r$, we know that

$$
0 < \frac{4(L_{12}^F)^2 + 8M_J^2(L_{22}^F)^2}{r} + (1-\zeta) + 6(1 + \frac{1}{s^2})\sqrt{n}\gamma^2\left(L_{G_x^y}^2 + L_{G_y^y}^2 M_J^2\right) < 1.
$$

In addition, thanks to assumption on $s$ and $\gamma$, we have $0 < (1+s^2)\sqrt{n}(1-\gamma\mu_G)^2 < 1$. Using the assumption on $r$, we know

$$
0 < \frac{8(C_y^F)^2}{r} + (1+s^2)\sqrt{n}(1-\gamma\mu_G)^2 < 1.
$$

Therefore, there exists $\delta \in (0, 1)$ such that (16) and (17) holds.

Thus, using (16) and (17), (22) can be further passed to

$$
\begin{aligned}
& f(x^{l+1}) + \|y(x^{l+1}) - y^{l+2}\|^2 + \|(J(y(x^{l+1})) - J^{l+2}\|_F^2 \\
& \leq f(x^l) + \left(\frac{L_f + 4r - 2\beta^{-1}}{2} + C_2\right)\|x^{l+1} - x^l\|^2 + \|y(x^l) - y^{l+1}\|^2 + \|J(y(x^l)) - J^{l+1}\|_F^2 \\
& - \delta\|y(x^l) - y^{l+1}\|^2 - \delta\|J(y(x^l)) - J^{l+1}\|_F^2.
\end{aligned}
\tag{23}
$$

Using (18), the inequality (23) can be further passed to

$$
\begin{aligned}
& f(x^{l+1}) + \|y(x^{l+1}) - y^{l+2}\|^2 + \|J(y(x^{l+1}) - J^{l+2}\|_F^2 \\
& \leq f(x^l) - \Delta\|x^{l+1} - x^l\|^2 + \|y(x^l) - y^{l+1}\|^2 + \|J(y(x^l)) - J^{l+1}\|_F^2 \\
& = f(x^l) - \Delta\|x^{l+1} - x^l\|^2 + \Delta\|x^l - x^{l-1}\|^2 + \|y(x^l) - y^{l+1}\|^2 + \|J(y(x^l)) - J^{l+1}\|_F^2 \\
& - \Delta\|x^l - x^{l-1}\|^2 - \delta\|y(x^l) - y^{l+1}\|^2 - \delta\|J(y(x^l)) - J^{l+1}\|_F^2.
\end{aligned}
$$

Rearranging the above inequality we obtain

$$
\begin{aligned}
H^{l+1} & = f(x^{l+1}) + \Delta\|x^{l+1} - x^l\|^2 + \|y(x^{l+1}) - y^{l+2}\|^2 + \|J(y(x^{l+1})) - (J^{l+2})^T\|^2 \\
& \leq f(x^l) + \Delta\|x^l - x^{l-1}\|^2 + \|y(x^l) - y^{l+1}\|^2 + \|J(y(x^l)) - J^{l+1}\|_F^2 \\
& - \Delta\|x^l - x^{l-1}\|^2 - \delta\|y(x^l) - y^{l+1}\|^2 - \delta\|J(y(x^l)) - J^{l+1}\|_F^2 \\
& = H^l - \Delta\|x^l - x^{l-1}\|^2 - \delta\|y(x^l) - y^{l+1}\|^2 - \delta\|J(y(x^l)) - J^{l+1}\|_F^2.
\end{aligned}
$$

Since $H^l$ is nonincreasing, $F$ is bounded from below and $\infty < F(x^l, y(x^l)) < H^k$, we know that $H^k$ is convergent.

$\square$

## B.2 Properties of the limits of the generated sequences

The following corollary states the detailed version of Corollary 1.

**Corollary 2.** *Let assumptions in Theorem 1 hold and suppose $F$ is level-bounded. Then:*

(i) $\lim_{l\to\infty} \|x^{l+1} - x^l\| = \lim_{l\to\infty} \|y(x^l) - y^{l+1}\| = \lim_{l\to\infty} \|J(y(x^l)) - J^{l+1}\|_F = 0$. *In addition, the sequence $\{(x^l, y^l, J^{l+1})\}$ is bounded.*

(ii) *Any accumulation point $x^*$ of $\{x^l\}$ satisfies $\nabla f(x^*) = 0$. In addition, any accumulation point $y^*$ of $\{y^l\}$ is an optimal solution of the lower level problem in* (1) *defined with $x^*$.*

*Proof.* Noting that $\{H(x^{l+1}, x^l, y^{l+2}, J^{l+2})\}$ is nonincreasing, we know that

$$F(x^{l+1}, y(x^{l+1})) = f(x^{l+1}) \leq H^l \leq H^1 < \infty.$$

Since $F$ is level-bounded, we have that $\{(x^{l+1}, y(x^{l+1}))\}$ is bounded.

Summing (19) from $l = 0$ to $N$, we have that

$$H^{N+1} \leq H^0 - \Delta \sum_{l=0}^{N+1} \|x^{l+1} - x^l\|^2 - \delta \sum_{l=0}^{N+1} \|y(x^l) - y^{l+1}\|^2 - \delta \sum_{l=0}^{N+1} \|J(y(x^l)) - J^{l+1}\|_F^2.$$

Rearranging the above inequality, we have that

$$\Delta \sum_{l=0}^{N+1} \|x^{l+1} - x^l\|^2 + \delta \sum_{l=0}^{N+1} \|y(x^l) - y^{l+1}\|^2 + \delta \sum_{l=0}^{N+1} \|J(y(x^l)) - J^{l+1}\|_F^2 \leq H^0 - H^{N+1}$$
$$\leq H^0 - \lim_{l\to\infty} H^l < \infty.$$

Taking the $N$ in the above inequality to $\infty$, we have that $\lim_{l\to\infty} \|x^{l+1} - x^l\| = 0$, $\lim_{l\to\infty} \|y(x^l) - y^{l+1}\| = 0$ and $\lim_{l\to\infty} \|J(y(x^l)) - J^{l+1}\|_F = 0$. This together with the boundedness of $\{(x^{l+1}, y(x^{l+1}))\}$ and the continuity of $y(x)$ and $J(y(x))$ w.r.t $x$ guaranteed by Lemma 1 and Lemma 2, we have that $\{y^l\}$ and $\{J^{l+1}\}$ are bounded.

For (ii), since $x^{l+1}$ is the minimizer of $\min_x \left\langle \hat\nabla f(x^l), x - x^l \right\rangle + \frac{1}{2\beta}\|x - x^l\|^2$, it holds that

$$0 \in \beta\hat\nabla f(x^l) + (x^{l+1} - x^l) \Leftrightarrow \nabla f(x^l) - \hat\nabla f(x^l) - \frac{1}{\beta}(x^{l+1} - x^l) \in \nabla f(x^l).$$

Thus, using (12), we have that

$$\|\nabla f(x^l)\| \leq \|\nabla f(x^l) - \hat\nabla f(x^l)\| + \frac{1}{\beta}\|x^{l+1} - x^l\|$$

$$\leq \sqrt{\left(2(L_{12}^F)^2 + 4M_J^2(L_{22}^F)^2\right)\|y(x^l) - y^{l+1}\|^2 + 4(C_y^F)^2\|J(y(x^l)) - J^{l+1}\|_F^2} + \frac{1}{\beta}\|x^{l+1} - x^l\| \quad (24)$$

$$\leq \sqrt{2(L_{12}^F)^2 + 4M_J^2(L_{22}^F)^2}\|y(x^l) - y^{l+1}\| + 2C_y^F\|J(y(x^l)) - J^{l+1}\|_F + \frac{1}{\beta}\|x^{l+1} - x^l\|,$$

where the second inequality uses (12).

Now let $\{(x^{l_j}, y^{l_j})\}$ be the subsequence of $\{(x^l, y^l)\}$ such that $\lim_j (x^{l_j}, y^{l_j}) = (x^*, y^*)$. Using Lemma 1, we have that,

$$\|\nabla f(x^*)\| = \lim_j \|\nabla f(x^{l_j})\| \leq \lim_j \sqrt{2(L_{12}^F)^2 + 4M_J^2(L_{22}^F)^2}\|y(x^{l_j}) - y^{l_j+1}\|$$

$$+ 2C_y^F\|J(y(x^{l_j})) - J^{l_j+1}\|_F + \frac{1}{\beta}\|x^{l_j+1} - x^{l_j}\| = 0,$$

where the last equality uses (i). Thus $\nabla f(x^*) = 0$.

Finally, let $y^*$ be an accumulation point of $\{y^l\}$. Then, we have

$$\|y^* - y(x^*)\| \leq \lim_l \|y^* - y^{l+1}\| + \|y^{l+1} - y(x^l)\| + \|y(x^l) - y(x^*)\|$$

$$\leq \lim_l \|y^* - y^{l+1}\| + \|y^{l+1} - y(x^l)\| + L_y\|x^l - x^*\| = 0$$

where the second inequality use Lemma 1 and the last equality uses (i). Therefore, we have $y^*$ is a solution of the lower level problem defined by $x^*$. □

## B.3 Proofs of Theorem 2

To prove Theorem 2, we first prove the following lemma.

**Lemma 5.** *Let assumptions in Corollary 2 hold.*

(i) *Denote the set of accumulation points of $\{(x^l, x^{l-1}, y^{l+1}, J^{l+1})\}$ as $\Omega$. Then $\Omega$ is bounded and $H$ is constant on $\Omega$.*

(ii) *It holds that*

$$\text{dist}(0, \partial H(x^{l+1}, x^l, y^{l+2}, J^{l+2})) \leq \left(\sqrt{2(L_{12}^F)^2 + 4L_y^2(L_{22}^F)^2} + 2(L_y+1)\right)\|y(x^{l+1}) - y^{l+2}\|$$

$$+ (2L_F + 2(L_J+1))\|J(y(x^{l+1})) - J^{l+2}\|_F + \frac{1}{\beta}\|x^{l+2} - x^{l+1}\| + \sqrt{2}\Delta\|x^{l+1} - x^l\|. \tag{25}$$

*Proof.* First, the boundedness of $\Omega$ follows from Corollary 2. Now, let $(x^*, x^{**}, y^*, y^{**}, J^*)$ be any point in $\Omega$. Then there exists a sequence $\{(x^{l_j}, x^{l_j-1}, y^{l_j+1}, J^{l_j+1})\}_j$ such that $\lim_j(x^{l_j}, x^{l_j-1}, y^{l_j+1}, J^{l_j+1}) = (x^*, x^{**}, y^{**}, J^*)$. Thanks to Lemmas 1 and Lemma 2, we have that $H$ is continuous. Thus,

$$H(x^*, x^{**}, y^{**}, J^*) = \lim_j H(x^{l_j}, x^{l_j-1}, y^{l_j+1}, J^{l_j+1}) = H^*,$$

where the last inequality uses Theorem 4.

Now we prove (ii). Denoting $Z := (x, x', y, J)$ and using Corollary 10.9 of Rockafellar & Wets (1998), we have that

$$\partial_Z H(x, x', y, J) \supseteq \hat{\partial}_Z H(x, x', y, J)$$

$$= \hat{\partial}_Z f(x) + \hat{\partial}_Z \frac{\Delta}{2}\|x - x'\|^2 + \hat{\partial}_Z\|y(x) - y\|^2 + \hat{\partial}_Z\|J(y(x)) - J\|_F^2$$

$$= \begin{bmatrix} \nabla f(x) \\ 0 \\ 0 \\ 0 \end{bmatrix} + \begin{bmatrix} \Delta(x - x') \\ -\Delta(x - x') \\ 0 \\ 0 \end{bmatrix} + \begin{bmatrix} u_1 \\ 0 \\ u_2 \\ 0 \end{bmatrix} + \begin{bmatrix} u_3 \\ 0 \\ 0 \\ u_4 \end{bmatrix}, \tag{26}$$

where $(u_1, u_2) \in \hat{\partial}_{(u_1, u_2)}\|y(x) - y\|^2$ and $(u_3, u_4) \in \hat{\partial}_{(u_3, u_4)}\|J(y(x)) - J\|_F^2$ and the second equality uses Proposition 10.5 Rockafellar & Wets (1998).

Using the definition of regular subgradient, we have that

$$
\begin{aligned}
0 \leq\ & \liminf_{(x',y')\neq(x,y),(x',y')\to(x,y)} \frac{\|y(x')-y'\|^2 - \|y(x)-y\|^2 - \langle(u_1,u_2),(x',y')-(x,y)\rangle}{\|(x',y')-(x,y)\|} \\
=\ & \liminf_{(x',y')\neq(x,y),(x',y')\to(x,y)} \frac{(\|y(x')-y'\|+\|y(x)-y\|)(\|y(x')-y'\|-\|y(x)-y\|)}{\|(x',y')-(x,y)\|} \\
& - \limsup_{(x',y')\neq(x,y),(x',y')\to(x,y)} \frac{\langle(u_1,u_2),(x',y')-(x,y)\rangle}{\|(x',y')-(x,y)\|} \\
\leq\ & \liminf_{(x',y')\neq(x,y),(x',y')\to(x,y)} \frac{(\|y(x')-y'\|+\|y(x)-y\|)(\|y(x')-y'-y(x)+y\|)}{\|(x',y')-(x,y)\|} \\
& - \limsup_{(x',y')\neq(x,y),(x',y')\to(x,y)} \frac{\langle(u_1,u_2),(x',y')-(x,y)\rangle}{\|(x',y')-(x,y)\|} \\
\leq\ & \liminf_{(x',y')\neq(x,y),(x',y')\to(x,y)} \frac{(\|y(x')-y'\|+\|y(x)-y\|)(\|y(x')-y(x)\|+\|y-y'\|)}{\|(x',y')-(x,y)\|} \\
& - \limsup_{(x',y')\neq(x,y),(x',y')\to(x,y)} \frac{\langle(u_1,u_2),(x',y')-(x,y)\rangle}{\|(x',y')-(x,y)\|} \\
\leq\ & \liminf_{(x',y')\neq(x,y),(x',y')\to(x,y)} (\|y(x')-y'\|+\|y(x)-y\|)\left(\frac{\|y(x')-y(x)\|}{\|x'-x\|}+\frac{\|y-y'\|}{\|y'-y\|}\right) \\
& - \limsup_{(x',y')\neq(x,y),(x',y')\to(x,y)} \frac{\langle(u_1,u_2),(x',y')-(x,y)\rangle}{\|(x',y')-(x,y)\|} \\
\leq\ & \liminf_{(x',y')\neq(x,y),(x',y')\to(x,y)} (\|y(x')-y'\|+\|y(x)-y\|)(L_y+1) - \frac{\langle(u_1,u_2),(x',y')-(x,y)\rangle}{\|(x',y')-(x,y)\|}
\end{aligned}
$$

This together with Lemma 1 gives

$$
0 \leq 2(L_y+1)\|y(x)-y\| - \limsup_{(x',y')\neq(x,y),(x',y')\to(x,y)} \frac{\langle(u_1,u_2),(x',y')-(x,y)\rangle}{\|(x',y')-(x,y)\|},
$$

Thus, we deduce that $\|(u_1,u_2)\| \leq 2(L_y+1)\|y(x)-y\|$. Similarly, we have that $\|(u_3,u_4)\| \leq 2(L_J+1)\|J(y(x))-J\|_F$. Combining two facts with (26), we have that

$$
\begin{aligned}
& \mathrm{dist}(0,\partial H(x^{l+1},x^l,y^{l+2},J^{l+2})) \\
& \leq \|\nabla f(x^{l+1})\| + \sqrt{2}\Delta\|x^{l+1}-x^l\| + 2(L_y+1)\|y(x^{l+1})-y^{l+2}\| \\
& \quad + 2(L_J+1)\|J(y(x^{l+1}))-J^{l+2}\|_F \\
& \leq \sqrt{(L_{12}^F)^2 + 4L_y^2(L_{22}^F)^2}\|y(x^{l+1})-y^{l+2}\|^2 + 2C_y^F\|J(y(x^{l+1}))-J^{l+2}\|_F^2 + \frac{1}{\beta}\|x^{l+2}-x^{l+1}\| \\
& \quad + \sqrt{2}\Delta\|x^{l+1}-x^l\| + 2(L_y+1)\|y(x^{l+1})-y^{l+2}\| + 2(L_J+1)\|J(y(x^{l+1}))-J^{l+2}\|_F \\
& = \left(\sqrt{2(L_{12}^F)^2 + 4L_y^2(L_{22}^F)^2} + 2(L_y+1)\right)\|y(x^{l+1})-y^{l+2}\| \\
& \quad + (2C_y^F + 2(L_J+1))\|J(y(x^{l+1}))-J^{l+2}\|_F + \frac{1}{\beta}\|x^{l+2}-x^{l+1}\| + \sqrt{2}\Delta\|x^{l+1}-x^l\|,
\end{aligned}
$$

where the second inequality uses (24). $\qquad\square$

Now we are ready to prove Theorem 2.

*Proof.* Denote $Z^l = (x^l,x^{l-1},y^{l+1},J^{l+1})$. Note that to show the convergence of $\{(x^l,y^l,J^l)\}$, it is sufficient to show the same convergence of $\{Z^l\}$.

We first show that $\{Z^l\}$ is convergent when $H$ is KL. Suppose there exists $\bar{l}\in\mathbb{N}_+$ such that $H^l = H^{\bar{l}}$. Then thanks to Theorem 4, we know that $H^l = H^*$, $\|x^{l+1}-x^l\| = \|y(x^l)-y^{l+1}\| = \|J(y(x^l))-J^{l+1}\| = 0$ for all

$l \geq \bar{l}$. This implies that $\{Z^{l+1}\}$ converges finitely to $(x^{\bar{l}}, x^{\bar{l}}, y(x^{\bar{l}}), J(x^{\bar{l}}))$. Thus, in the rest of the proof, we only consider the case where $H^l > H^*$ for all $l$.

Since $H$ is a KL function and thanks to Lemma 5, $H$ is constant on $\Omega$, using Lemma 6 of Bolte et al. (2014), there exist $\epsilon > 0$ and a function $\phi : [0, a) \to [0, \infty)$ with $\phi(0) = 0$ such that

$$\phi'(H(Z) - H^*)\mathrm{dist}(0, \partial H(Z)) \geq 1$$

when $Z \in \{Z : \mathrm{dist}(Z, \Omega) < \epsilon\} \cap \{Z : H^* < H(Z) < H^* + \varepsilon\}$. Since $\Omega$ is the set of accumulation points of $Z^l$, we know that there exists $l_1$ such that $\mathrm{dist}(Z^l, \Omega) < \epsilon$ when $l \geq l_1$. In addition, using Theorem 4, we know that there exists $l_2$ such that $H(z^l) < H^* + \varepsilon$ when $l \geq l_2$. Thus, when $l \geq \max\{l_1, l_2\}$, we have that

$$\phi'(H(Z^l) - H^*)\mathrm{dist}(0, \partial H(Z^l)) \geq 1. \tag{27}$$

Using the concavity of $\phi$, it holds that

$$\begin{aligned}
&\left(\phi(H^l - H^*) - \phi(H^{l+1} - H^*)\right)\mathrm{dist}(0, \partial H(Z^l)) \\
&\geq \phi'(H^l - H^*)\mathrm{dist}(0, \partial H(Z^l))\left(H^l - H^{l+1}\right) \\
&\geq H^l - H^{l+1},
\end{aligned} \tag{28}$$

where the last inequality uses (27) and Theorem 4. Now, we denote $D_1^l = \|x^l - x^{l-1}\|$, $D_2^l = \|y(x^l) - y^{l+1}\|$, $D_3^l = \|J(y(x^l)) - J^{l+1}\|_F$, $A^l = D_1^l + D_2^l + D_3^l$. Combining these definitions with (28), (19) and (25), we obtain that

$$\begin{aligned}
\min\{\Delta, \delta\}(A^l)^2 &\leq 3\min\{\Delta, \delta\}((D_1^l)^2 + (D_2^l)^2 + (D_3^l)^2) \\
&\leq 3\Delta\|x^l - x^{l-1}\|^2 + 3\delta\|y(x^l) - y^{l+1}\|^2 + 3\delta\|J(y(x^l)) - J^{l+1}\|_F^2 \\
&\overset{(19)}{\leq} H^l - H^{l+1} \overset{(28)}{\leq} \phi\left((H^l - H^*) - (H^{l+1} - H^*)\right)\mathrm{dist}(0, \partial H(Z^l)) \\
&\overset{(25)}{\leq} \phi\left((H^l - H^*) - (H^{l+1} - H^*)\right)\left(C_1 D_2^{l+1} + \tilde{C}_1 D_3^{l+1} + \frac{1}{\beta}D_1^{l+1} + \sqrt{2}\Delta D_1^l\right) \\
&\leq \max\{C_1, \tilde{C}_1, \frac{1}{\beta}, \sqrt{2}\Delta\}\phi\left((H^l - H^*) - (H^{l+1} - H^*)\right)\left(A^{l+1} + A^l\right)
\end{aligned}$$

where $C_1 := \sqrt{2(L_{12}^F)^2 + 4M_J^2(L_{22}^F)^2} + 2(L_y + 1)$ and $\tilde{C}_1 := 2C_y^F + 2(L_J + 1)$. Rearranging the above inequality and taking square root on both sides, we have that

$$\begin{aligned}
A^l &\leq \sqrt{\frac{2\max\{C_1, \tilde{C}_1, \frac{1}{\beta}, \sqrt{2}\Delta\}}{\min\{\Delta, \delta\}}\phi\left((H^l - H^*) - (H^{l+1} - H^*)\right)} \cdot \sqrt{\frac{A^{l+1} + A^l}{2}} \\
&\leq \frac{\max\{C_1, \tilde{C}_1, \frac{1}{\beta}, \sqrt{2}\Delta\}}{\min\{\Delta, \delta\}}\phi\left((H^l - H^*) - (H^{l+1} - H^*)\right) + \frac{A^{l+1} + A^l}{4}.
\end{aligned}$$

Rearranging the above inequality, we have that

$$\frac{1}{2}A^l \leq \frac{\max\{C_1, \tilde{C}_1, \frac{1}{\beta}}{,}\sqrt{2}\Delta\}\min\{\Delta, \delta\}\phi\left((H^l - H^*) - (H^{l+1} - H^*)\right) + \frac{1}{4}\left(A^l - A^{l+1}\right). \tag{29}$$

Summing the above inequality from any $\underline{l} \geq \max\{l_1, l_2\} + 2$ and recalling $H^l$ is convergence to $H^*$ given by Theorem 4, we obtain that

$$\sum_{l=\underline{l}}^{\infty} A^l \leq 2\frac{\max\{C_1, \tilde{C}_1, \frac{1}{\beta}, \sqrt{2}\Delta\}}{\min\{\Delta, \delta\}}\phi(H^{\underline{l}} - H^*) + \frac{1}{2}A^{\underline{l}} < \infty, \tag{30}$$

Thanks to the Lipschitz continuity of $y(x)$ and $J(y(x))$ guaranteed by Lemmas 1 and 2, we have that

$$
\begin{aligned}
\sum_{l=\underline{1}}^{\infty} \|(x^l, y^l, J^l) - (x^{l-1}, y^{l-1}, J^{l-1})\| &\leq \sum_{l=\underline{1}}^{\infty} \left( \|x^l - x^{l-1}\| + \|y^l - y^{l-1}\| + \|J^l - J^{l-1}\|_F \right) \\
&\leq \sum_{l=\underline{1}}^{\infty} \left( D_1^l + D_2^{l-1} + D_2^{l-2} + \|y(x^{l-1}) - y(x^{l-2})\| + D_3^{l-1} + D_3^{l-2} + \|J(x^{l-1}) - J(x^{l-2})\|_F \right) \\
&\leq \sum_{l=\underline{1}}^{\infty} \left( D_1^l + D_2^{l-1} + D_2^{l-2} + L_y D_1^{l-1} + D_3^{l-1} + D_3^{l-2} + L_J D_1^{l-1} \right) \\
&\leq \max\{1, L_y, L_J\} \left( \sum_{l=\underline{1}}^{\infty} 3A^l + 2A^{l-1} + A^{l-2} \right) < \infty,
\end{aligned}
\tag{31}
$$

where the last inequality uses (30). This inequality implies that the sequence $\{(x^l, y^l, J^l)\}$ and $\{Z^l\}$ are convergent.

Now, we show the convergence rate when $H$ is a KL function with exponent $\vartheta$. First, when $\vartheta = 0$, there exists $\bar{l}$ such that $H^l == H^*$ when $l \geq \bar{l}$. In fact, suppose to the contrary that $H^l > H^*$ for some $l > \bar{l}$. Since $Z^l$ is convergent (denote $\lim_l Z^l = Z^*$) and $H^l$ is nonincreasing thanks to Theorem 4, noting that $\phi(s) = cs$ and the KL inequality holds for large $l$, it holds that $\mathrm{dist}(0, \partial H(Z^l)) \geq \frac{1}{c}$, contradicting (25). Thus, there exists $\bar{l}$ such that $H^l == H^*$ when $l \geq \bar{l}$ and recalling the arguments in the beginning of this theorem, we have $\{Z^l\}$ converges finitely.

Finally, we consider the case where $\vartheta \in (0, 1)$ and $H^l > H^*$ for all $l$. Denote $B^l = \sum_{k=l}^{\infty} A^k$, which is well defined thanks to (31). Then, for any $l > \underline{1}$ using (29)

$$
\begin{aligned}
B^l &\leq \sum_{k=l}^{\infty} \left( \frac{2\max\{C_1, \tilde{C}_1, \frac{1}{\beta}, \sqrt{2}\Delta\}}{\min\{\Delta, \delta\}} \phi\left((H^k - H^*) - (H^{k+1} - H^*)\right) + \frac{1}{2} \sum_{k=l-2}^{\infty} \left(A^k - A^{k+1}\right) \right) \\
&\leq 2\frac{\max\{C_1, \tilde{C}_1, \frac{1}{\beta}, \sqrt{2}\Delta\}}{\min\{\Delta, \delta\}} \phi(H^l - H^*) + \frac{1}{2} A^{l-2} \\
&= 2\frac{\max\{C_1, \tilde{C}_1, \frac{1}{\beta}, \sqrt{2}\Delta\}}{\min\{\Delta, \delta\}} \phi(H^l - H^*) + \frac{1}{2}(B^{l-2} - B^{l-1}) \\
&\leq 2\frac{\max\{C_1, \tilde{C}_1, \frac{1}{\beta}, \sqrt{2}\Delta\}}{\min\{\Delta, \delta\}} \phi(H^l - H^*) + \frac{1}{2}(B^{l-2} - B^l).
\end{aligned}
$$

Now, following the classical steps for analyzing the convergence rate under the KL assumptions see (Attouch et al. (2010); Attouch & Bolte (2009); Liu et al. (2019); Wen et al. (2018) for examples, we have

(i) $B^l$ converges linearly when $\vartheta \in (0, \frac{1}{2}]$.

(ii) $B^l$ convergence sublinearly when $\vartheta \in (\frac{1}{2}, 1)$.

This together with

$$
\begin{aligned}
\|(x^{\underline{1}}, y^{\underline{1}}, J^{\underline{1}}) - (x^*, y^*, J^*)\| &\leq \sum_{l=\underline{1}}^{\infty} \|(x^l, y^l, J^l) - (x^{l-1}, y^{l-1}, J^{l-1})\| \\
&\leq \max\{1, M_J, L_J\} \left( \sum_{l=\underline{1}}^{\infty} 3A^l + 2A^{l-1} + A^{l-2} \right) \\
&= \max\{1, M_J, L_J\} \left( 3B^{\underline{1}} + 2A^{\underline{1}-1} + A^{\underline{1}-2} \right) \leq 2\max\{1, M_J, L_J\} B^{\underline{1}-2}
\end{aligned}
$$

guaranteed by the convergence of $\{(x^l, y^l, J^l)\}$ and (31), we obtain the same convergence rate of $(x^l_\perp, y^l_\perp, J^l_\perp)$ as $B^l_\perp$. This completes the proof. $\qquad\square$

## B.4 Details of results in Section Convergence Analysis for Algorithms 3

**Lemma 6.** *Consider* (1) *and suppose Assumptions 1, 2 and 3 hold. Let* $\{(x^l, y^l)\}$ *be generated by algorithm 3. Denote* $\tilde{\zeta}_s := \frac{\mu_G - 2c^2}{\frac{1}{2\alpha} + \frac{\mu_G}{2}}$ *with* $c \in \left(0, \sqrt{\frac{\mu_g}{2}}\right)$. *Denote* $\zeta_s := 1 - (1 + d^2_{y_s})(1 - \tilde{\zeta}_s)$ *with* $d_{y_s} \in \left(0, \sqrt{\frac{1}{1-\tilde{\zeta}_s} - 1}\right)$. *Suppose* $\alpha < \frac{1}{L_G}$. *Let* $x^{-1} = x^0$ *and pick any* $\{u^{-2}, u^{-1}\} \subseteq \mathbb{R}^n$ *and* $\{v^{-2}, v^{-1}\} \subseteq \mathbb{R}^m$. *Let* $\mathcal{S}^{-1}$, $\mathcal{S}^{-2}$, $\mathcal{B}^{-1}$ *and* $\mathcal{B}^{-2}$ *be any sample batches. Then* $\zeta_s \in (0,1)$ *and for any* $l \geq 0$,

$$
\begin{aligned}
\mathbb{E}_{\tilde{R}^l} \|y^{l+1} - y(x^l)\|^2 &\leq (1 - \zeta_s)\mathbb{E}_{\tilde{R}^{l-1}}\|y(x^{l-1}) - y^l\|^2 \\
&+ (1 + d^{-2}_{y_s})(1 - \tilde{\zeta}_s)\mathbb{E}_{R^{l-1}}L^2_y \|x^l - x^{l-1}\|^2 + \frac{2c^{-2}}{\left(\frac{\mu_G}{2} + \frac{1}{2\alpha}\right)}\sigma^2_G
\end{aligned}
\tag{32}
$$

*or equivalently,*

$$
\begin{aligned}
\mathbb{E}_{\tilde{R}^{l-1}}\|y(x^{l-1}) - y^l\|^2 &\leq \frac{1}{\zeta_s}\left(\mathbb{E}_{\tilde{R}^{l-1}}\|y(x^{l-1}) - y^l\|^2 - \mathbb{E}_{\tilde{R}^l}\|y^{l+1} - y(x^l)\|^2\right) \\
&+ \frac{(1 + d^{-2}_{y_s})(1 - \tilde{\zeta}_s)}{\zeta_s}L^2_y\mathbb{E}_{R^{l-1}}\|x^l - x^{l-1}\|^2 + \frac{1}{\zeta_s}\frac{2c^{-2}}{\frac{\mu_G}{2} + \frac{1}{2\alpha}}\sigma^2_G.
\end{aligned}
\tag{33}
$$

*Proof.* Thanks to Assumption 2, for $l \geq 0$, we have that

$$
\begin{aligned}
G(x^l, y^{l+1}) &\leq G(x^l, y^l) + \langle \nabla_y G(x^l, y^l), y^{l+1} - y^l \rangle + \frac{L_G}{2}\|y^{l+1} - y^l\|^2 \\
&= G(x^l, y^l) + \langle \nabla_y G(x^l, y^l; \mathcal{S}^l), y^{l+1} - y^l \rangle + \frac{1}{2\alpha}\|y^{l+1} - y^l\|^2 \\
&+ \langle \nabla G(x^l, y^l) - \nabla_y G(x^l, y^l; \mathcal{S}^l), y^{l+1} - y^l \rangle + \frac{L_G - \frac{1}{\alpha}}{2}\|y^{l+1} - y^l\|^2 \\
&\leq G(x^l, y^l) + \langle \nabla_y G(x^l, y^l; \mathcal{S}^l), y(x^l) - y^l \rangle + \frac{1}{2\alpha}\|y(x^l) - y^l\|^2 - \frac{1}{2\alpha}\|y(x^l) - y^{l+1}\|^2 \\
&+ \langle \nabla_y G(x^l, y^l) - \nabla_y G(x^l, y^l; \mathcal{S}^l), y^{l+1} - y^l \rangle + \frac{L_G - \frac{1}{\alpha}}{2}\|y^{l+1} - y^l\|^2 \\
&\leq G(x^l, y^l) + \langle \nabla_y G(x^l, y^l; \mathcal{S}^l), y(x^l) - y^l \rangle + \frac{1}{2\alpha}\|y(x^l) - y^l\|^2 - \frac{1}{2\alpha}\|y(x^l) - y^{l+1}\|^2 \\
&+ \frac{2}{c^2}\|\nabla_y G(x^l, y^l) - \nabla_y G(x^l, y^l; \mathcal{S}^l)\|^2 + 2c^2\|y(x^l) - y^l\|^2 + \frac{L_G - \frac{1}{\alpha}}{2}\|y^{l+1} - y^l\|^2,
\end{aligned}
$$

where the second inequality is because $y^{l+1}$ in Step 3 of Algorithm 2 is the minimizer of $\arg\min G(x^l, y^l) + \langle \nabla_y G(x^l, y^l; \mathcal{S}^l), y^{l+1} - y^l \rangle + \frac{1}{2\alpha}\|y^{l+1} - y^l\|^2$ and the objective of this subproblem is strongly convex with modulus $\frac{1}{\alpha}$. Taking the conditional expectation of $\mathcal{S}^l$ on both sides of the above inequality and recalling Assumption 3, for $l \geq 0$, we have that

$$
\begin{aligned}
\mathbb{E}_{\mathcal{S}^l | R^{l-1}} G(x^l, y^{l+1}) &\leq G(x^l, y^l) + \langle \nabla_y G(x^l, y^l), y(x^l) - y^l \rangle + \frac{1}{2\alpha}\|y(x^l) - y^l\|^2 \\
&- \mathbb{E}_{\mathcal{S}^l | R^{l-1}}\frac{1}{2\alpha}\|y(x^l) - y^{l+1}\|^2 + \frac{2}{c^2}\sigma^2_G + 2c^2\|y(x^l) - y^l\| \\
&+ \mathbb{E}_{\mathcal{S}^l | R^{l-1}}\frac{L_G - \frac{1}{\alpha}}{2}\|y^{l+1} - y^l\|^2 \\
&\leq G(x^l, y(x^l)) - \frac{\mu_G}{2}\|y(x^l) - y^l\|^2 + \frac{1}{2\alpha}\|y(x^l) - y^l\|^2 - \mathbb{E}_{\mathcal{S}^l | R^{l-1}}\frac{1}{2\alpha}\|y(x^l) - y^{l+1}\|^2 \\
&+ \frac{2}{c^2}\sigma^2_G + 2c^2\|y(x^l) - y^l\| + \mathbb{E}_{\mathcal{S}^l | R^{l-1}}\frac{L_G - \frac{1}{\alpha}}{2}\|y^{l+1} - y^l\|^2,
\end{aligned}
$$

where the second inequality uses the strong convexity of $G(x, \cdot)$. Rearranging the above inequality, for $l \geq 0$, it holds that

$$
\begin{aligned}
\mathbb{E}_{S^l | R^{l-1}}(G(x^l, y^{l+1}) - G(x^l, y(x^l))) \leq & -\frac{\mu_G}{2}\|y(x^l) - y^l\|^2 + \frac{1}{2\alpha}\|y(x^l) - y^l\|^2 \\
& - \mathbb{E}_{S^l | R^{l-1}}\frac{1}{2\alpha}\|y(x^l) - y^{l+1}\|^2 + \frac{2}{c^2}\sigma_G^2 + 2c^2\|y(x^l) - y^l\| \\
& + \mathbb{E}_{S^l | R^{l-1}}\frac{L_G - \frac{1}{\alpha}}{2}\|y^{l+1} - y^l\|^2.
\end{aligned}
\tag{34}
$$

Since $G(x, \cdot)$ is strongly convex with modulus $\mu_G$ and $y(x^l)$ is $\arg\min_y G(x^l, y)$ by definition, for $l \geq 0$, we have

$$
\begin{aligned}
\mathbb{E}_{S^l | R^{l-1}}\frac{\mu_G}{2}\|y^{l+1} - y(x^l)\|^2 & \leq \mathbb{E}_{S^l | R^{l-1}}(G(x^l, y^{l+1}) - G(x^l, y(x^l))) \\
& \leq -\frac{\mu_G}{2}\|y(x^l) - y^l\|^2 + \frac{1}{2\alpha}\|y(x^l) - y^l\|^2 + 2c^2\|y(x^l) - y^l\| \\
& \quad - \mathbb{E}_{S^l | R^{l-1}}\frac{1}{2\alpha}\|y(x^l) - y^{l+1}\|^2 + \frac{2}{c^2}\sigma_G^2 + \mathbb{E}_{S^l | R^{l-1}}\frac{L_G - \frac{1}{\alpha}}{2}\|y^{l+1} - y^l\|^2 \\
& \leq -\frac{\mu_G}{2}\|y(x^l) - y^l\|^2 + \left(\frac{1}{2\alpha} + 2c^2\right)\|y(x^l) - y^l\|^2 - \mathbb{E}_{S^l | R^{l-1}}\frac{1}{2\alpha}\|y(x^l) - y^{l+1}\|^2 + \frac{2}{c^2}\sigma_G^2,
\end{aligned}
$$

where the second inequality uses (34) and the third inequality is because the assumption that $L_G - \frac{1}{\alpha} < 0$. Rearranging the above inequality and dividing both sides with $\frac{\mu_G}{2} + \frac{1}{2\alpha}$, for $l \geq 0$, we obtain that

$$
\mathbb{E}_{S^l | R^{l-1}}\|y^{l+1} - y(x^l)\|^2 \leq (1 - \tilde{\zeta}_s)\|y(x^l) - y^l\|^2 + \frac{2}{c^2(\frac{\mu_G}{2} + \frac{1}{2\alpha})}\sigma_G^2,
$$

where $\tilde{\zeta}_s = \frac{\mu_G - 2c^2}{\frac{1}{2\alpha} + \frac{\mu_G}{2}} \in (0, 1)$ thanks to the assumption that $0 < c < \sqrt{\frac{\mu_g}{2}}$ and $\alpha < \frac{1}{L_G}$. Now taking the expectation on $\{R^{l-1}\}$ on both sides, for $l \geq 0$, we have that

$$
\begin{aligned}
\mathbb{E}_{\tilde{R}^l}\|y^{l+1} - y(x^l)\|^2 & \leq (1 - \tilde{\zeta}_s)\mathbb{E}_{R^{l-1}}\|y(x^l) - y^l\|^2 + \frac{2}{c^2(\frac{\mu_G}{2} + \frac{1}{2\alpha})}\sigma_G^2 \\
& \leq (1 + d_{y_s}^2)(1 - \tilde{\zeta}_s)\mathbb{E}_{R^{l-1}}\|y(x^{l-1}) - y^l\|^2 + (1 + d_{y_s}^{-2})(1 - \tilde{\zeta}_s)\mathbb{E}_{R^{l-1}}\|y(x^l) - y(x^{l-1})\|^2 \\
& \quad + \frac{2}{c^2(\frac{\mu_G}{2} + \frac{1}{2\alpha})}\sigma_G^2 \\
& \leq (1 + d_{y_s}^2)(1 - \tilde{\zeta}_s)\mathbb{E}_{R^{l-1}}\|y(x^{l-1}) - y^l\|^2 + (1 + d_{y_s}^{-2})(1 - \tilde{\zeta}_s)\mathbb{E}_{R^{l-1}}L_y^2\|x^l - x^{l-1}\|^2 \\
& \quad + \frac{2}{c^2(\frac{\mu_G}{2} + \frac{1}{2\alpha})}\sigma_G^2 \\
& = (1 - \zeta_s)\mathbb{E}_{\tilde{R}^{l-1}}\|y(x^{l-1}) - y^l\|^2 + (1 + d_{y_s}^{-2})(1 - \tilde{\zeta}_s)\mathbb{E}_{R^{l-1}}L_y^2\|x^l - x^{l-1}\|^2 \\
& \quad + \frac{2}{c^2(\frac{\mu_G}{2} + \frac{1}{2\alpha})}\sigma_G^2,
\end{aligned}
$$

where the third inequality uses Lemma 1 and the last inequality uses the definition of $\zeta_s$. Thanks to the assumption that $d_{y_s}^2 < \frac{1}{1 - \tilde{\zeta}_s} - 1$, we have that $0 < \zeta_s < 1$.

Rearranging the above inequality, we have that

$$
\begin{aligned}
\mathbb{E}_{\tilde{R}^{l-1}}\|y(x^{l-1}) - y^l\|^2 \leq & \frac{1}{\zeta_s}\left(\mathbb{E}_{\tilde{R}^{l-1}}\|y(x^{l-1}) - y^l\|^2 - \mathbb{E}_{\tilde{R}^l}\|y^{l+1} - y(x^l)\|^2\right) + \\
& + \frac{1 + d_{y_s}^{-2}}{\zeta_s}\left(1 - \tilde{\zeta}_s\right)L_y^2\mathbb{E}_{R^{l-1}}\|x^l - x^{l-1}\|^2 + \frac{1}{\zeta_s}\frac{2}{c^2(\frac{\mu_G}{2} + \frac{1}{2\alpha})}\sigma_G^2.
\end{aligned}
$$

$\square$

Next we estimate $\hat{\nabla}_{xy}G(x^l, y^l)$ and $\hat{\nabla}_{yy}G(x^l, y^l)$. We first present the following basic properties shown in Nesterov & Spokoiny (2017), see equation (21), Lemma 3 and Lemma 5 in Nesterov & Spokoiny (2017) respectively.

**Lemma 7.** *Let $h : \mathbb{R}^n \to \mathbb{R}$ be a differentiable function with $L$-Lipschitz gradient. Define $h_\mu(x) = \mathbb{E}_u[h(x + \mu u)]$, where $\mu > 0$ and $u$ is a standard Gaussian random vector. We have that:*

*(i) $h_\mu$ is differentiable and $\nabla h_\mu(x) = \mathbb{E}_u \widehat{\nabla} h(x, u)$, where $\widehat{\nabla} h(x, u) = \frac{h(x+\mu u) - h(x)}{\mu} u$.*

*(ii) $\|\nabla h_\mu(x) - \nabla h(x)\| \leq \frac{\mu^2}{2} L(n+3)^{\frac{3}{2}}$.*

*(iii) $\mathbb{E}_u \left\| \widehat{\nabla} h(x, u) \right\|^2 \leq 4(n+4)\|\nabla h_\mu(x)\|^2 + 3\mu^2 L^2 (n+4)^3$.*

Now we bound nabla $\nabla_{xy}G$ and $\nabla_{xy}G$.

**Lemma 8.** *Consider* (1) *and let Assumptions 2 and 3 hold. Let $\{(x^l, y^l)\}$ be generated by Algorithm 3. Denote*

$$\Delta_{xy} = \frac{2m}{Q^2} \left( (2Q^2 + 4Q(4n+15)) \frac{\mu^4}{4} L_{G_x^x}^2 (n+3)^3 + 6Q\mu^2 L_{G_x^x}^2 (n+4)^3 \right)$$
$$+ \frac{C_{Gxy}^2 + \sigma_{yy}^2}{Q} 8(4n+15)$$

*and*

$$\Delta_{yy} = \frac{2m}{Q^2} \left( (2Q^2 + 4Q(4m+15)) \frac{\mu^4}{4} L_{G_y^y}^2 (m+3)^3 + 6Q\mu^2 L_{G_y^y}^2 (m+4)^3 \right)$$
$$+ 4(4m+15) \frac{C_{Gyy}^2 + \sigma_{yy}^2}{Q}.$$

*Then it holds that*

$$\mathbb{E}_{u^l, \mathcal{S}^l | R^{l-1}} \|\nabla_{xy}G(x^l, y^l) - \hat{\nabla}_{xy}G(x^l, y^l; \mathcal{S}^l)\|^2. \leq 2\sigma_{xy}^2 + \Delta_{xy}$$

*and*

$$\mathbb{E}_{v^l, \mathcal{S}^l | R^{l-1}} \|\nabla_{yy}G(x^l, y^l) - \widehat{\nabla}_{yy}G(x^l, y^l; \mathcal{S}^l)\|^2 \leq 2\sigma_{yy}^2 + \Delta_{yy} \text{ for } \hat{\nabla}_{yy}G(x^l, y^l).$$

*Proof.* Thanks to Assumption 3, it holds that

$$\mathbb{E}_{u^l | \tilde{R}^l} \|\nabla_{xy}G(x^l, y^l) - \hat{\nabla}_{xy}G(x^l, y^l; \mathcal{S}^l)\|^2$$
$$\leq 2\mathbb{E}_{u^l | \tilde{R}^l} \|\nabla_{xy}G(x^l, y^l) - \nabla_{xy}G(x^l, y^l; \mathcal{S}^l)\|^2 \tag{35}$$
$$+ 2\mathbb{E}_{u^l | \tilde{R}^l} \|\nabla_{xy}G(x^l, y^l; \mathcal{S}^l) - \hat{\nabla}_{xy}G(x^l, y^l; \mathcal{S}^l)\|^2$$
$$\leq 2\sigma_{xy}^2 + 2\mathbb{E}_{u^l | \tilde{R}^l} \|\nabla_{xy}G(x^l, y^l; \mathcal{S}^l) - \hat{\nabla}_{xy}G(x^l, y^l; \mathcal{S}^l)\|^2.$$

By the definition of $\nabla_{xy}G(x^l, y^l; \mathcal{S}^l)$ and $\hat{\nabla}_{xy}G(x^l, y^l)$, we have that

$$\mathbb{E}_{u^l | \tilde{R}^l} \|\nabla_{xy}G(x^l, y^l; \mathcal{S}^l) - \hat{\nabla}_{xy}G(x^l, y^l)\|^2$$
$$\leq \sum_{i=1}^m \mathbb{E}_{u^l | \tilde{R}^l} \left\| \frac{1}{|\mathcal{S}^l|} \sum_{r=1}^{|\mathcal{S}^l|} \left( \nabla_x(\nabla_{y_i}G(x^l, y^l; \xi_S^r)) - \frac{1}{Q} \sum_{j=1}^Q \delta_{y_i}^j u_j^l \right) \right\|^2$$
$$\leq \sum_{i=1}^m \frac{1}{|\mathcal{S}^l|} \sum_{r=1}^{|\mathcal{S}^l|} \mathbb{E}_{u^l | \tilde{R}^l} \left\| \nabla_x(\nabla_{y_i}G(x^l, y^l; \xi_S^r)) - \frac{1}{Q} \sum_{j=1}^Q \delta_{y_i}^j u_j^l \right\|^2 \tag{36}$$
$$= \frac{\frac{1}{Q^2}}{|\mathcal{S}^l|} \sum_{\substack{r=1, \\ i=1}}^{\substack{r=|\mathcal{S}^l|, \\ i=m}} \mathbb{E}_{u^l | \tilde{R}^l} \left\| \sum_{j=1}^Q (\nabla_x(\nabla_{y_i}G(x^l, y^l; \xi_S^r)) - \delta_{y_i}^j u_j^l) \right\|^2.$$

where $\{\xi_S^r\}_{r=1}^{|\mathcal{S}^l|}$ are the elements in $\mathcal{S}^l$. Denoting

$$\nabla_{y_i}^\mu G(x, y^l; \xi_S^r) := \mathbb{E}_{u^l|\tilde{R}^l} \nabla_{y_i}^\mu G(x + \mu u^l, y^l; \xi_S^r),$$

we have that,

$$
\mathbb{E}_{u^l|\tilde{R}^l} \left\| \sum_{j=1}^Q \left( \nabla_x(\nabla_{y_i} G(x^l, y^l; \xi_S^r)) - \frac{\nabla_{y_i} G(x^l + \mu u_j^l, y^l; \xi_S^r) - \nabla_{y_i} G(x^l, y^l; \xi_S^r)}{\mu} u_j^l \right) \right\|^2
$$

$$
\leq 2 \left\| \sum_{j=1}^Q \left( \nabla_x(\nabla_{y_i} G(x^l, y^l; \xi_S^r)) - \nabla_x(\nabla_{y_i}^\mu G(x^l, y^l; \xi_S^r)) \right) \right\|^2
$$

$$
+ 2\mathbb{E}_{u^l|\tilde{R}^l} \left\| \sum_{j=1}^Q \left( \nabla_x(\nabla_{y_i}^\mu G(x^l, y^l; \xi_S^r)) - \frac{\nabla_{y_i} G(x^l + \mu u_j^l, y^l; \xi_S^r) - \nabla_{y_i} G(x^l, y^l; \xi_S^r)}{\mu} u_j^l \right) \right\|^2 \tag{37}
$$

$$
= 2Q \sum_{j=1}^Q \left\| \nabla_x(\nabla_{y_i} G(x^l, y^l; \xi_S^r)) - \nabla_x(\nabla_{y_i}^\mu G(x^l, y^l; \xi_S^r)) \right\|^2
$$

$$
+ 2\mathbb{E}_{u^l|\tilde{R}^l} \left\| \sum_{j=1}^Q \left( \nabla_x(\nabla_{y_i}^\mu G(x^l, y^l; \xi_S^r)) - \frac{\nabla_{y_i} G(x^l + \mu u_j^l, y^l; \xi_S^r) - \nabla_{y_i} G(x^l, y^l; \xi_S^r)}{\mu} u_j^l \right) \right\|^2.
$$

For the first term in the above inequality, using Lemma 7 (ii) together with Assumption 2 (i), we have that

$$
\left\| \nabla_x(\nabla_{y_i} G(x^l, y^l; \xi_S^r)) - \nabla_x(\nabla_{y_i}^\mu G(x^l, y^l; \xi_S^r)) \right\|^2 \leq \frac{\mu^4}{4} L_{G_x^x}^2 (n+3)^3. \tag{38}
$$

For the second term in (37), denote

$$
r_j := \nabla_x(\nabla_{y_i}^\mu G(x^l, y^l; \xi_S^r)) - \frac{\nabla_{y_i} G(x^l + \mu u_j^l, y^l; \xi_S^r) - \nabla_{y_i} G(x^l, y^l; \xi_S^r)}{\mu} (u_j^l).
$$

Then we have

$$
\left\| \sum_{j=1}^Q \left( \nabla_x(\nabla_{y_i}^\mu G(x^l, y^l; \xi_S^r)) - \frac{\nabla_{y_i} G(x^l + \mu u_j^l, y^l; \xi_S^r) - \nabla_{y_i} G(x^l, y^l; \xi_S^r)}{\mu} u_j^l \right) \right\|^2
$$

$$
= \sum_{j=1}^Q \left\| \nabla_x(\nabla_{y_i}^\mu G(x^l, y^l; \xi_S^r)) - \frac{\nabla_{y_i} G(x^l + \mu u_j^l, y^l; \xi_S^r) - \nabla_{y_i} G(x^l, y^l; \xi_S^r)}{\mu} u_j^l \right\|^2 + 2 \sum_{j_1 < j_2} \langle r_{j_1}, r_{j_2} \rangle. \tag{39}
$$

Thanks to Lemma 7 (i) and that $u_{j_1}^l$ and $u_{j_2}^l$ are independent, we have that

$$
\mathbb{E}_{u_{j_1}^l, u_{j_2}^l | \tilde{R}^l} \sum_{j_1 < j_2} \langle r_{j_1}, r_{j_2} \rangle = 0, \forall\, j_1 \neq j_2.
$$

Thus, taking expectation of $u^l$, (39) can be further passed to

$$
\mathbb{E}_{u^l|\tilde{R}^l}\left\|\sum_{j=1}^{Q}\left(\nabla_x(\nabla_{y_i}^{\mu}G(x^l,y^l;\xi_S^r))-\frac{\nabla_{y_i}G(x^l+\mu u_j^l,y^l;\xi_S^r)-\nabla_{y_i}G(x^l,y^l;\xi_S^r)}{\mu}u_j^l\right)\right\|^2
$$

$$
=\sum_{j=1}^{Q}\mathbb{E}_{u_j^l|\tilde{R}^l}\left\|\nabla_x(\nabla_{y_i}^{\mu}G(x^l,y^l;\xi_S^r))-\frac{\nabla_{y_i}G(x^l+\mu u_j^l,y^l;\xi_S^r)-\nabla_{y_i}G(x^l,y^l;\xi_S^r)}{\mu}u_j^l\right\|^2
$$

$$
\stackrel{(a)}{=}\sum_{j=1}^{Q}\left(\mathbb{E}_{u_j^l|\tilde{R}^l}\left\|\frac{\nabla_{y_i}G(x^l+\mu u_j^l,y^l;\xi_S^r)-\nabla_{y_i}G(x^l,y^l;\xi_S^r)}{\mu}u_j^l\right\|^2-\left\|\nabla_x(\nabla_{y_i}^{\mu}G(x^l,y^l;\xi_S^r))\right\|^2\right)
$$

$$
\stackrel{(b)}{\leq}\sum_{j=1}^{Q}\left(4(n+4)\left\|\nabla_x(\nabla_{y_i}^{\mu}G(x^l,y^l;\xi_S^r))\right\|^2+3\mu^2 L_{G_x^x}^2(n+4)^3-\left\|\nabla_x(\nabla_{y_i}^{\mu}G(x^l,y^l;\xi_S^r))\right\|^2\right) \tag{40}
$$

$$
=\sum_{j=1}^{Q}\left((4n+15)\left\|\nabla_x(\nabla_{y_i}^{\mu}G(x^l,y^l;\xi_S^r))\right\|^2+3\mu^2 L_{G_x^x}^2(n+4)^3\right)
$$

$$
\leq\sum_{j=1}^{Q}(2(4n+15)\left\|\nabla_x(\nabla_{y_i}^{\mu}G(x^l,y^l;\xi_S^r))-\nabla_x(\nabla_{y_i}G(x^l,y^l;\xi_S^r))\right\|^2
$$

$$
+\sum_{j=1}^{Q}2(4n+15)\left\|\nabla_x(\nabla_{y_i}G(x^l,y^l;\xi_S^r))\right\|^2+\sum_{j=1}^{Q}3\mu^2 L_{G_x^x}^2(n+4)^3
$$

$$
\leq\sum_{j=1}^{Q}\left(2(4n+15)\left(\frac{\mu^4}{4}L_{G_x^x}^2(n+3)^3\right)+2(4n+15)\left\|\nabla_x(\nabla_{y_i}G(x^l,y^l;\xi_S^r))\right\|^2+3\mu^2 L_{G_x^x}^2(n+4)^3\right),
$$

where (a) is because $\mathbb{E}_{u_j^l|\tilde{R}^l}\left\|\nabla_x(\nabla_{y_i}^{\mu}G(x^l,y^l;\xi_S^r))-\frac{\nabla_{y_i}G(x^l+\mu u_j^l,y^l;\xi_S^r)-\nabla_{y_i}G(x^l,y^l;\xi_S^r)}{\mu}u_j^l\right\|^2$ is the variance of $\frac{\nabla_{y_i}G(x^l+\mu u_j^l,y^l;\xi_S^r)-\nabla_{y_i}G(x^l,y^l;\xi_S^r)}{\mu}u_j^l$, (b) uses Lemma 7 (iii) and the last inequality uses Lemma 7 (ii) and Assumption 2 (i).

Now, combining (37), (38) and (40), we have that

$$
\mathbb{E}_{u^l|\tilde{R}^l}\left\|\sum_{j=1}^{Q}\left(\nabla_x(\nabla_{y_i}G(x^l,y^l;\xi_S^r))-\frac{\nabla_{y_i}G(x^l+\mu u_j^l,y^l;\xi_S^r)-\nabla_{y_i}G(x^l,y^l;\xi_S^r)}{\mu}u_j^l\right)\right\|^2
$$

$$
\leq 2Q\sum_{j=1}^{Q}\frac{\mu^4}{4}L_{G_x^x}^2(n+3)^3+2\sum_{j=1}^{Q}2(4n+15)\left(\frac{\mu^4}{4}L_{G_x^x}^2(n+3)^3\right)
$$

$$
+2\sum_{j=1}^{Q}2(4n+15)\left\|\nabla_x(\nabla_{y_i}G(x^l,y^l;\xi_S^r))\right\|^2+2\sum_{j=1}^{Q}3\mu^2 L_{G_x^x}^2(n+4)^3
$$

$$
=\left(2Q^2+4Q(4n+15)\right)\frac{\mu^4}{4}L_{G_x^x}^2(n+3)^3+6Q\mu^2 L_{G_x^x}^2(n+4)^3
$$

$$
+4Q(4n+15)\left\|\nabla_x(\nabla_{y_i}G(x^l,y^l;\xi_S^r))\right\|^2.
$$

This together with (36) gives

$$\mathbb{E}_{u^l|\tilde{R}^l}\|\nabla_{xy}G(x^l,y^l;\mathcal{S}^l) - \hat{\nabla}_{xy}G(x^l,y^l;\mathcal{S}^l)\|^2$$

$$\leq \frac{1}{|\mathcal{S}^l|Q^2}\sum_{r=1}^{|\mathcal{S}^l|}\sum_{i=1}^{m}\left(2Q^2 + 4Q(4n+15)\right)\frac{\mu^4}{4}L_{G_x^x}^2(n+3)^3 + \frac{1}{|\mathcal{S}^l|Q^2}\sum_{r=1}^{|\mathcal{S}^l|}\sum_{i=1}^{m}6Q\mu^2 L_{G_x^x}^2(n+4)^3$$

$$+ \frac{1}{|\mathcal{S}^l|Q^2}\sum_{r=1}^{|\mathcal{S}^l|}\sum_{i=1}^{m}4Q(4n+15)\left\|\nabla_x(\nabla_{y_i}G(x^l,y^l;\xi_S^r))\right\|^2$$

$$= \frac{m}{|\mathcal{S}^l|Q^2}\sum_{r=1}^{|\mathcal{S}^l|}\left(2Q^2 + 4Q(4n+15)\right)\frac{\mu^4}{4}L_{G_x^x}^2(n+3)^3 + \frac{m}{|\mathcal{S}^l|Q^2}\sum_{r=1}^{|\mathcal{S}^l|}6Q\mu^2 L_{G_x^x}^2(n+4)^3$$

$$+ 4(4n+15)\frac{1}{|\mathcal{S}^l|Q}\sum_{r=1}^{|\mathcal{S}^l|}\sum_{i=1}^{m}\left\|\nabla_x(\nabla_{y_i}G(x^l,y^l;\xi_S^r))\right\|^2.$$

Taking the conditional expextion of $\mathcal{S}^l$ on both side of the above inequality, recalling Assumption 2 (i) and Assumption 3, we have that

$$\mathbb{E}_{u^l,\mathcal{S}^l|R^{l-1}}\|\nabla_{xy}G(x^l,y^l;\mathcal{S}^l) - \hat{\nabla}_{xy}G(x^l,y^l;\mathcal{S}^l)\|^2$$

$$\leq \frac{m}{Q^2}\left(\left(2Q^2 + 4Q(4n+15)\right)\frac{\mu^4}{4}L_{G_x^x}^2(n+3)^3 + 6Q\mu^2 L_{G_x^x}^2(n+4)^3\right)$$

$$+ \frac{C_{Gxy}^2 + \sigma_{xy}^2}{Q}4(4n+15).$$

This together with (35), we have

$$\mathbb{E}_{u^l,\mathcal{S}^l|R^{l-1}}\|\nabla_{xy}G(x^l,y^l) - \hat{\nabla}_{xy}G(x^l,y^l;\mathcal{S}^l)\|^2$$

$$\leq 2\sigma_{xy}^2 + \frac{2m}{Q^2}\left(\left(2Q^2 + 4Q(4n+15)\right)\frac{\mu^4}{4}L_{G_x^x}^2(n+3)^3 + 6Q\mu^2 L_{G_x^x}^2(n+4)^3\right)$$

$$+ \frac{C_{Gxy}^2 + \sigma_{xy}^2}{Q}8(4n+15).$$

In the same way, we can calculate that

$$\mathbb{E}_{u^l,\mathcal{S}^l|R^{l-1}}\|\nabla_{yy}G(x^l,y^l) - \widehat{\nabla}_{yy}G(x^l,y^l;\mathcal{S}^l)\|^2$$

$$\leq 2\sigma_{yy}^2 + \frac{2m}{Q^2}\left(\left(2Q^2 + 4Q(4m+15)\right)\frac{\mu^4}{4}L_{G_y^y}^2(m+3)^3 + 6Q\mu^2 L_{G_y^y}^2(m+4)^3\right)$$

$$+ 4(4m+15)\frac{C_{Gyy}^2 + \sigma_{yy}^2}{Q}.$$

$\square$

Next we bound of $\|J(y(x^l)) - \hat{J}^{l+1}\|^2$.

**Lemma 9.** *Assume assumptions in Lemma 8 hold. Denote $\tau = 1 - (1 + c_J^2)(1 + d_J^2)(1 - \gamma\mu_G)^2$ with $d_J \in (0, \sqrt{\frac{1}{(1-\gamma\mu_G)^2} - 1})$ and $c_J \in (0, \sqrt{\frac{1}{(1+d_J^2)(1-\gamma\mu_G)^2} - 1})$. Suppose $\gamma \in (0, \sqrt{\frac{1}{\mu_G}})$. Denote $C_x := 3(1 + c_J^2)(1 + d_J^{-2})(1 - \gamma\mu_G)^2 L_J^2 + \left(4(1 + c_J^{-2})L_{G_x^y}^2 + 6(1 + c_J^2)(1 + d_J^{-2})L_{G_y^y}^2 M_J^2\right)L_y^2$. Denote $C_y := 4(1 + c_J^{-2})L_{G_x^y}^2 + 6(1 + c_J^2)(1 + d_J^{-2})L_{G_y^y}^2 M_J^2$. Let $\iota \in (0, \tau)$. Then, $\iota \in (0, 1)$ and it holds that*

$$\mathbb{E}_{R^l}\|J(y(x^l)) - \hat{J}^{l+1}\|^2 \leq C_y\gamma^2\mathbb{E}_{R^{l-1}}\|y(x^{l-1}) - y^l\|^2 + C_x\gamma^2\mathbb{E}_{R^{l-1}}\|x^l - x^{l-1}\|^2$$
$$+ \Delta_H + (1 - \iota)\mathbb{E}_{R^{l-1}}\|J(y(x^{l-1})) - \hat{J}^l\|^2 + \Delta_L, \tag{41}$$

where $\Delta_H = 2(1+c_J^{-2})\gamma^2\left(2\sigma_{xy}^2 + \Delta_{xy}\right)$ and $\Delta_L := 6(1+c_J^2)(1+d_J^{-2})\gamma^2\left(2\sigma_{yy} + \Delta_{yy}\right)M_J^2$ when $\hat{\nabla}_{xy}G(x^l, y^l)$ and $\hat{\nabla}_{yy}G(x^l, y^l)$ are generated by (ii) of Step 4 of Algorithm 3, $\gamma < \sqrt{\frac{\tau-\iota}{W}}$ with $W := 6(1+c_J^2)(1+d_J^{-2})(2\sigma_{yy}^2 + \Delta_{yy})$.

*Proof.* First, using the facts that $J(y(x^l)) = J(y(x^l))(I - \gamma\nabla_{yy}G(x, y(x))) - \gamma\nabla_{xy}G(x, y(x^l))$ and $\hat{J}^{l+1} = \hat{J}^l\left(I - \gamma\hat{\nabla}_{yy}G(x^l, y^l)\right) - \gamma\hat{\nabla}_{xy}G(x^l, y^l)$, we have that

$$
\begin{aligned}
&\|J(y(x^l)) - \hat{J}^{l+1}\|^2 \\
&\leq (1+c_J^{-2})\gamma^2\|\nabla_{xy}G(x^l, y(x^l)) - \hat{\nabla}_{xy}G(x^l, y^l)\|^2 \\
&\quad + (1+c_J^2)\|J(y(x^l))(I - \gamma\nabla_{yy}G(x^l, y(x^l))) - \hat{J}^l\left(I - \gamma\hat{\nabla}_{yy}G(x^l, y^l)\right)\|^2 \\
&\leq 2(1+c_J^{-2})\gamma^2\|\nabla_{xy}G(x^l, y(x^l)) - \nabla_{xy}G(x^l, y^l)\| \\
&\quad + 2(1+c_J^{-2})\gamma^2\|\nabla_{xy}G(x^l, y^l) - \hat{\nabla}_{xy}G(x^l, y^l)\|^2 \\
&\quad + (1+c_J^2)\|J(y(x^l))(I - \gamma\nabla_{yy}G(x^l, y(x^l))) - \hat{J}^l\left(I - \gamma\hat{\nabla}_{yy}G(x^l, y^l)\right)\|^2 \\
&\stackrel{(a)}{\leq} 2(1+c_J^{-2})\gamma^2 L_{G_x^y}^2\|y(x^l) - y^l\|^2 + 2(1+c_J^{-2})\gamma^2\|\nabla_{xy}G(x^l, y^l) - \hat{\nabla}_{xy}G(x^l, y^l)\|^2 \\
&\quad + (1+c_J^2)\|J(y(x^l))(I - \gamma\nabla_{yy}G(x^l, y(x^l))) - \hat{J}^l\left(I - \gamma\hat{\nabla}_{yy}G(x^l, y^l)\right)\|^2,
\end{aligned}
\tag{42}
$$

where (a) is thanks to Assumption 2(iii). Note that

$$
\begin{aligned}
&\|J(y(x^l))(I - \gamma\nabla_{yy}G(x^l, y(x^l))) - \hat{J}^l\left(I - \gamma\hat{\nabla}_{yy}G(x^l, y^l)\right)\|^2 \\
&\leq 3(1+d_J^{-2})\|J(y(x^l))(I - \gamma\nabla_{yy}G(x^l, y(x^l))) - J(y(x^l))(I - \gamma\nabla_{yy}G(x^l, y^l)))\|^2 \\
&\quad + 3(1+d_J^{-2})\|J(y(x^l))(I - \gamma\nabla_{yy}G(x^l, y^l))) - J(y(x^{l-1}))(I - \gamma\nabla_{yy}G(x^l, y^l)))\|^2 \\
&\quad + (1+d_J^2)\|J(y(x^{l-1}))(I - \gamma\nabla_{yy}G(x^l, y^l))) - \hat{J}^l(I - \gamma\nabla_{yy}G(x^l, y^l)))\|^2 \\
&\quad + 3(1+d_J^{-2})\|\hat{J}^l(I - \gamma\nabla_{yy}G(x^l, y^l))) - \hat{J}^l\left(I - \gamma\hat{\nabla}_{yy}G(x^l, y^l)\right)\|^2 \\
&\leq 3(1+d_J^{-2})\gamma^2\|\nabla_{yy}G(x^l, y(x^l)) - \nabla_{yy}G(x^l, y^l)\|^2\|J(y(x^l))\|^2 \\
&\quad + 3(1+d_J^{-2})\|I - \gamma\nabla_{yy}G(x^l, y^l)\|^2\|J(y(x^l)) - J(y(x^{l-1}))\|^2 \\
&\quad + (1+d_J^2)\|I - \gamma\nabla_{yy}G(x^l, y^l)\|^2\|J(y(x^{l-1})) - \hat{J}^l\|^2 \\
&\quad + 3(1+d_J^{-2})\gamma^2\|\nabla_{yy}G(x^l, y^l) - \hat{\nabla}_{yy}G(x^l, y^l)\|^2\|\hat{J}^l\|^2,
\end{aligned}
$$

where the first inequality uses $\|a_1 + a_2 + a_3 + a_4\|^2 \leq (1+d_J^2)a_1^2 + (1+d_J^{-2})\|a_2 + a_3 + a_4\|^2 \leq (1+d_J^2)a_1^2 + 3(1+d_J^{-2})\|a_2\|^2 + 3(1+d_J^{-2})\|a_3\|^2 + 3(1+d_J^{-2})\|a_4\|^2$. Using Assumption 2 (ii), the above inequality can be further passed to

$$
\begin{aligned}
&\|J(y(x^l))(I - \gamma\nabla_{yy}G(x^l, y(x^l))) - \hat{J}^l\left(I - \gamma\hat{\nabla}_{yy}G(x^l, y^l)\right)\|^2 \\
&\leq 3(1+d_J^{-2})\gamma^2\|\nabla_{yy}G(x^l, y(x^l)) - \nabla_{yy}G(x^l, y^l)\|^2\|J(y(x^l))\|^2 \\
&\quad + 3(1+d_J^{-2})(1 - \gamma\mu_G)^2\|J(y(x^l)) - J(y(x^{l-1}))\|^2 + (1+d_J^2)(1 - \gamma\mu_G)^2\|J(y(x^{l-1})) - \hat{J}^l\|^2 \\
&\quad + 3(1+d_J^{-2})\gamma^2\|\nabla_{yy}G(x^l, y^l) - \hat{\nabla}_{yy}G(x^l, y^l)\|^2\|\hat{J}^l\|^2 \\
&\leq 3(1+d_J^{-2})\gamma^2 L_{G_y^y}^2 M_J^2\|y(x^l) - y^l\|^2 + 3(1+d_J^{-2})(1 - \gamma\mu_G)^2 L_J^2\|x^l - x^{l-1}\|^2 \\
&\quad + (1+d_J^2)(1 - \gamma\mu_G)^2\|J(y(x^{l-1})) - \hat{J}^l\|^2 + 3(1+d_J^{-2})\gamma^2\|\nabla_{yy}G(x^l, y^l) - \hat{\nabla}_{yy}G(x^l, y^l)\|^2\|\hat{J}^l\|^2,
\end{aligned}
\tag{43}
$$

where the second inequality follows from Assumption 2 (iii) and Lemma 2.

Plugging (43) into (42), we have that

$$
\begin{aligned}
\|J(y(x^l)) - \hat{J}^{l+1}\|^2 &\leq 2(1+c_J^{-2})\gamma^2 L_{G_x^y}^2 \|y(x^l) - y^l\|^2 + 2(1+c_J^{-2})\gamma^2 \|\nabla_{xy}G(x^l,y^l) - \hat{\nabla}_{xy}G(x^l,y^l)\|^2 \\
&+ 3(1+c_J^2)(1+d_J^{-2})\gamma^2 L_{G_y^y}^2 M_J^2 \|y(x^l) - y^l\|^2 + 3(1+c_J^2)(1+d_J^{-2})(1-\gamma\mu_G)^2 L_J^2 \|x^l - x^{l-1}\|^2 \\
&+ (1+c_J^2)(1+d_J^2)(1-\gamma\mu_G)^2 \|J(y(x^{l-1})) - \hat{J}^l\|^2 \\
&+ 3(1+c_J^2)(1+d_J^{-2})\gamma^2 \|\nabla_{yy}G(x^l,y^l) - \hat{\nabla}_{yy}G(x^l,y^l)\|^2 \|\hat{J}^l\|^2 \\
&\leq \left(4(1+c_J^{-2})L_{G_x^y}^2 + 6(1+c_J^2)(1+d_J^{-2})L_{G_y^y}^2 M_J^2\right)\gamma^2 \|y(x^{l-1}) - y^l\|^2 \\
&+ \left(4(1+c_J^{-2})L_{G_x^y}^2 + 6(1+c_J^2)(1+d_J^{-2})L_{G_y^y}^2 M_J^2\right)\gamma^2 \|y(x^{l-1}) - y(x^l)\|^2 \\
&+ \left(3(1+c_J^2)(1+d_J^{-2})(1-\gamma\mu_G)^2 L_J^2\right) \|x^l - x^{l-1}\|^2 \\
&+ 2(1+c_J^{-2})\gamma^2 \|\nabla_{xy}G(x^l,y^l) - \hat{\nabla}_{xy}G(x^l,y^l)\|^2 \\
&+ (1+c_J^2)(1+d_J^2)(1-\gamma\mu_G)^2 \|J(y(x^{l-1})) - \hat{J}^l\|^2 \\
&+ 3(1+c_J^2)(1+d_J^{-2})\gamma^2 \|\nabla_{yy}G(x^l,y^l) - \hat{\nabla}_{yy}G(x^l,y^l)\|^2 \|\hat{J}^l\|^2,
\end{aligned}
$$

Using Lemma 1, the above inequality can be further passed to

$$
\begin{aligned}
\|J(y(&x^l)) - \hat{J}^{l+1}\|^2 \\
&\leq \left(4(1+c_J^{-2})L_{G_x^y}^2 + 6(1+c_J^2)(1+d_J^{-2})L_{G_y^y}^2 M_J^2\right)\gamma^2 \|y(x^{l-1}) - y^l\|^2 \\
&+ \left(4(1+c_J^{-2})L_{G_x^y}^2 + 6(1+c_J^2)(1+d_J^{-2})L_{G_y^y}^2 M_J^2\right)L_y^2\gamma^2 \|x^{l-1} - x^l\|^2 \\
&+ \left(3(1+c_J^2)(1+d_J^{-2})(1-\gamma\mu_G)^2 L_J^2\right) \|x^l - x^{l-1}\|^2 \\
&+ 2(1+c_J^{-2})\gamma^2 \|\nabla_{xy}G(x^l,y^l) - \hat{\nabla}_{xy}G(x^l,y^l)\|^2 \\
&+ (1+c_J^2)(1+d_J^2)(1-\gamma\mu_G)^2 \|J(y(x^{l-1})) - \hat{J}^l\|^2 \\
&+ 3(1+c_J^2)(1+d_J^{-2})\gamma^2 \|\nabla_{yy}G(x^l,y^l) - \hat{\nabla}_{yy}G(x^l,y^l)\|^2 \|\hat{J}^l\|^2 \\
&= \left(4(1+c_J^{-2})L_{G_x^y}^2 + 6(1+c_J^2)(1+d_J^{-2})L_{G_y^y}^2 M_J^2\right)\gamma^2 \|y(x^{l-1}) - y^l\|^2 + C_x\gamma^2 \|x^l - x^{l-1}\|^2 \\
&+ 2(1+c_J^{-2})\gamma^2 \|\nabla_{xy}G(x^l,y^l) - \hat{\nabla}_{xy}G(x^l,y^l)\|^2 + (1-\tau)\|J(y(x^{l-1})) - \hat{J}^l\|^2 \\
&+ 3(1+c_J^2)(1+d_J^{-2})\gamma^2 \|\nabla_{yy}G(x^l,y^l) - \hat{\nabla}_{yy}G(x^l,y^l)\|^2 \|\hat{J}^l\|^2 \\
&\leq \left(4(1+c_J^{-2})L_{G_x^y}^2 + 6(1+c_J^2)(1+d_J^{-2})L_{G_y^y}^2 M_J^2\right)\gamma^2 \|y(x^{l-1}) - y^l\|^2 + C_x\gamma^2 \|x^l - x^{l-1}\|^2 \\
&+ 2(1+c_J^{-2})\gamma^2 \|\nabla_{xy}G(x^l,y^l) - \hat{\nabla}_{xy}G(x^l,y^l)\|^2 + (1-\tau)\|J(y(x^{l-1})) - \hat{J}^l\|^2 \\
&+ 6(1+c_J^2)(1+d_J^{-2})\gamma^2 \|\nabla_{yy}G(x^l,y^l) - \hat{\nabla}_{yy}G(x^l,y^l)\|^2 \|J(y(x^{l-1})) - \hat{J}^l\|^2 \\
&+ 6(1+c_J^2)(1+d_J^{-2})\gamma^2 \|\nabla_{yy}G(x^l,y^l) - \hat{\nabla}_{yy}G(x^l,y^l)\|^2 \|J(y(x^{l-1}))\|^2 \\
&= \left(4(1+c_J^{-2})L_{G_x^y}^2 + 6(1+c_J^2)(1+d_J^{-2})L_{G_y^y}^2 M_J^2\right)\gamma^2 \|y(x^{l-1}) - y^l\|^2 + C_x\gamma^2 \|x^l - x^{l-1}\|^2 \\
&+ 2(1+c_J^{-2})\gamma^2 \|\nabla_{xy}G(x^l,y^l) - \hat{\nabla}_{xy}G(x^l,y^l)\|^2 \\
&+ \underbrace{\left(1-\tau + 6(1+c_J^2)(1+d_J^{-2})\gamma^2 \|\nabla_{yy}G(x^l,y^l) - \hat{\nabla}_{yy}G(x^l,y^l)\|^2\right)}_{A}\|J(y(x^{l-1})) - \hat{J}^l\|^2 \\
&+ 6(1+c_J^2)(1+d_J^{-2})\|\nabla_{yy}G(x^l,y^l) - \hat{\nabla}_{yy}G(x^l,y^l)\|^2 \|J(y(x^{l-1}))\|^2,
\end{aligned}
\tag{44}
$$

where the equality uses the definitions of $\tau$ and $C_x$. Note that the assumptions on $c_J$, $d_J$ and $\gamma$ give that $\tau \in (0,1)$. Now we show that $A$ satisfies

$$
\mathbb{E}_{u^l|\tilde{R}^l} A \leq 1 - \iota. \tag{45}
$$

Using Lemma 8, we have that

$$
\mathbb{E}_{u^l, \mathcal{S}^l|R^{l-1}} A \leq 1 - \tau + 6(1+c_J^2)(1+d_J^{-2})\gamma^2(2\sigma_{yy}^2 + \Delta_{yy}) = 1 - \tau + \gamma^2 W \leq 1 - \iota,
$$

where the equality uses the definition of $W$ and the second uses the assumption that $\gamma < \sqrt{\frac{\tau-\iota}{W}}$.

Therefore, (45) holds. Combining (45) with (44), we obtain

$$\mathbb{E}_{v^l,u^l,\mathcal{S}^l|R^{l-1}}\|J(y(x^l)) - \hat{J}^{l+1}\|^2$$

$$\leq \left(4(1+c_J^{-2})L_{G_x^y}^2 + 6(1+c_J^2)(1+d_J^{-2})L_{G_y^y}^2 M_J^2\right)\gamma^2\|y(x^{l-1})-y^l\|^2 + C_x\gamma^2\|x^l - x^{l-1}\|^2$$

$$+ 2(1+c_J^{-2})\gamma^2\mathbb{E}_{v^l,u^l,\mathcal{S}^l|R^{l-1}}\|\nabla_{xy}G(x^l,y^l) - \hat{\nabla}_{xy}G(x^l,y^l)\|^2$$

$$+ (1-\iota)\|J(y(x^{l-1})) - \hat{J}^l\|^2$$

$$+ 6(1+c_J^2)(1+d_J^{-2})\mathbb{E}_{v^l,u^l,\mathcal{S}^l|R^{l-1}}\|\nabla_{yy}G(x^l,y^l) - \hat{\nabla}_{yy}G(x^l,y^l)\|^2\|J(y(x^{l-1}))\|^2$$

$$\leq \left(4(1+c_J^{-2})L_{G_x^y}^2 + 6(1+c_J^2)(1+d_J^{-2})L_{G_y^y}^2 M_J^2\right)\gamma^2\|y(x^{l-1})-y^l\|^2 + C_x\gamma^2\|x^l - x^{l-1}\|^2$$

$$+ \Delta_H + (1-\iota)\|J(y(x^{l-1})) - \hat{J}^l\|^2 + \Delta_L,$$

where the last inequality uses Assumption 2 (i), Lemma 8 and the definition of $\Delta_H$ and $\Delta_L$.

Taking expectation on both side with $R^{l-1}$, we obtain

$$\mathbb{E}_{R^l}\|J(y(x^l)) - \hat{J}^{l+1}\|^2 = \mathbb{E}_{R^{l-1}}\mathbb{E}_{v^l,u^l,\mathcal{S}^l|R^{l-1}}\|J(y(x^l)) - \hat{J}^{l+1}\|^2$$

$$\leq \left(4(1+c_J^{-2})L_{G_x^y}^2 + 6(1+c_J^2)(1+d_J^{-2})L_{G_y^y}^2 M_J^2\right)\gamma^2\mathbb{E}_{R^{l-1}}\|y(x^{l-1})-y^l\|^2$$

$$+ C_x\gamma^2\mathbb{E}_{R^{l-1}}\|x^l - x^{l-1}\|^2 + \Delta_H + (1-\iota)\mathbb{E}_{R^{l-1}}\|J(y(x^{l-1})) - \hat{J}^l\|^2 + \Delta_L.$$

$\square$

Before showing the details of Theorem 3. We first give the following lemma that estimates the error between $\nabla f(x^l)$ and $\hat{\nabla}f(x^l)$.

**Lemma 10.** *Suppose assumptions in Lemmas 6 and 9 hold. Then it holds that*

$$\mathbb{E}_{R^l}\|\nabla f(x^l) - \hat{\nabla}f(x^l)\|^2$$

$$\leq \left(4(L_{12}^F)^2 + 4M_J^2(L_{22}^F)^2 + \frac{16\sigma_B^2 + 16(C_y^F)^2}{\iota}C_y\gamma^2\right)\mathbb{E}_{\tilde{R}^l}\|y(x^l) - y^{l+1}\|^2$$

$$+ \frac{16\sigma_B^2 + 16(C_y^F)^2}{\iota}C_x\gamma^2\mathbb{E}_{R^l}\|x^{l+1} - x^l\|^2$$

$$+ \frac{16\sigma_B^2 + 16(C_y^F)^2}{\iota}\Delta^l + \frac{16\sigma_B^2 + 16(C_y^F)^2}{\iota}(\Delta_H + \Delta_L) + (4 + 8M_J^2)\sigma_B^2,$$

*where $\Delta^l = \mathbb{E}_{R^l}\|J(y(x^l)) - \hat{J}^{l+1}\|^2 - \mathbb{E}_{R^{l+1}}\|J(y(x^{l+1})) - \hat{J}^{l+2}\|^2$.*

*Proof.* First, using chain rule we have that

$$\nabla f(x^l) = \nabla_x F(x^l, y(x^l)) + J^T(y(x^l))\nabla_y F(x^l, y(x^l)).$$

This together with the definition of $\hat{\nabla}f(x^l)$, we have that

$$\mathbb{E}_{R^l}\|\nabla f(x^l) - \hat{\nabla}f(x^l)\|^2$$

$$\leq 2\mathbb{E}_{R^l}\|\nabla_x F(x^l, y(x^l)) - \nabla_x F(x^l, y^{l+1}; \mathcal{B}^l)\|^2 \qquad (46)$$

$$+ 2\mathbb{E}_{R^l}\|J^T(x^l)\nabla_y F(x^l, y(x^l)) - (\hat{J}^{l+1})^T\nabla_y F(x^l, y^{l+1}; \mathcal{B}^l)\|^2.$$

For the first term in (46), we have that

$$\mathbb{E}_{R^l}\|\nabla_x F(x^l, y(x^l)) - \nabla_x F(x^l, y^{l+1}; \mathcal{B}^l)\|^2$$

$$\leq 2\mathbb{E}_{\tilde{R}^l}\|\nabla_x F(x^l, y(x^l)) - \nabla_x F(x^l, y^{l+1})\|^2 + 2\mathbb{E}_{\tilde{R}^l}\mathbb{E}_{\mathcal{B}^l|\tilde{R}^l}\|\nabla_x F(x^l, y^{l+1}) - \nabla_x F(x^l, y^{l+1}; \mathcal{B}^l)\|^2 \qquad (47)$$

$$\leq 2(L_{12}^F)^2\mathbb{E}_{\tilde{R}^l}\|y(x^l) - y^{l+1}\|^2 + 2\sigma_B^2,$$

where the last inequality uses Assumptions 1 (i) and 3.

For the second term in (46), it holds that

$$
\begin{aligned}
&\mathbb{E}_{R^l}\|J(y(x^l))^T\nabla_y F(x^l, y(x^l)) - (\hat{J}^{l+1})^T\nabla_y F(x^l, y^{l+1}; \mathcal{B}^l)\|^2 \\
&\leq 2\mathbb{E}_{\tilde{R}^l}\|J(y(x^l))^T\nabla_y F(x^l, y(x^l)) - J(y(x^l))^T\nabla_y F(x^l, y^{l+1})\|^2 \\
&\quad + 2\mathbb{E}_{R^l}\|J(y(x^l))^T\nabla_y F(x^l, y^{l+1}) - (\hat{J}^{l+1})^T\nabla_y F(x^l, y^{l+1}; \mathcal{B}^l)\|^2 \\
&\leq 2\mathbb{E}_{\tilde{R}^l}\|J(y(x^l))^T\nabla_y F(x^l, y(x^l)) - J(y(x^l))^T\nabla_y F(x^l, y^{l+1})\|^2 \\
&\quad + 4\mathbb{E}_{U^l, V^l, \mathcal{S}_0^l, \mathcal{B}_0^{l-1}}\mathbb{E}_{\mathcal{B}^l|U^l, V^l, \mathcal{S}_0^l, \mathcal{B}_0^{l-1}}\|J(y(x^l))^T\nabla_y F(x^l, y^{l+1}) - J(y(x^l))^T\nabla_y F(x^l, y^{l+1}; \mathcal{B}^l)\|^2 \\
&\quad + 4\mathbb{E}_{U^l, V^l, \mathcal{S}_0^l, \mathcal{B}_0^{l-1}}\mathbb{E}_{\mathcal{B}^l|U^l, V^l, \mathcal{S}_0^l, \mathcal{B}_0^{l-1}}\|J(y(x^l))^T\nabla_y F(x^l, y^{l+1}; \mathcal{B}^l) - (\hat{J}^{l+1})^T\nabla_y F(x^l, y^{l+1}; \mathcal{B}^l)\|^2 \\
&\leq 2\mathbb{E}_{\tilde{R}^l}\|J(y(x^l))^T\nabla_y F(x^l, y(x^l)) - J(y(x^l))^T\nabla_y F(x^l, y^{l+1})\|^2 \\
&\quad + 4\mathbb{E}_{U^l, V^l, \mathcal{S}_0^l, \mathcal{B}_0^{l-1}}\mathbb{E}_{\mathcal{B}^l|U^l, V^l, \mathcal{S}_0^l, \mathcal{B}_0^{l-1}}\|J(y(x^l))^T\nabla_y F(x^l, y^{l+1}) - J(y(x^l))^T\nabla_y F(x^l, y^{l+1}; \mathcal{B}^l)\|^2 \\
&\quad + 4\mathbb{E}_{U^l, V^l, \mathcal{S}_0^l, \mathcal{B}_0^{l-1}}\mathbb{E}_{\mathcal{B}^l|U^l, V^l, \mathcal{S}_0^l, \mathcal{B}_0^{l-1}}\|J(y(x^l))^T - (\hat{J}^{l+1})^T\|^2\|\nabla_y F(x^l, y^{l+1}; \mathcal{B}^l)\|^2 \\
&\overset{(a)}{\leq} 2\mathbb{E}_{\tilde{R}^l}\|J(y(x^l))^T\nabla_y F(x^l, y(x^l)) - J(y(x^l))^T\nabla_y F(x^l, y^{l+1})\|^2 \\
&\quad + 4\mathbb{E}_{U^l, V^l, \mathcal{S}_0^l, \mathcal{B}_0^{l-1}}\mathbb{E}_{\mathcal{B}^l|U^l, V^l, \mathcal{S}_0^l, \mathcal{B}_0^{l-1}}\|J(y(x^l))^T\nabla_y F(x^l, y^{l+1}) - J(y(x^l))^T\nabla_y F(x^l, y^{l+1}; \mathcal{B}^l)\|^2 \\
&\quad + 4\mathbb{E}_{U^l, V^l, \mathcal{S}_0^l, \mathcal{B}_0^{l-1}}\|J(y(x^l))^T - (\hat{J}^{l+1})^T\|^2(2\sigma_B^2 + 2(C_y^F)^2) \\
&\overset{(b)}{\leq} 2\mathbb{E}_{\tilde{R}^l}\|J(y(x^l))^T\|^2(L_{22}^F)^2\|y(x^l) - y^{l+1}\|^2 \\
&\quad + 4\mathbb{E}_{U^l, V^l, \mathcal{S}_0^l, \mathcal{B}_0^{l-1}}\mathbb{E}_{\mathcal{B}^l|U^l, V^l, \mathcal{S}_0^l, \mathcal{B}_0^{l-1}}\|J(y(x^l))^T\nabla_y F(x^l, y^{l+1}) - J(y(x^l))^T\nabla_y F(x^l, y^{l+1}; \mathcal{B}^l)\|^2 \\
&\quad + 4\mathbb{E}_{U^l, V^l, \mathcal{S}_0^l, \mathcal{B}_0^{l-1}}\|J(y(x^l))^T - (\hat{J}^{l+1})^T\|^2(2\sigma_B^2 + 2(C_y^F)^2) \\
&\overset{(c)}{\leq} 2\mathbb{E}_{\tilde{R}^l}\|J(y(x^l))^T\|^2(L_{22}^F)^2\|y(x^l) - y^{l+1}\|^2 + 4\mathbb{E}_{U^l, V^l, \mathcal{S}_0^l, \mathcal{B}_0^{l-1}}\|J(y(x^l))^T\|^2\sigma_B^2 \\
&\quad + 4\mathbb{E}_{U^l, V^l, \mathcal{S}_0^l, \mathcal{B}_0^{l-1}}\|J(y(x^l))^T - (\hat{J}^{l+1})^T\|^2(2\sigma_B^2 + 2(C_y^F)^2) \\
&\leq 2M_J^2(L_{22}^F)^2\mathbb{E}_{\tilde{R}^l}\|y(x^l) - y^{l+1}\|^2 + 4M_J^2\sigma_B^2 \\
&\quad + 4\mathbb{E}_{U^l, V^l, \mathcal{S}_0^l, \mathcal{B}_0^{l-1}}\|J(y(x^l))^T - (\hat{J}^{l+1})^T\|^2(2\sigma_B^2 + 2(C_y^F)^2) \\
&= 2M_J^2(L_{22}^F)^2\mathbb{E}_{\tilde{R}^l}\|y(x^l) - y^{l+1}\|^2 + 4M_J^2\sigma_B^2 \\
&\quad + 4\mathbb{E}_{R^l}\|J(y(x^l))^T - (\hat{J}^{l+1})^T\|^2(2\sigma_B^2 + 2(C_y^F)^2),
\end{aligned}
\tag{48}
$$

where (a) is thanks to Assumptions 1 and 3, (b) is thanks to Assumption 1, (c) uses assumption 3, and the last inequality is thanks to Lemma 2.

Now using Lemma 9 and the definition of $\Delta^l$, (48) can be further passed to

$$
\begin{aligned}
&\mathbb{E}_{R^l}\|J(y(x^l))^T\nabla_y F(x^l, y(x^l)) - (\hat{J}^{l+1})^T\nabla_y F(x^l, y^{l+1}; \mathcal{B}^l)\|^2 \\
&\leq 2M_J^2(L_{22}^F)^2\mathbb{E}_{\tilde{R}^l}\|y(x^l) - y^{l+1}\|^2 + 4M_J^2\sigma_B^2 + \frac{8\sigma_B^2 + 8(C_y^F)^2}{\iota}C_y\gamma^2\mathbb{E}_{R^l}\|y(x^l) - y^{l+1}\|^2 \\
&\quad + \frac{8\sigma_B^2 + 8(C_y^F)^2}{\iota}C_x\gamma^2\mathbb{E}_{R^l}\|x^{l+1} - x^l\|^2 + \frac{8\sigma_B^2 + 8(C_y^F)^2}{\iota}\Delta^l + \frac{8\sigma_B^2 + 8(C_y^F)^2)}{\iota}(\Delta_H + \Delta_L).
\end{aligned}
$$

Combining this with (46) and (47), we have that

$$
\begin{aligned}
\mathbb{E}_{R^l} &\|\nabla f(x^l) - \hat{\nabla} f(x^l)\|^2 \\
&\leq 4(L_{12}^F)^2 \mathbb{E}_{\tilde{R}^l}\|y(x^l) - y^{l+1}\|^2 + 4\sigma_B^2 + 4M_J^2(L_{22}^F)^2 \mathbb{E}_{\tilde{R}^l}\|y(x^l) - y^{l+1}\|^2 + 8M_J^2\sigma_B^2 \\
&+ \frac{16\sigma_B^2 + 16(C_y^F)^2}{\iota}C_y\gamma^2 \mathbb{E}_{\tilde{R}^l}\|y(x^l) - y^{l+1}\|^2 + \frac{16\sigma_B^2 + 16(C_y^F)^2}{\iota}C_x\gamma^2 \mathbb{E}_{R^l}\|x^{l+1} - x^l\|^2 \\
&+ \frac{16\sigma_B^2 + 16(C_y^F)^2}{\iota}\Delta^l + \frac{16\sigma_B^2 + 16(C_y^F)^2}{\iota}(\Delta_H + \Delta_L) \\
&= \left(4(L_{12}^F)^2 + 4M_J^2(L_{22}^F)^2 + \frac{16\sigma_B^2 + 16(C_y^F)^2}{\iota}C_y\gamma^2\right)\mathbb{E}_{\tilde{R}^l}\|y(x^l) - y^{l+1}\|^2 \\
&+ \frac{16\sigma_B^2 + 16(C_y^F)^2}{\iota}C_x\gamma^2 \mathbb{E}_{R^l}\|x^{l+1} - x^l\|^2 \\
&+ \frac{16\sigma_B^2 + 16(C_y^F)^2}{\iota}\Delta^l + \frac{16\sigma_B^2 + 16(C_y^F)^2}{\iota}(\Delta_H + \Delta_L) + 4\sigma_B^2 + 8M_J^2\sigma_B^2.
\end{aligned}
$$

$\square$

Now we the details of Theorem 3.

**Theorem 5.** *Suppose assumptions in Lemmas 6 and 9 hold. Denote*

$$
D_x := \frac{1}{2\beta} - \beta\frac{8\sigma_B^2 + 8(C_y^F)^2}{\iota}C_x\gamma^2 - \frac{L_f}{2}
$$

*and*

$$
D_y := \frac{\beta}{2}\left(4(L_{12}^F)^2 + 4M_J^2(L_{22}^F)^2 + \frac{16\sigma_B^2 + 16(C_y^F)^2}{\iota}C_y\gamma^2\right).
$$

*Suppose $\beta$ is small enough such that $D_x - \frac{(1+d_{y_s}^{-2})D_y(1-\tilde{\zeta}_s)}{\zeta_s}L_y^2 \geq 0$. Then*

$$
\begin{aligned}
\frac{1}{N+1}\sum_{l=0}^{N}\mathbb{E}\|\nabla f(x^l)\|^2 &\leq \frac{1}{N+1}\frac{2}{\beta}\left(f(x^0) - f^* + (C_y^F)^2\right) \\
&+ \frac{1}{N+1}\left(\frac{2D_y}{\zeta_s\beta}(1-\zeta_s) + \frac{2(8\sigma_B^2 + 8(C_y^F)^2)}{\iota}C_y\gamma^2\right)\|y(x^0) - y^0\|^2 \\
&+ \frac{1}{N+1}\frac{2D_y}{\zeta_s\beta}\frac{2c^{-2}}{(\frac{\mu_G}{2} + \frac{1}{2\alpha})}\sigma_G^2 + \frac{1}{N+1}\frac{2(8\sigma_B^2 + 8(C_y^F)^2)}{\iota}\|J(y(x^{-1})) - \hat{J}^0\|^2 \\
&+ \frac{N+2}{N+1}\frac{16\sigma_B^2 + 16(C_y^F)^2}{\iota}(\Delta_H + \Delta_L) + 4\sigma_B^2 + 8M_J^2\sigma_B^2 + \frac{1}{\zeta_s}\frac{4D_yc^{-2}}{\frac{\mu_G\beta}{2} + \frac{\beta}{2\alpha}}\sigma_G^2.
\end{aligned}
$$

*Proof.* Thanks to Lemma 1, we have that

$$
\begin{aligned}
f(x^{l+1}) &\leq f(x^l) + \langle\nabla f(x^l), x^{l+1} - x^l\rangle + \frac{L_f}{2}\|x^{l+1} - x^l\|^2 \\
&= f(x^l) - \beta\left\langle\nabla f(x^l), \hat{\nabla} f(x^l)\right\rangle + \frac{L_f}{2}\|x^{l+1} - x^l\|^2 \\
&= f(x^l) - \beta\left(\frac{1}{2}\|\nabla f(x^l)\|^2 + \frac{1}{2}\|\hat{\nabla} f(x^l)\|^2 - \frac{1}{2}\|\nabla f(x^l) - \hat{\nabla} f(x^l)\|^2\right) + \frac{L_f}{2}\|x^{l+1} - x^l\|^2 \\
&= f(x^l) - \frac{1}{2}\beta\|\nabla f(x^l)\|^2 + \frac{1}{2}\beta\|\nabla f(x^l) - \hat{\nabla} f(x^l)\|^2 + \left(\frac{L_f}{2} - \frac{1}{2\beta}\right)\|x^{l+1} - x^l\|^2,
\end{aligned}
$$

where the first equality and last equality use Step 7 of Algorithm 3. Taking expectation w.r.t $R^l$, we have that

$$\mathbb{E}_{R^l} f(x^{l+1}) \leq \mathbb{E}_{R^{l-1}} f(x^l) - \frac{1}{2}\beta\mathbb{E}_{R^{l-1}}\|\nabla f(x^l)\|^2 + \mathbb{E}_{R^l}\frac{1}{2}\beta\|\nabla f(x^l) - \hat{\nabla} f(x^l)\|^2$$
$$+ \left(\frac{L_f}{2} - \frac{1}{2\beta}\right)\mathbb{E}_{R^l}\|x^{l+1} - x^l\|^2.$$

Using Lemma 10, the above inequality can be further passed to

$$\mathbb{E}_{R^l} f(x^{l+1}) \leq \mathbb{E}_{R^{l-1}} f(x^l) - \frac{1}{2}\beta\mathbb{E}_{R^{l-1}}\|\nabla f(x^l)\|^2$$
$$+ \frac{\beta}{2}\left(4(L_{12}^F)^2 + 4M_J^2(L_{22}^F)^2 + \frac{16\sigma_B^2 + 16(C_y^F)^2}{\iota}C_y\gamma^2\right)\mathbb{E}_{\tilde{R}^l}\|y(x^l) - y^{l+1}\|^2$$
$$+ \beta\frac{8\sigma_B^2 + 8(C_y^F)^2}{\iota}C_x\gamma^2\mathbb{E}_{R^l}\|x^{l+1} - x^l\|^2$$
$$+ \beta\frac{8\sigma_B^2 + 8(C_y^F)^2}{\iota}\Delta^l + \frac{(8\sigma_B^2 + 8(C_y^F)^2)\beta}{\iota}(\Delta_H + \Delta_L) + (2 + 4M_J^2)\beta\sigma_B^2$$
$$+ \left(\frac{L_f}{2} - \frac{1}{2\beta}\right)\mathbb{E}_{R^l}\|x^{l+1} - x^l\|^2$$
$$= \mathbb{E}_{R^{l-1}} f(x^l) - \frac{1}{2}\beta\mathbb{E}_{R^{l-1}}\|\nabla f(x^l)\|^2 + D_y\mathbb{E}_{\tilde{R}^l}\|y(x^l) - y^{l+1}\|^2 - D_x\mathbb{E}_{R^l}\|x^{l+1} - x^l\|^2$$
$$+ \beta\frac{8\sigma_B^2 + 8(C_y^F)^2}{\iota}\Delta^l + \frac{(8\sigma_B^2 + 8(C_y^F)^2)\beta}{\iota}(\Delta_H + \Delta_L) + (2 + 4M_J^2)\beta\sigma_B^2,$$

where the last inequality uses the definition of $D_x$ and $D_y$.

Using (33), the above inequality can be further passed to

$$\mathbb{E}_{R^l} f(x^{l+1}) \leq \mathbb{E}_{R^{l-1}} f(x^l) - \frac{1}{2}\beta\mathbb{E}_{R^{l-1}}\|\nabla f(x^l)\|^2 + \frac{D_y}{\zeta_s}\left(\mathbb{E}_{\tilde{R}^l}\|y(x^l) - y^{l+1}\|^2 - \mathbb{E}_{\tilde{R}^{l+1}}\|y^{l+2} - y(x^{l+1})\|^2\right)$$
$$+ \frac{(1 + d_{y_s}^{-2})D_y(1 - \tilde{\zeta}_s)}{\zeta_s}L_y^2\mathbb{E}_{R^l}\|x^{l+1} - x^l\|^2 + \frac{D_y}{\zeta_s}\frac{2c^{-2}}{\frac{\mu_G}{2} + \frac{1}{2\alpha}}\sigma_G^2 - D_x\mathbb{E}_{R^l}\|x^{l+1} - x^l\|^2$$
$$+ \beta\frac{8\sigma_B^2 + 8(C_y^F)^2}{\iota}\Delta^l + \frac{(8\sigma_B^2 + 8(C_y^F)^2)\beta}{\iota}(\Delta_H + \Delta_L) + (2 + 4M_J^2)\beta\sigma_B^2$$
$$= \mathbb{E}_{R^{l-1}} f(x^l) - \frac{1}{2}\beta\mathbb{E}_{R^{l-1}}\|\nabla f(x^l)\|^2 + \frac{D_y}{\zeta_s}\left(\mathbb{E}_{\tilde{R}^l}\|y(x^l) - y^{l+1}\|^2 - \mathbb{E}_{\tilde{R}^{l+1}}\|y^{l+2} - y(x^{l+1})\|^2\right)$$
$$- \left(D_x - \frac{(1 + d_{y_s}^{-2})D_y(1 - \tilde{\zeta}_s)}{\zeta_s}L_y^2\right)\mathbb{E}_{R^l}\|x^{l+1} - x^l\|^2 + \frac{D_y}{\zeta_s}\frac{2c^{-2}}{\frac{\mu_G}{2} + \frac{1}{2\alpha}}\sigma_G^2$$
$$+ \beta\frac{8\sigma_B^2 + 8(C_y^F)^2}{\iota}\Delta^l + \frac{(8\sigma_B^2 + 8(C_y^F)^2)\beta}{\iota}(\Delta_H + \Delta_L) + (2 + 4M_J^2)\beta\sigma_B^2.$$

Thanks to the assumption that $\beta$ is small enough such that $D_x - \frac{(1+d_{y_s}^{-2})D_y(1-\tilde{\zeta}_s)}{\zeta_s}L_y^2 \geq 0$, the above inequality can be further passed to

$$\mathbb{E}_{R^l} f(x^{l+1}) \leq \mathbb{E}_{R^{l-1}} f(x^l) - \frac{1}{2}\beta\mathbb{E}_{R^{l-1}}\|\nabla f(x^l)\|^2 + \frac{D_y}{\zeta_s}\left(\mathbb{E}_{\tilde{R}^l}\|y(x^l) - y^{l+1}\|^2 - \mathbb{E}_{\tilde{R}^{l+1}}\|y^{l+2} - y(x^{l+1})\|^2\right)$$
$$+ \frac{D_y}{\zeta_s}\frac{2c^{-2}}{\frac{\mu_G}{2} + \frac{1}{2\alpha}}\sigma_G^2 + \beta\frac{8\sigma_B^2 + 8(C_y^F)^2}{\iota}\Delta^l + \frac{(8\sigma_B^2 + 8(C_y^F)^2)\beta}{\iota}(\Delta_H + \Delta_L) + (2 + 4M_J^2)\beta\sigma_B^2$$

Rearranging the above inequality, it holds that

$$\mathbb{E}_{R^{l-1}}\|\nabla f(x^l)\|^2 \leq \frac{2}{\beta}\left(\mathbb{E}_{R^{l-1}}f(x^l) - \mathbb{E}_{R^l}f(x^{l+1})\right) + \frac{2D_y}{\zeta_s\beta}\left(\mathbb{E}_{\tilde{R}^l}\|y(x^l) - y^{l+1}\|^2 - \mathbb{E}_{\tilde{R}^{N+1}}\|y^{l+2} - y(x^{l+1})\|^2\right)$$
$$+ 2\frac{8\sigma_B^2 + 8(C_y^F)^2}{\iota}\Delta^l + \frac{16\sigma_B^2 + 16(C_y^F)^2}{\iota}(\Delta_H + \Delta_L) + 4\sigma_B^2 + 8M_J^2\sigma_B^2 + \frac{1}{\zeta_s}\frac{4D_y c^{-2}}{\frac{\mu_G\beta}{2} + \frac{\beta}{2\alpha}}\sigma_G^2.$$

Taking expectation on $R^l$ and $\tilde{R}^l$, we have that

$$\mathbb{E}\|\nabla f(x^l)\|^2 \leq \frac{2}{\beta}\left(\mathbb{E}f(x^l) - \mathbb{E}f(x^{l+1})\right) + \frac{2D_y}{\zeta_s\beta}\left(\mathbb{E}\|y(x^l) - y^{l+1}\|^2 - \mathbb{E}\|y^{l+2} - y(x^{l+1})\|^2\right)$$
$$+ 2\frac{8\sigma_B^2 + 8(C_y^F)^2}{\iota}\Delta^l + \frac{16\sigma_B^2 + 16(C_y^F)^2}{\iota}(\Delta_H + \Delta_L) + 4\sigma_B^2 + 8M_J^2\sigma_B^2 + \frac{1}{\zeta_s}\frac{4D_y c^{-2}}{\frac{\mu_G\beta}{2} + \frac{\beta}{2\alpha}}\sigma_G^2.$$

Summing the above inequality from $l = 0$ to $N$, we have that

$$\begin{aligned}
\sum_{l=0}^{N}\mathbb{E}\|\nabla f(x^l)\|^2 &\leq \frac{2}{\beta}\left(\mathbb{E}f(x^0) - \mathbb{E}f(x^{N+1})\right) + \frac{2D_y}{\zeta_s\beta}\left(\mathbb{E}\|y(x^0) - y^1\|^2 - \mathbb{E}\|y^{N+2} - y(x^{N+1})\|^2\right) \\
&\quad + \frac{16\sigma_B^2 + 16(C_y^F)^2}{\iota}\left(\mathbb{E}\|J(y(x^0)) - \hat{J}^1\|^2 - \mathbb{E}\|J(y(x^{N+1})) - \hat{J}^{N+2}\|^2\right) \\
&\quad + (N+1)\frac{16\sigma_B^2 + 16(C_y^F)^2}{\iota}(\Delta_H + \Delta_L) + 4(N+1)\sigma_B^2 + 8(N+1)M_J^2\sigma_B^2 \\
&\quad + \frac{N+1}{\zeta_s}\frac{4D_y c^{-2}}{\frac{\mu_G\beta}{2} + \frac{\beta}{2\alpha}}\sigma_G^2 \\
&\leq \frac{2}{\beta}\left(f(x^0) - \mathbb{E}f(x^{N+1})\right) + \frac{2D_y}{\zeta_s\beta}\mathbb{E}\|y(x^0) - y^1\|^2 \\
&\quad + \frac{16\sigma_B^2 + 16(C_y^F)^2}{\iota}\mathbb{E}\|J(y(x^0)) - \hat{J}^1\|^2 \\
&\quad + (N+1)\frac{16\sigma_B^2 + 16(C_y^F)^2}{\iota}(\Delta_H + \Delta_L) + 4(N+1)\sigma_B^2 + 8(N+1)M_J^2\sigma_B^2 \\
&\quad + \frac{N+1}{\zeta_s}\frac{4D_y c^{-2}}{\frac{\mu_G\beta}{2} + \frac{\beta}{2\alpha}}\sigma_G^2.
\end{aligned} \tag{49}$$

Now we define $x^{-2} = x^{-1} = x^0$ and $y^{-1} = y^0$. Let $\{u_1^{-1}, \ldots, u_Q^{-1}, u_1^{-2}, \ldots, u_Q^{-2}\}$ be any vectors in $\mathbb{R}^n$ and $\{v_1^{-1}, \ldots, v_Q^{-1}, v_1^{-2}, \ldots, v_Q^{-2}\}$ be any vectors in $\mathbb{R}^m$. Let $\mathcal{S}^{-1}, \mathcal{S}^{-2}, \mathcal{B}^{-1}, \mathcal{B}^{-2}$ be full batches. Then since $\{x^{-1}, x^{-2}\}$ are deterministic, using (41), using Lemma 1, we have that

$$\mathbb{E}\|\nabla f(x^{-2})\|^2 = \|\nabla f(x^{-1})\|^2 \leq (C_y^F)^2.$$

Using this together, (41) with $l = 1$ and the fact that $x^{-1} = x^0$ and $y^{-1} = y^0$, we have that

$$\begin{aligned}
\mathbb{E}\|\nabla f(x^{-1})\|^2 &+ \frac{2(8\sigma_B^2 + 8(C_y^F)^2)}{\iota}\mathbb{E}\|J(y(x^0)) - \hat{J}^1\|^2 \leq (C_y^F)^2 + \frac{2(8\sigma_B^2 + 8(C_y^F)^2)}{\iota}\left(C_y\gamma^2\right)\|y(x^{-1}) - y^0\|^2 \\
&+ \frac{8(2\sigma_B^2 + 2(C_y^F)^2)(1-\iota)}{\iota}\mathbb{E}\|J(y(x^{-1})) - \hat{J}^0\|^2 + \frac{2(8\sigma_B^2 + 8(C_y^F)^2)}{\iota}(\Delta_H + \Delta_L) \\
&\leq (C_y^F)^2 + \frac{2(8\sigma_B^2 + 8(C_y^F)^2)}{\iota}\left(C_y\gamma^2\right)\|y(x^{-1}) - y^0\|^2 \\
&+ \frac{2(8\sigma_B^2 + 8(C_y^F)^2)}{\iota}\|J(y(x^{-1})) - \hat{J}^0\|^2 + \frac{2(8\sigma_B^2 + 8(C_y^F)^2)}{\iota}(\Delta_H + \Delta_L),
\end{aligned}$$

where the last inequality follows from the fact that $\iota \in (0,1)$ and the fact that $\hat{J}^0$ is deterministic. This implies

$$
\begin{aligned}
\frac{2(8\sigma_B^2 + 8(C_y^F)^2)}{\iota}\mathbb{E}\|J(y(x^0)) - \hat{J}^1\|^2 &\leq (C_y^F)^2 + \frac{2(8\sigma_B^2 + 8(C_y^F)^2)}{\iota}\left(C_y\gamma^2\right)\|y(x^{-1}) - y^0\|^2 \\
&+ \frac{2(8\sigma_B^2 + 8(C_y^F)^2)}{\iota}\Delta^{-1} + \frac{2(8\sigma_B^2 + 8(C_y^F)^2)}{\iota}(\Delta_H + \Delta_L).
\end{aligned}
\tag{50}
$$

On the other hand, using (32), it holds that

$$
\mathbb{E}\|y^1 - y(x^0)\|^2 \leq (1 - \zeta_s)\|y(x^{-1}) - y^0\|^2 + (1 + d_{y_s}^{-2})(1 - \tilde{\zeta}_s)L_y^2\|x^0 - x^{0-1}\|^2 + \frac{2c^{-2}}{\left(\frac{\mu_G}{2} + \frac{1}{2\alpha}\right)}\sigma_G^2.
\tag{51}
$$

Sum (50), (51) and (49), we have that

$$
\begin{aligned}
\sum_{l=0}^{N}\mathbb{E}\|\nabla f(x^l)\|^2 &\leq \frac{2}{\beta}\left(f(x^0) - \mathbb{E}f(x^{N+1})\right) \\
&+ \frac{2D_y}{\zeta_s\beta}\left((1 - \zeta_s)\|y(x^{-1}) - y^0\|^2 + (1 + d_{y_s}^{-2})(1 - \tilde{\zeta}_s)L_y^2\|x^0 - x^{0-1}\|^2 + \frac{2c^{-2}}{\left(\frac{\mu_G}{2} + \frac{1}{2\alpha}\right)}\sigma_G^2\right) \\
&+ (C_y^F)^2 + \frac{2(8\sigma_B^2 + 8(C_y^F)^2)}{\iota}\left(C_y\gamma^2\right)\|y(x^{-1}) - y^0\|^2 + \frac{2(8\sigma_B^2 + 8(C_y^F)^2)}{\iota}\Delta^{-1} + \frac{2(8\sigma_B^2 + 8(C_y^F)^2)}{\iota}(\Delta_H + \Delta_L) \\
&+ (N+1)\frac{16\sigma_B^2 + 16(C_y^F)^2}{\iota}(\Delta_H + \Delta_L) + 4(N+1)\sigma_B^2 + 8(N+1)M_J^2\sigma_B^2 + \frac{N+1}{\zeta_s}\frac{4D_y c^{-2}}{\frac{\mu_G\beta}{2} + \frac{\beta}{2\alpha}}\sigma_G^2.
\end{aligned}
$$

Since $f$ is lower bounded by $f^*$, $x^{-1} = x^0$ and $\Delta^{-1} = \|J(y(x^{-1})) - \hat{J}^0\|^2$ by the definition of $\Delta^l$ in Lemma 10, the above inequality can be further passed to

$$
\begin{aligned}
\sum_{l=0}^{N}\mathbb{E}\|\nabla f(x^l)\|^2 &\leq \frac{2}{\beta}\left(f(x^0) - f^*\right) + \frac{2D_y}{\zeta_s\beta}\left((1 - \zeta_s)\|y(x^0) - y^0\|^2 + \frac{2c^{-2}}{\left(\frac{\mu_G}{2} + \frac{1}{2\alpha}\right)}\sigma_G^2\right) \\
&+ (C_y^F)^2 + \frac{2(8\sigma_B^2 + 8(C_y^F)^2)}{\iota}\left(C_y\gamma^2\right)\|y(x^0) - y^0\|^2 + \frac{2(8\sigma_B^2 + 8(C_y^F)^2)}{\iota}\|J(y(x^{-1})) - \hat{J}^0\|^2 \\
&+ (N+2)\frac{16\sigma_B^2 + 16(C_y^F)^2}{\iota}(\Delta_H + \Delta_L) + 4(N+1)\sigma_B^2 + 8(N+1)M_J^2\sigma_B^2 + \frac{N+1}{\zeta_s}\frac{4D_y c^{-2}}{\frac{\mu_G\beta}{2} + \frac{\beta}{2\alpha}}\sigma_G^2.
\end{aligned}
$$

Dividing the above inequality on both sides by $N+1$, and rearranging terms, we obtain

$$
\begin{aligned}
\frac{1}{N+1}\sum_{l=0}^{N}\mathbb{E}\|\nabla f(x^l)\|^2 &\leq \frac{1}{N+1}\frac{2}{\beta}\left(f(x^0) - f^* + (C_y^F)^2\right) \\
&+ \frac{1}{N+1}\left(\frac{2D_y}{\zeta_s\beta}(1 - \zeta_s) + \frac{2(8\sigma_B^2 + 8(C_y^F)^2)}{\iota}C_y\gamma^2\right)\|y(x^0) - y^0\|^2 \\
&+ \frac{1}{N+1}\frac{2D_y}{\zeta_s\beta}\frac{2c^{-2}}{\left(\frac{\mu_G}{2} + \frac{1}{2\alpha}\right)}\sigma_G^2 + \frac{1}{N+1}\frac{2(8\sigma_B^2 + 8(C_y^F)^2)}{\iota}\|J(y(x^{-1})) - \hat{J}^0\|^2 \\
&+ \frac{N+2}{N+1}\frac{16\sigma_B^2 + 16(C_y^F)^2}{\iota}(\Delta_H + \Delta_L) + 4\sigma_B^2 + 8M_J^2\sigma_B^2 + \frac{1}{\zeta_s}\frac{4D_y c^{-2}}{\frac{\mu_G\beta}{2} + \frac{\beta}{2\alpha}}\sigma_G^2.
\end{aligned}
$$

$\square$

## C  Additional Experimental Details

The code is available in a anonymous way at https://anonymous.4open.science/r/Bilevel_ITD-5F46/. The formulation of the hyper-representation task is as follows:

$$\min_{\lambda} l_{val}\big(\lambda, w^*(\lambda)\big) = \mathbb{E}[l_{val}\big(\lambda, w^*(\lambda)\big); \xi)] = \frac{1}{|D_{\mathcal{V},\xi}|} \sum_{(x_i,y_i)\in D_{\mathcal{V},\xi}} l((\omega_\lambda^*)^T \phi(x_i; \lambda), y_i)$$

$$\text{s.t.} \quad w^*(\lambda) = \arg\min_{w} l_{tr}(\lambda, w) := \frac{1}{|D_{\mathcal{T},\xi}|} \sum_{(x_i,y_i)\in D_{\mathcal{T},\xi}} l(\omega^T \phi(x_i; \lambda), y_i) + C\|w\|^2,$$

where $l(\cdot)$ denotes the cross entropy loss, $D_{\mathcal{T},\xi}$ and $D_{\mathcal{V},\xi}$ are training and validation dataset for a randomly sampled meta task. Here $\lambda = \{\lambda_i\}_{i\in\mathcal{D}_{\mathcal{T}}}$ are hyper-representations and $C \geq 0$ is a tuning parameter to gaurantee the inner problem to be strongly convex. In experiment, we set $C = 0.01$.

The Omniglot dataset includes 1623 characters from 50 different alphabets and each character consists of 20 samples. We follow the experimental protocols of Vinyals et al. (2016) to to divide the alphabets to train/validation/test with 33/5/12, respectively. We perform $N$-way-$K$-shot classification, more specifically, for each task, we randomly sample $N$ characters from the alphabet over that client and for each character, we sample $K$ samples for training and 15 samples for validation. We augment the characters by performing rotation operations ( multipliers of 90 degrees). We use a 4-layer convolutional neural networks and each convolutional layer has 64 filters of 3×3 and is followed by batch-normalization layers Finn et al. (2017b). The parameters of convolutional layers are treated as hyper-representation and the last linear layer is the fined tune inner parameters. For all experiments, we use mini-batch size 4, outer learning rate 0.1, inner learning rate 0.4, perform 4 inner gradient steps. In particular, for F2SA, we set the Lagrange multiplier as 2.

