# OpenReview forum: "Hessian Free Efficient Single Loop Iterative Differentiation Methods for Bi-Level Optimization Problems"
_TMLR — Accepted by TMLR_

### Review · Reviewer_v4oY · 2024-07-16

**Summary Of Contributions:**

This paper proposes a single-loop algorithm to address bilevel problem under a KL property. The algorithm does not require computation of Hessian Inverse or Hessian Vector Multiplication. Convergence analysis and experimental results have supported the efficiency of the proposed algorithms.

**Audience:**

Yes

**Broader Impact Concerns:**

No.

**Claims And Evidence:**

Yes

**Requested Changes:**

See Above.

**Strengths And Weaknesses:**

***Strengths***

1. This submission has provided solutions to avoid computing Hessian Inverse or Hessian Vector Multiplication.

2. Convergence analysis have been established.

3. Experiments looks promising on the considered task.

***Weakness***

I have several questions that the authors could hopefully address in the revision.

1.  The following paper has also proposed stochastic single-loop algorithms (SMB, SBMA) that Neumann series to approximate Hessian Inverse. Thus, can the authors please discuss the difference and novelty of this submission on this matter?  I guess the main difference is that this submission used Gaussian smoothing to get rid of Hessian Vector Multiplication.

Guo, Zhishuai, Yi Xu, Wotao Yin, Rong Jin, and Tianbao Yang. "A novel convergence analysis for algorithms of the adam family and beyond." arXiv preprint arXiv:2104.14840 (2021).

2. The following paper has also leveraged KL property to establish (imply) the sequential convergence and get rid of Hessian Inverse and Hessian Vector Multiplication. Therefore, can the authors please discuss the difference and novelty of this submission on this matter? Is it still safe to claim "As far as we know, this is the first work that provides the convergence rate of the sequences generated by single loop methods."? Please check it out.

Liu, Bo, Mao Ye, Stephen Wright, Peter Stone, and Qiang Liu. "Bome! bilevel optimization made easy: A simple first-order approach." Advances in neural information processing systems 35 (2022): 17248-17262.

3. In experiment section or in the appendix, can the authors discuss more on the hyper-representation task? So the audience who are not familiar with this task, like myself, can understand it more easily. In particular, what are $\lambda$'s.

4. Minor issues:

i) The complexity rate for ES-ITDM has been missed in Table 1. Is it intended?

ii) The first sentence in Section 7 Experiment seems incomplete, do you mean "we test XXX"?

iii) The first problem formulation in Section 7 Experiment has both $\omega^*_\lambda$ and $\omega^*(\lambda)$. You may want to have only one expression for this.

---

> ### Author Response · Authors · 2024-09-24
>
> Response to  weakness 1: Thanks for pointing this out. One difference between our work and the SMB and SBMA methods mentioned in the cited work is that we use Gaussian smoothing to avoid Hessian-vector multiplication. Another difference is that, in SMB or SBMA, they calculate the Neumann series at the same updates $(x^t, y^t)$, while we utilize the previous updates ${(x^0, y^0), \dots, (x^t, y^t)}$. Additionally, the number of series terms increases as the iterations progress.
>
> The motivation for this is that in a single-loop method, the upper-level parameter is only updated once, which means that the error between $y^t$ and the minimizer of the lower-level problem can be large, as can the approximation of the Jacobian by the corresponding Neumann series. However, in a double-loop method, where there are multiple lower-level updates, there is an inductive relation when calculating the Jacobian (please see (4)). Observing this, we propose to update the Jacobian using the inductive relation in (5), resulting in the Jacobian estimation in (6), which resembles the Neumann series. In this way, we incorporate as much information about the lower-level updates as the double-loop method does, while retaining the single-loop structure.
>
> We have now emphasized this point in the contribution section.
>
> Response to  weakness 2:
>  Thank you for pointing this out. It is true that they also investigated the convergence of the generated sequence, but they only showed that the accumulation point is a stationary point. In our analysis, we additionally provide the convergence {\bf rate} of the generated sequence. Under the KL assumption, we show that it can converge linearly, sublinearly, or finitely. We have now included a discussion of the differences between their work and ours in the related work section.
>
> Response to  weakness 3:  Thank you for the comment. The term "Hyper-representation" was used in [Franceschi et al. (2018)] in the early study of bilevel programming. It refers to a shared representation (or shared deep neural network) across multiple tasks in a meta-learning framework. The parameters in the shared representation are referred to as hyperparameters. Let $\phi(\cdot; \lambda)$ denote the hyper-representation mapping, parameterized by $\lambda$. When applying this to solve a specific classification task, a linear layer $w$ is added on top of the hyper-representation, and only the parameters $w$ are trained, while the parameters $\lambda$ in the shared "hyper-representation" remain fixed. To train the parameters in the hyper-representation, [Franceschi et al. (2018)] formulated it as a bilevel programming problem, where the upper-level objective minimizes a validation loss, and the lower-level objective minimizes a task-specific training loss.
>
> We have now added this explanation to the experiment section.
>
> Reference:
>
> Franceschi, L., Frasconi, P., Salzo, S., Grazzi, R., Pontil, M. (2018). Bilevel programming for hyperparameter optimization and meta-learning. In Proceedings of the 35th International Conference on Machine Learning (ICML 2018), Stockholmsmässan, Stockholm, Sweden, July 10-15, 2018.
>
>
>
>
>  Response to weakness 4: Minor issues:
>
> i) Thanks for pointing this out. The complexity of ES-ITDM can be analyzed in the same way as we analyze SES-ITDM with the batch being the full size of the dataset. Since we consider the constant stepsize and batch size in out method, the analysis is exactly the same. We now added the complexity of ES-ITDM in Table 1.
>
> ii)Thanks for pointing this out. Yes we mean "we test xxx". We now have fixed this typo.
>
> iii） Thanks for pointing this out. We now only use the expression $\omega^{*}(\lambda)$ in our formulation.

---

> > ### Comment · Reviewer_v4oY · 2024-10-09
> >
> > Thank you for the clarifications and revision. My concerns have been addressed.

---

> > > ### Author Response · Authors · 2024-10-09
> > >
> > > We sincerely appreciate your time and invaluable feedback. Your suggestions have greatly contributed to improving our manuscript. Thank you for your thoughtful consideration of our work.

---

### Review · Reviewer_XGse · 2024-08-05

**Summary Of Contributions:**

This paper presents convergence results for two single-loop algorithms for bilevel optimization. One algorithm is for deterministic problems, and is proved under a Kurdyka-Łojasiewicz (KL) assumption. The other is for stochastic problems, and does not require the KL assumption. The results are tested on a hyperparameter optimization problem.

**Audience:**

Yes

**Claims And Evidence:**

No

**Requested Changes:**

In my view, Theorem 1 alone is not strong enough for publication.

To merit publication, I would recommend:
- Fix Theorem 3 or remove it.
- Get a version of Theorem 2 that doesn't rely on an uncheckable assumption, or remove it.
- At least one of (the fixed versions of) Theorem 2 or 3 should still be in the paper, even if the other is removed.

**Strengths And Weaknesses:**

# Strengths
- The setup is clear
- The assumptions are well-stated
- The results have the potential to be interesting
- It was tested numerically.

# Weaknesses
- Theorem 3 is incorrect, as we will demonstrate below with a counterexample.
- Theorem 2 appears to be plausible, but it relies on an uncheckable KL assumption,which makes the contribution dubious.
- The writing is sloppy.

## A counterexample for Theorem 3

Let $x$ and $y$ be scalars. Consider the functions
\begin{align*}
F(x,y;\xi)&=F(x,y)=cx\\\\
G(x,y;\eta)&=G(x,y)=y^2
\end{align*}
where $c$ is a constant to be chosen later.

These functions satisfy Assumptions 1, 2, and 3. (Note that the definition of Lipschitz continuity is stated incorrectly in Assumption 1.) We can take $\sigma_B=\sigma_G=0$. Also, $G$ does not depend on $x$, so that $\bar\Delta_{xy}=0$.  It follows that $E_2 = A_2 \bar\Delta_{yy}$.

In this case, Algorithm 3 is just gradient descent, and in particular we have that $\nabla f(x^\ell)=c$ for all $\ell\ge 0$. Then Theorem 3 states that
$$
\frac{1}{N+1}\sum_{\ell=0}^N\\|\nabla f(x^\ell)\\|^2 \le \frac{E_1}{N+1}+E_2,
$$
where $E_1$ is a constant (depending potentially on the initial conditions) and $E_2$ is a constant which does not depend on $f$. In particular, $E_2$ does not depend on the choice of $c$.

But, by our choice of $f$, Theorem 3 would imply that
$$
c^2 \le \frac{E_1}{N+1}+E_2,
$$
which fails for all sufficiently large $N$ and $c$.

## Issues with the KL assumption

Theorem 2 relies on assuming the function $H$ defined in Theorem 1 has the KL property. However, this function depends on the solution to the lower-level minimization, $y(x)$, and the corresponding Jacobian $J(y(x))$. These will typically be very complicated functions. As a result, it is unclear if the KL property could ever actually be checked in an example.

---

> ### Author Response · Authors · 2024-09-24
>
> Responses to weakness 1: Thank you for checking the correctness of Theorem 3. We would like to point out that, although the example you mentioned satisfies Assumptions 1, 2, and 3, it does not satisfy the assumption $f^* > -\infty$ stated in Theorem 3, where $f^*$ is the optimal value of the considered problem. In your example, $\inf_x F(x) = \inf_x cx = -\infty$, which violates this assumption.
>
> However, our conclusion still holds even without the assumption that $f^* > -\infty$. This is because when $f^* = -\infty$, the term $E_1$ on the right-hand side becomes infinite, and thus Theorem 3 holds trivially. Therefore, in either case, Theorem 3 remains valid.
>
> Responses to weakness 2: Thank you for pointing this out. In fact, many functions are KL functions. As mentioned after defining the KL property in Definition 1, it is known that proper closed semi-algebraic functions (i.e., functions whose graphs are unions and intersections of polynomial functions) satisfy the KL property. Semi-algebraic functions already cover widely used losses such as quadratic loss, L2 loss, Huber loss, hinge loss, and 0-1 loss. Therefore, this assumption is not difficult to satisfy.
>
> The definition of the KL property might seem complex, but this is simply because it is a very general property that applies even when smoothness is absent. In our bilevel case, since the lower-level problem is strongly convex, the lower-level minimizer is unique. When the lower-level problem is semi-algebraic, and since $y(x)$ is the infimum projection of a semi-algebraic function, its graph is also the intersection of polynomial functions, making $y(x)$ semi-algebraic, as well as its Jacobian $J(y(x))$. If the upper-level objective is also semi-algebraic, then the potential function $H$ is a KL function, satisfying the assumption in Theorem 2.
>
> In summary, the KL assumption is not difficult to verify but is rather easy to identify. We have now added the above explanation as a remark after Definition 1 and Theorem 2 to make it easier for readers to understand.

---

> > ### Comment · Reviewer_XGse · 2024-10-07
> > **Thank you for the patient explanations!**
> >
> > I would like to thank the authors for patiently explaining my misunderstandings. The authors also did a good job in clarifying my concerns about Theorem 2, both in the response and in the revision.

---

> > > ### Author Response · Authors · 2024-10-08
> > >
> > > We sincerely appreciate your review. Your feedback has been invaluable in improving the manuscript. Thank you for your time and careful consideration of our work.

---

### Review · Reviewer_HiMK · 2024-09-09

**Summary Of Contributions:**

The authors propose a new method to calculate the Jacobian matrix for bi-level optimization problems, which involves historical accumulation and random finite-difference approximation. The authors prove that under KL condition the proposed stochastic algorithm converges to the neighbor of the first-order stationary point. The experiments show the improvement of the proposed algorithm.

**Audience:**

Yes

**Broader Impact Concerns:**

I don't have any concerns.

**Claims And Evidence:**

Yes

**Requested Changes:**

1.  Solve weaknesses or correct me if there is some misunderstanding, especially for 1,2,3,6 and 7.

2. The authors have to go through to correct typos besides the following:

    (i) Page 4 we denote $\nabla_x F(x,y)$..... $\frac{\partial F(x,y)}{\partial y}$. It should be $\frac{\partial F(x,y)}{\partial x}$.

    (ii) At the beginning of Section 6.1. "We first give a decent lemma" and the following is Theorem 1.

    (iii) In Definition 1, $dist(0,\partial f(x)) \geq \cdots$, a right bracket is missing.

    (iv) In Theorem 3, what is the definition of $\bar{\nabla}$?

    (v) The problem formulation in Experiments, both $w^*_\lambda$ and $w^*(\lambda)$ are used.

    (vi) Page 16, the last but the second line of the first inequality part, there is a redundant $||$.

    (vii) Section B.4 there is a strange F.

**Strengths And Weaknesses:**

Strengths:

1. The authors propose a new Jacobian approximation method that works in the bi-level optimization at least for the experiments the authors provide.

2. The proposed algorithm has a convergence guarantee that the algorithm can converge to the neighborhood of the solution.

Weaknesses:
1. Some claims in the paper may not be correct.

    (i) In the introduction when the authors introduce the Neumann series, the claim says that the larger b is, the less error this approximation has. However, although the difference from the expected value of the approximation to the real inverse will be small, since p is randomly sampled from $\{1,2,\cdots,b\}$, the variance of the approximation will increase. The error will not decrease when the variance error dominates the approximation error, especially for large b.

   (ii) For the definition of ITD, I don't think that  Ghadimi & Wang, 2018, Ji et al. (2021) or even the proposed method belong to the ITD. ITD means using the differentiation among the iterates of y while these methods are based on the implicit function that tells to use Jacobian.

   (iii) I don't understand why computing $J^{t+1}\nabla_y F(x^l,y^{l+1})$ needs $O(l^2)$ HVMs. It seems that by knowing $J^{l-1}$ only 1 HVM is needed.

2. The experiments should include the comparison with [1], a fully first-order method for bi-level optimization. Moreover, what's the dimension of w in the experiment？

3. Theorem 1 and 2 seem to be meaningless. Instead of sequence converging, it is more important to show where the sequence converges. Algorithm 2 should have a convergence analysis like Theorem 3.

4. Theorem 3 does not clearly state the algorithm whether it includes Guassian sampling or not.

5. Some assumptions are redundant. For example when F is $L_F$ Lipschitz, $\|\nabla_y F(x,y)\| \leq L_F$ automatically.

6. Does Equation (8) relate to the algorithm? According to the proof, it seems that the gradient and Hessian related to $x^k,y^k$ should be calculated with batch $S^k$ instead of $S^l$.

7. For remark 1, it seems that when Q>1, we should use direct computation instead of the Gaussian technique, which degrades the contribution of introducing this technique.

[1] Kwon, Jeongyeol, Dohyun Kwon, Stephen Wright, and Robert D. Nowak. "A fully first-order method for stochastic bilevel optimization." In International Conference on Machine Learning, pp. 18083-18113. PMLR, 2023.

---

> ### Author Response · Authors · 2024-09-24
>
> Response to weakness 1:
>
> (i) Thank you for pointing this out. We have now modified the claim to:
> "In particular, the intuition is to approximate the inverse $\nabla_{yy}G(x, y)^{-1}$ by $\sum_{i=1}^b \left(I - \nabla_{yy}G(x, y)\right)^i$ with some $b \in \mathbb{N}_+$,"
> in order to introduce the Neumann series.
>
> (ii) Thank you for your comment. We also view the iterative differentiation method as using differentiation among the iterates of $y$. The purpose of performing iterative differentiation on $y$ is to approximate the Jacobian of $y(x)$, where $y(x)$ is the solution to the lower-level problem when fixing $x$. Therefore, the key distinction that sets the ITD method apart from the AID method is the approximation of the Jacobian. From this perspective, our method falls into the category of ITD methods.
> Additionally, other previous works that makes use of the Neumann series to approximate the Jacobian—such as Lemma 1 in [Junyi Li et al., 2022]—demonstrate that this approach can be represented as a backward mode that calculates the iterative Jacobian of the iterates $y$. Thus, they can also be viewed as ITD-type methods. With this in mind, we would like to retain our definition while adding further explanation in the introduction.
>
> [Junyi Li, Bin Gu, Heng Huang: A Fully Single Loop Algorithm for Bilevel Optimization without Hessian Inverse. AAAI 2022: 7426-7434]
>
>
> (iii) We have two equivalent ways to update $J^{l+1} \nabla_y F(x^l, y^{l+1})$: either by using (5) and making use of $J^l$, or by using (6) without involving $J^l$.
>
> When applying deduction (5) in the algorithm, we need $n+1$ HVM computations. However, when using (5) to update $J^{l+1}$, we must compute $J^{l+1} = J^l \left(I - \gamma \nabla_{yy} G(x^l, y^l) \right) - \gamma \nabla_{xy} G(x^l, y^l)$, which involves an $\mathbb{R}^{n \times m} \times \mathbb{R}^{m \times m}$ matrix operation. When calculating $J^l \times \nabla_{yy} G(x^l, y^l)$, it requires $n$ HVM computations. Therefore, in each iteration, we actually need $n+1$ HVM computations, which is less than $O(k^2)$. Thus, our method is advantageous when $n+1 \leq k^2$.
>
> On the other hand, when using (6), we require $l^2$ HVM computations. In summary, our method outperforms the classical ITD method when either $n+1 \leq k^2$ or $l^2 < k^2$. We now add the above discussion about the complexity of using $J^l$ in the updates in the discussion following (6).
>
>
> Response to weakness 2: Thanks for mentioning this work. We now have added it in the experiments for comparison. As we can see from the new Figure 1, the our method is still outperforming the other methods in most cases.
>
> Response to weakness 3: The sequence convergence is an asymptotic property of an algorithm. Unlike complexity, it provides more detailed information about the behavior of the algorithm. For example, while complexity shows the rate at which the gradient approaches zero, it does not necessarily imply that the algorithm is stable, especially in non-convex settings. Even when the gradient is zero, the stationary point may not be unique, and the parameters may oscillate significantly or revolve around the stationary point. So, even when the gradient goes to zero, it does not mean that the parameters we train are stable. Our sequential analysis shows that if the objective function has certain general properties, our method guarantees that the parameters we train asymptotically converge to a unique point, even in non-convex settings. Therefore, it is still meaningful to analyze the asymptotic convergence of the iterates.
>
>
> Response to weakness 4. Thank you for pointing this out. Algorithm 3 uses (11), which is generated based on Gaussian sampling. We have now added more description to Theorem 3 to clarify this further.
>
> Response to weakness 5. Thank you for mentioning this. We agree with your observation. We intentionally separated them into two items for ease of reading.
>
> Response to weakness 6. Thank you for pointing this out. We use the batch $\mathcal{B}^k$ for the upper-level objective $F$ and the batch $\mathcal{S}^l$ for the lower-level objective $G$.
>
> Response to weakness 7. Thank you for mentioning this. It is true that when $Q > 1$, it becomes more computationally expensive. However, based on our experiments, we find that $Q = 1$ is efficient enough in practice.
>
> Responses for requested changes.
>
> 1. Please refer to the responses provided above.
>
> 2. We appreciate you pointing out the typos. We have now corrected these and other typos throughout the manuscript. Please refer to the respective blue revisions. Regarding (iv), the definition of $\bar{\nabla}$ should be $\hat{\nabla}$. $\hat{\nabla}{xy} G(x^l, y^l)$ and $\hat{\nabla}{yy} G(x^l, y^l)$ are defined in (9) and (10), respectively.

---

> > ### Comment · Reviewer_HiMK · 2024-10-09
> >
> > Thank you for the detailed clarification and explanation.  However, from my perspective, Theorem 1 and Theorem 2 should include the properties of the limit point. Otherwise, when we choose all learning rate to be 0, the algorithm can converge in one step, but it converges to some meaningless point. Therefore, I would suggest the author to discuss the properties of the limit point in Theorem 1 and 2. e.g. $H^*$ is the global optimal solution, or the limit point of $(x^l,y^l,J^l)$ satisfies some optimality condition.

---

> > > ### Author Response · Authors · 2024-10-09
> > >
> > > We thank the reviewer for the helpful suggestion. We now have added Corollary 1 in the main paper following Theorem 1, and included Remark 2 after Theorem 2 to highlight that the limiting point satisfies the first-order optimality condition. The corresponding proofs can be found in Corollary 2 (ii) in the appendix Section B.5. Please refer to the further revisions colored in teal.

---

> > > > ### Comment · Reviewer_HiMK · 2024-10-10
> > > >
> > > > Thank you for the further clarification. I have no more concerns.

---

> > > > > ### Author Response · Authors · 2024-10-10
> > > > >
> > > > > We sincerely appreciate your review. Your feedback has been very helpful in improving our manuscript. Thank you for your time and careful consideration of our work.

---

### Author Response · Authors · 2024-09-24

We sincerely thank the reviewers for their time and effort in reviewing our work. We have carefully addressed the concerns raised and revised the submission in line with the provided suggestions. The changes are highlighted in blue in the revised version of the manuscript for your reference.

---

### Public Comment · ~Stephen_J._Wright1 · 2025-08-29
**small technical point**

There is a typo at the end of the proof of Lemma 4 on p.18 concerning the relationship between $||A||$ and $||A||_F$.
Possibly relatedly, I believe that $\sqrt{n}$ in the bound (15) should be $n$.
This carries forward into much later analysis, but does not significantly impact the results, I believe.

---

> ### Author Response · Authors · 2025-09-26
>
> Thank you for pointing this out. You are correct --- the $\sqrt{n}$ in equation (15) should be $n$, and only Theorem 4 is affected. Specifically, all $\sqrt{n}$'s in Theorem 4 should be replaced by $n$. The analysis that makes use of Theorem 4 only relies on the fact that $H^l$ is non-increasing, so the rest of the proofs and conclusions should be safe. We appreciate your careful reading and helpful feedback.

---

### Decision · Action_Editor_EoeA · 2024-10-14

**Recommendation:** Accept as is

**Comment:**

This article aims to address the task of developing efficient methods for nonconvex bilevel optimization problems. Specifically, the authors focus on creating single-loop iterative differentiation (ITD) methods that can approximate the hypergradient while avoiding computationally expensive Hessian calculations. By proposing a Hessian-free method, they aim to reduce the number of backpropagations and lower the computational cost, while also ensuring convergence properties and maintaining efficiency compared to other state-of-the-art bilevel optimization approaches. The reviewers have all provided positive feedback. Through the rebuttal, the authors have addressed most of the reviewers' concerns. Therefore, this paper has reached a level suitable for acceptance.

**Audience:**

This article holds certain value for researchers working in optimization and gradient analysis.

**Claims And Evidence:**

The article makes the following claims:

1. The authors propose an efficient single-loop ITD-type method (ES-ITDM) that uses historical updates to approximate the hypergradient.
2. They introduce a new Hessian-free method based on ES-ITDM, which reduces computational cost by requiring fewer backpropagations.
3. The paper analyzes the convergence properties of the proposed methods, providing convergence rates based on the Kurdyka-Łojasiewicz (KL) property for ES-ITDM. And they demonstrate that the Hessian-free stochastic ES-ITDM achieves the best-known complexity while being computationally cheaper.

The authors provided some theoretical analysis and experimental validation for the above content. Although it is not possible to fully guarantee the correctness of every detail of the method, it can be concluded that there are no apparent errors overall. The reviewers also gave positive feedback.

---

> ### Author Response · Authors · 2024-10-22
>
> We sincerely thank the reviewers for their thoughtful feedback and constructive comments, which have significantly enhanced the quality of our work. We also appreciate the action editor for the valuable insights, guidance throughout the review process, and efforts in selecting appropriate reviewers.